# Towards Understanding the Dynamics of Gaussian–Stein Variational Gradient Descent

**Tianle Liu**
Department of Statistics
Harvard University
Cambridge, MA 02138
tianleliu@fas.harvard.edu

**Promit Ghosal**
Department of Mathematics
Massachusetts Institute of Technology
Waltham, MA 02453
promit@mit.edu

**Krishnakumar Balasubramanian**
Department of Statistics
University of California, Davis
Davis, CA 95616
kbala@ucdavis.edu

**Natesh S. Pillai**
Department of Statistics
Harvard University
Cambridge, MA 02138
pillai@fas.harvard.edu

## Abstract

Stein Variational Gradient Descent (SVGD) is a nonparametric particle-based deterministic sampling algorithm. Despite its wide usage, understanding the theoretical properties of SVGD has remained a challenging problem. For sampling from a Gaussian target, the SVGD dynamics with a bilinear kernel will remain Gaussian as long as the initializer is Gaussian. Inspired by this fact, we undertake a detailed theoretical study of the Gaussian–SVGD, i.e., SVGD projected to the family of Gaussian distributions via the bilinear kernel, or equivalently Gaussian variational inference (GVI) with SVGD. We present a complete picture by considering both the mean-field PDE and discrete particle systems. When the target is strongly log-concave, the mean-field Gaussian–SVGD dynamics is proven to converge linearly to the Gaussian distribution closest to the target in KL divergence. In the finite-particle setting, there is both uniform in time convergence to the mean-field limit and linear convergence in time to the equilibrium if the target is Gaussian. In the general case, we propose a density-based and a particle-based implementation of the Gaussian–SVGD, and show that several recent algorithms for GVI, proposed from different perspectives, emerge as special cases of our unifying framework. Interestingly, one of the new particle-based instance from this framework empirically outperforms existing approaches. Our results make concrete contributions towards obtaining a deeper understanding of both SVGD and GVI.

## 1 Introduction

Sampling from a given target density arises frequently in Bayesian statistics, machine learning and applied mathematics. Specifically, given a potential $V : \mathbb{R}^d \to \mathbb{R}$, the target density is given by

$$\rho(x) := Z^{-1}e^{-V(x)}, \quad \text{where} \quad Z := \int e^{-V(x)}dx \quad \text{is the normalizing constant.}$$

Traditionally-used Markov Chain Monte Carlo (MCMC) sampling algorithms are invariably not scalable to large-scale datasets [8, 61]. Variational inference and particle-based methods are two related alternatives proposed in the literature, both motivated by viewing sampling as optimization over the space of densities. We refer to [33, 39] for additional details related to this line of works.

In the literature on variational inference, recent efforts have focused on the Gaussian Variational Inference (GVI) problem. On the theoretical side, this is statistically motivated by the Bernstein-von

37th Conference on Neural Information Processing Systems (NeurIPS 2023).

Mises theorem, which posits that in the limit of large samples posterior distributions tend to be Gaussian distributed under certain regularity assumptions. We refer to [81, Chapter 10] for details of the classical results, and to [34, 73] for some recent non-asymptotic analysis. On the algorithmic side, efficient algorithms with both statistical and computational guarantees are developed for GVI [14, 20, 1, 35, 43, 19]. From a practical point-of-view, several works [64, 78, 80, 67] have shown superior performance of GVI, especially in the presence of large datasets.

Turning to particle-based methods, [53] proposed the Stein Variational Gradient Descent (SVGD) algorithm, a kernel-based deterministic approach for sampling. It has gained significant attention in the machine learning and applied mathematics communities due to its intriguing theoretical properties and wide applicability [23, 31, 56, 87]. Researchers have also developed variations of SVGD motivated by algorithmic and applied challenges [89, 49, 18, 28, 84, 12, 55, 71]. In its original form, SVGD could be viewed as a nonparametric variational inference method with a kernel-based practical implementation.

The flexibility offered by the *nonparametric* aspect of SVGD also leads to unintended consequences. On one hand, from a practical perspective, the question of how to pick the right kernel for implementing the SVGD algorithm is unclear. Existing approaches are mostly ad-hoc and do not provide clear instructions on the selection of kernels. On the other hand, developing a deeper theoretical understanding of SVGD dynamics is challenging due to its nonparametric formulation. Notably [58] derived the continuous-time PDE for the evolving density that emerges as the mean-field limit of the finite-particle SVGD systems, and shows the well-posedness of the PDE solutions. In general, the following different types of convergences could be examined regarding SVGD:

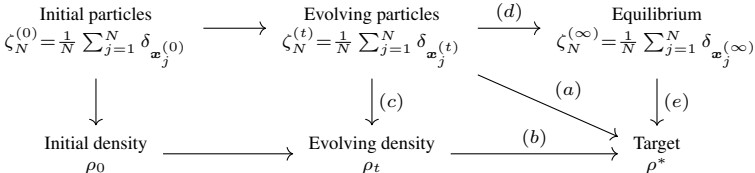

(a) Unified convergence of the empirical measure for $N$ finite particles to the continuous target as time $t$ and $N$ jointly grow to infinity;
(b) Convergence of mean-field SVGD to the target distribution over time;
(c) Convergence of the empirical measure for finite particles to the mean-field distribution at any finite given time $t \in [0, \infty)$;
(d) Convergence of finite-particle SVGD to the equilibrium over time;
(e) Convergence of the empirical measure for finite particles to the continuous target at time $t = \infty$.

From a practical point of view (a) is the ideal type of result that fully characterizes the algorithmic behavior of SVGD, which could be obtained by combining either (b) and (c) or (d) and (e). Regarding (b), [51] showed the convergence of mean-field SVGD in kernel Stein discrepancy (KSD, [17, 52, 29]), which is known to imply weak convergence under appropriate assumptions. [40, 15, 70, 75, 22] sharpened the results with weaker conditions or explicit rates. [32] extended the above result to the stronger Fisher information metric and Kullback–Leibler divergence based on a regularization technique. [58, 30, 40] obtained time-dependent mean-field convergence (c) of $N$ particles under various assumptions using techniques from the literature of *propagation of chaos*. [72] obtained even stronger results for (c) and combined (b) to get the first unified convergence (a) in terms of KSD. However, they have a rather slow rate $1/\sqrt{\log \log N}$, resulting from the fact that their bounds for (c) still depends on the time $t$ (sum of step sizes) double-exponentially. Moreover, there has not been any work that studies the convergence (d) and (e) for SVGD, which illustrate a new way to characterize the unified convergence (a).

In an attempt to overcome the drawbacks of the nonparametric formulation of SVGD and also taking cue from the GVI literature, in this work we study the dynamics of Gaussian–SVGD, a parametric formulation of SVGD. Our contributions in this work are three-fold:

- *Mean-field results*: We study the dynamics of Gaussian–SVGD in the mean-field setting and establish linear convergence for both Gaussian and strongly log-concave targets. As an example of the obtained results, Table 1 shows the convergence rates of covariance for centered Gaussian

Table 1: Convergence rates of SVGD with different bilinear kernels for Gaussian families.

| | $K_1$-SVGD | $K_2$-SVGD | WGF ($K_3$-SVGD) | R-SVGD ($K_4$-SVGD) |
|---|---|---|---|---|
| Centered Gaussian | $\mathcal{O}\big(e^{-2t}\big)$ [3.2] | $\mathcal{O}(e^{-2t})$ [F.2] | $\mathcal{O}(e^{-\frac{2t}{\lambda}})$ [F.1] | $\mathcal{O}\big(e^{-\frac{2t}{(1-\nu)\lambda+\nu}}\big)$ [3.4] |
| General Gaussian | Theorem 3.1 | $\mathcal{O}\big(e^{-\frac{1}{\lambda}\wedge 2t}\big)$ [F.2] | $\mathcal{O}\big(e^{-\frac{t}{\lambda}}\big)$ [F.1] | $\mathcal{O}\big(e^{-\frac{1}{\lambda}\wedge\frac{2}{(1-\nu)\lambda+\nu}t}\big)$ [3.4] |

families for several revelant algorithms. All of them will be introduced later in Sections 2 and 3. We also establish the well-posedness of the solutions for the mean-field PDE and discrete particle systems that govern SVGD with bilinear kernels (see Appendix C). Prior work [58] requires that the kernel be radial which rules out the important class of bilinear kernels that we consider. [32] relaxed the radial kernel assumption, however, they required boundedness assumptions which we avoid in this work for the case of bilinear kernels.

- ***Finite-particle results***: We study the finite-particle SVGD systems in both continuous and discrete time for Gaussian targets. We show that for SVGD with a bilinear kernel if the target and initializer are both Gaussian, the mean-field convergence can be uniform in time (See Theorem 3.7). To the best of our knowledge, this is the first uniform in time result for SVGD dynamics and should be contrasted with the double exponential dependency on $t$ for nonparametric SVGD [58, 72]. Our numerical simulations suggest that similar results should hold for certain classes of non-Gaussian target as well for Gaussian–SVGD. We also study the convergence (d) by directly solving the finite-particle systems (See Theorems 3.6 and 3.8). Moreover, in Theorem 3.10, assuming centered Gaussian targets, we obtain a linear rate for covariance convergence in the finite-particles, discrete-time setting, precisely characterizing the step size choice for the practical algorithm.

- ***Unifying algorithm frameworks***: We propose two unifying algorithm frameworks for finite-particle, discrete-time implementations of the Gaussian–SVGD dynamics. The first approach assumes access to samples from Gaussian densities with the mean and covariance depending on the current time instance of the dynamics. The second is a purely particle-based approach, in that, it assumes access only to an initial set of samples from a Gaussian density. In particular, we show that three previously proposed methods from [27, 43] for GVI emerge as special cases of the proposed frameworks, by picking different bilinear kernels, thereby further strengthening the connections between GVI and the kernel choice in SVGD. Furthermore, we conduct experiments for eight algorithms that can be implied from our framework, and observe that the particle-based algorithms are invariably more stable than density-based ones. Notably one of the new particle-based algorithms emerging from our analysis outperforms existing approaches.

## 2   Preliminaries

Denote the space of probability densities on $\mathbb{R}^d$ by $\mathcal{P}(\mathbb{R}^d) := \big\{\rho \in \mathcal{F}(\mathbb{R}^d) : \int \rho \,\mathrm{d}\boldsymbol{x} = 1, \rho \geq 0\big\}$, where $\mathcal{F}(\mathbb{R}^d)$ is the set of smooth functions. As studied in [42], $\mathcal{P}(\mathbb{R}^d)$ forms a Fréchet manifold called the density manifold. For any "point" $\rho \in \mathcal{P}(\mathbb{R}^d)$, we denote the tangent space and cotangent space at $\rho$ by $T_\rho\mathcal{P}(\mathbb{R}^d)$ and $T_\rho^*\mathcal{P}(\mathbb{R}^d)$ respectively. A Riemannian metric tensor assigns to each $\rho \in \mathcal{P}(\mathbb{R}^d)$ a positive definite inner product $g_\rho : T_\rho\mathcal{P}(\mathbb{R}^d) \times T_\rho\mathcal{P}(\mathbb{R}^d) \to \mathbb{R}$ and uniquely corresponds to an isomorphism $G_\rho$ (called the canonical isomorphism) between the tangent and cotangent bundles [66], i.e., we have $G_\rho : T_\rho\mathcal{P}(\mathbb{R}^d) \to T_\rho^*\mathcal{P}(\mathbb{R}^d)$ .

**Definition 2.1** (Wasserstein metric)**.** *The Wasserstein metric is induced by the following canonical isomorphism* $G_\rho^{\mathrm{Wass}} : T_\rho\mathcal{P}(\mathbb{R}^d) \to T_\rho^*\mathcal{P}(\mathbb{R}^d)$ *such that*

$$(G_\rho^{\mathrm{Wass}})^{-1}\Phi = -\nabla \cdot (\rho\nabla\Phi), \quad \Phi \in T_\rho^*\mathcal{P}(\mathbb{R}^d).$$

The Wasserstein gradient flow (WGF) can be seen as the natural gradient flow for KL divergence on the density manifold with respect to the Wasserstein metric. In specific, the mean-field PDE is given by the linear Fokker-Planck equation [33]:

$$\dot{\rho}_t = -(G_{\rho_t}^{\mathrm{Wass}})^{-1}\frac{\delta}{\delta\rho_t}\mathrm{KL}(\rho_t \parallel \rho^*) = \nabla \cdot \Big(\rho_t\nabla\frac{\delta}{\delta\rho_t}\mathrm{KL}(\rho_t \parallel \rho^*)\Big) = \nabla \cdot (\nabla\rho_t + \rho_t\nabla V), \quad (1)$$

where $\frac{\delta}{\delta \rho_t}$ denotes the variational derivative with respect to $\rho_t$, $\mathrm{KL}(\rho \parallel \rho^*)$ is the so-called energy function, and $V(\boldsymbol{x})$ is the potential function that satisfies $\rho^*(\boldsymbol{x}) \propto \exp(-V(\boldsymbol{x}))$.

Interestingly, the mean field flow of SVGD can also be seen as a gradient flow for the KL divergence but with respect to the Stein metric, where we perform kernelization in the cotangent space before taking the divergence [51, 22].

**Definition 2.2** (Stein metric). *The Stein metric is induced by the following canonical isomorphism* $G_\rho^{\mathrm{Stein}} : T_\rho \mathcal{P}(\mathbb{R}^d) \to T_\rho^* \mathcal{P}(\mathbb{R}^d)$ *such that*

$$(G_\rho^{\mathrm{Stein}})^{-1}\Phi = -\nabla \cdot \left( \rho(\cdot) \int K(\cdot, \boldsymbol{y})\rho(\boldsymbol{y})\nabla\Phi(\boldsymbol{y}) \, \mathrm{d}\boldsymbol{y} \right), \quad \Phi \in T_\rho^* \mathcal{P}(\mathbb{R}^d),$$

*where* $K : \mathbb{R}^d \times \mathbb{R}^d \to \mathbb{R}$ *is a positive-definite kernel.*

In particular, the mean field PDE of the SVGD algorithm can be written as

$$\dot{\rho}_t = -(G_{\rho_t}^{\mathrm{Stein}})^{-1}\frac{\delta}{\delta \rho_t} \mathrm{KL}(\rho_t \parallel \rho^*) = \nabla \cdot \left( \rho_t(\cdot) \int K(\cdot, \boldsymbol{y})\big(\nabla\rho_t(\boldsymbol{y}) + \rho_t(\boldsymbol{y})\nabla V(\boldsymbol{y})\big) \, \mathrm{d}\boldsymbol{y} \right). \quad (2)$$

**Gaussian Families as Submanifolds.** We consider the family of (multivariate) Gaussian densities $\rho_\theta \in \mathcal{P}(\mathbb{R}^d)$ where $\theta = (\boldsymbol{\mu}, \Sigma) \in \Theta = \mathbb{R}^d \times \mathrm{Sym}^+(d, \mathbb{R})$. Note that $\mathrm{Sym}^+(d, \mathbb{R})$ is the set of (symmetric) positive definite $d \times d$ matrices. In this way, $\Theta$ can be identified as a Riemannian submanifold of the density manifold $\mathcal{P}(\mathbb{R}^d)$ with the induced Riemannian structure. If we further restrict the Gaussian family to have zero mean, it further induces the submanifold $\Theta_0 = \mathrm{Sym}^+(d, \mathbb{R})$. Notably the Wasserstein metric on the density manifold induces the Bures–Wasserstein metric for the Gaussian families [6, 82]. In our paper, however, we consider the induced Stein metric on $\Theta$ or $\Theta_0$ (we call it the **Gaussian–Stein metric**; for details see Appendix B).

**Different Bilinear Kernels and Induced Metrics.** There are several different bilinear kernels that appear in literature. [54] considers the simple bilinear kernel $K_1(\boldsymbol{x}, \boldsymbol{y}) = \boldsymbol{x}^\top \boldsymbol{y} + 1$ while [27, 13] suggest the use of an affine-invariant bilinear kernel $K_2(\boldsymbol{x}, \boldsymbol{y}) = (\boldsymbol{x} - \boldsymbol{\mu})^\top(\boldsymbol{y} - \boldsymbol{\mu}) + 1$. [13] further points out that with the rescaled affine-invariant kernel $K_3(\boldsymbol{x}, \boldsymbol{y}) = (\boldsymbol{x} - \boldsymbol{\mu})^\top \Sigma^{-1}(\boldsymbol{y} - \boldsymbol{\mu}) + 1$, the Gaussian–Stein metric magically coincides with the Bures–Wasserstein metric on $\Theta$ (not true on the whole density manifold). Note that here $\boldsymbol{\mu}$ and $\Sigma$ are the mean and covariance of the current mean-field Gaussian distribution which could change with time. Moreover, [32] proposed a regularized version of SVGD (R-SVGD) that interpolates the dynamics between WGF and SVGD. Interestingly R-SVGD with the affine-invariant kernel $K_2$ for Gaussian families can be reformulated as Gaussian–SVGD with a new kernel $K_4(\boldsymbol{x}, \boldsymbol{y}) = (\boldsymbol{x} - \boldsymbol{\mu})^\top((1 - \nu)\Sigma + \nu I)^{-1}(\boldsymbol{y} - \boldsymbol{\mu})$, which interpolates between $K_2$ and $K_3$ (see Theorem 3.4). For clarity we present the results for $K_1$ in the main article while leave the analogues for $K_2$–$K_4$ in the appendix. We also point out that $K_1$ and $K_2$ are the same on $\Theta_0$.

**Gaussian–Stein Variational Gradient Descent.** With a bilinear kernel and Gaussian targets, we will prove in the next subsection that the SVGD dynamics would remain Gaussian as long as the initializer is Gaussian. However, this is not true in more general situations especially when the target is non-Gaussian. Fortunately for Gaussian variational inference we can still consider the gradient flow restricted to the Gaussian submanifold. In general, we denote by $G_\theta^{\mathrm{Stein}}$ the canonical isomorphism on $\Theta$ induced by $G_\rho^{\mathrm{Stein}}$, and define the Gaussian–Stein variational gradient descent as

$$\dot{\theta}_t = -(G_{\theta_t}^{\mathrm{Stein}})^{-1}\nabla_{\theta_t} \mathrm{KL}(\rho_{\theta_t} \parallel \rho^*),$$

where $\rho^*$ might not be a Gaussian density. Notably Gaussian–SVGD solves the following optimization problem

$$\min_{\theta \in \Theta} \mathrm{KL}(\rho_\theta \parallel \rho^*), \quad \text{where } \rho_\theta \text{ is the density of } \mathcal{N}(\boldsymbol{\mu}, \Sigma),$$

via gradient descent under the Gaussian–Stein metric.

## 3 Dynamics of Gaussian–SVGD for Gaussian Targets

### 3.1 Mean-Field Analysis from WGF to SVGD

The Wasserstein gradient flow (WGF) restricted to the general Gaussian family $\Theta$ is known as the Bures–Wasserstein gradient flow [43, 19]. For consistency in this subsection we always set

the initializer to be $\mathcal{N}(\boldsymbol{\mu}_0, \Sigma_0)$ with density $\rho_0$ and target $\mathcal{N}(\boldsymbol{b}, Q)$ with density $\rho^*$ for the general Gaussian family. Then the WGF at any time $t$ remains Gaussian, and can be fully characterized by the following dynamics of the mean $\boldsymbol{\mu}_t$ and covariance matrix $\Sigma_t$:

$$\dot{\boldsymbol{\mu}}_t = -Q^{-1}(\boldsymbol{\mu}_t - \boldsymbol{b}), \qquad \dot{\Sigma}_t = 2I - \Sigma_t Q^{-1} - Q^{-1}\Sigma_t. \tag{3}$$

For SVGD with bilinear kernels we have similar results:

**Theorem 3.1** (SVGD). *For any $t \geq 0$ the solution $\rho_t$ of SVGD (2) with the bilinear kernel $K_1$ remains a Gaussian density with mean $\boldsymbol{\mu}_t$ and covariance matrix $\Sigma_t$ given by*

$$\begin{cases} \dot{\boldsymbol{\mu}}_t = (I - Q^{-1}\Sigma_t)\boldsymbol{\mu}_t - (1 + \boldsymbol{\mu}_t^\top \boldsymbol{\mu}_t) Q^{-1}(\boldsymbol{\mu}_t - \boldsymbol{b}) \\ \dot{\Sigma}_t = 2\Sigma_t - \Sigma_t \left( \Sigma_t + \boldsymbol{\mu}_t(\boldsymbol{\mu}_t - \boldsymbol{b})^\top \right) Q^{-1} - Q^{-1} \left( \Sigma_t + (\boldsymbol{\mu}_t - \boldsymbol{b})\boldsymbol{\mu}_t^\top \right) \Sigma_t \end{cases}, \tag{4}$$

*which has a unique global solution on $[0, \infty)$ given any $\boldsymbol{\mu}_0 \in \mathbb{R}^d$ and $\Sigma_0 \in \mathrm{Sym}^+(d, \mathbb{R})$. And $\rho_t$ converges weakly to $\rho^*$ as $t \to \infty$ at the following rates*

$$\|\boldsymbol{\mu}_t - \boldsymbol{b}\| = \mathcal{O}(e^{-2(\gamma - \epsilon)t}), \quad \|\Sigma_t - Q\| = \mathcal{O}(e^{-2(\gamma - \epsilon)t}), \quad \forall \epsilon > 0,$$

*where $\gamma$ is the smallest eigenvalue of the matrix*

$$\begin{bmatrix} I_{d^2} & \frac{1}{\sqrt{2}}\boldsymbol{b} \otimes Q^{-1/2} \\ \frac{1}{\sqrt{2}}\boldsymbol{b}^\top \otimes Q^{-1/2} & \frac{1}{2}(1 + \boldsymbol{b}^\top \boldsymbol{b})Q^{-1} \end{bmatrix} \text{ with a lower bound } \gamma > \frac{1}{1 + \boldsymbol{b}^\top \boldsymbol{b} + 2\lambda},$$

*where $\lambda$ is the largest eigenvalue of $Q$.*

Note that for any vector $\boldsymbol{x}$, $\|\boldsymbol{x}\|$ denotes its Euclidean norm and for any matrix $A$ we use $\|A\|$ for its spectral norm, $\|A\|_*$ for the nuclear norm and $\|A\|_F$ for the Frobenius norm. All matrix convergence are considered under the spectral norm in default for technical simplicity (though all matrix norms are equivalent in finite dimensions).

If we restrict to the centered Gaussian family where both the initializer and target have zero mean (setting $\boldsymbol{\mu}_0 = \boldsymbol{b} = \boldsymbol{0}$), the dynamics can further be simplified.

**Theorem 3.2** (SVGD for centered Gaussian). *Let $\rho_0$ and $\rho^*$ be two centered Gaussian densities. Then for any $t \geq 0$ the solution $\rho_t$ of SVGD (2) with the bilinear kernel $K_1$ or $K_2$ remains a centered Gaussian density with the covariance matrix $\Sigma_t$ given by the following Riccati type equation:*

$$\dot{\Sigma}_t = 2\Sigma_t - \Sigma_t^2 Q^{-1} - Q^{-1}\Sigma_t^2, \tag{5}$$

*which has a unique global solution on $[0, \infty)$ given any $\Sigma_0, Q \in \mathrm{Sym}^+(d, \mathbb{R})$. If $\Sigma_0 Q = Q\Sigma_0$, we have the closed-form solution:*

$$\Sigma_t^{-1} = e^{-2t}\Sigma_0^{-1} + (1 - e^{-2t})Q^{-1}. \tag{6}$$

*In particular, if we let $\Sigma_0 = I$ and $Q = I + \eta \boldsymbol{v}\boldsymbol{v}^\top$ for some $\eta > 0$ and $\boldsymbol{v} \in \mathbb{R}^d$ such that $\boldsymbol{v}^\top \boldsymbol{v} = 1$, then $\Sigma_t$ can be rewritten as*

$$\Sigma_t = I + \frac{\eta(1 - e^{-2t})}{1 + \eta e^{-2t}} \boldsymbol{v}\boldsymbol{v}^\top. \tag{7}$$

[13] shows that for WGF if $\Sigma_0 Q = Q\Sigma_0$, then we have $\|\boldsymbol{\mu}_t - \boldsymbol{b}\| = \mathcal{O}(e^{-t/\lambda})$ and $\|\Sigma_t - Q\| = \mathcal{O}(e^{-2t/\lambda})$. For the centered Gaussian family SVGD converges faster if $\lambda > 1$. For the general Gaussian family WGF and SVGD have rather comparable rates (e.g., take $\lambda \gg \|\boldsymbol{b}\|$ then the lower bound here is roughly $\mathcal{O}(e^{-t/\lambda})$). Another observation is that the WGF rates depend on $Q$ alone but the SVGD rates here sometimes also depend on $\boldsymbol{b}$, which breaks the affine invariance of the system. This is a problem originated from the choice of kernels as addressed in [13], where they propose to use $K_2$ instead of $K_1$. Such approach has both advantages and disadvantages. The convention in SVGD is that the kernel should not depend on the mean-field density because the density is usually unknown and changes with time. But for GVI the affine-invariant bilinear kernel $K_2$ only requires estimating the means from Gaussian distributions and is not a big issue.

**Regularized Stein Variational Gradient Descent.** In Section 2 we show that SVGD can be regarded as WGF kernelized in the cotangent space $T_\rho^* \mathcal{P}(\mathbb{R}^d)$. The regularized Stein variational gradient descent (R-SVGD) [32] interpolates WGF and SVGD by pulling back part of the kernelized gradient of the cotangent vector $\Phi$, which is also seen as gradient flows under the regularized Stein metric:

**Definition 3.3** (Regularized Stein metric). *The regularized Stein metric is induced by the following canonical map*

$$(G_\rho^{\mathrm{RS}})^{-1}\Phi := -\nabla \cdot \left( \rho \left( (1-\nu)\mathcal{T}_{K,\rho} + \nu I \right)^{-1} \mathcal{T}_{K,\rho} \nabla \Phi \right),$$

*where $\mathcal{T}_{K,\rho}$ is the kernelization operator given by $\mathcal{T}_{K,\rho}f := \int K(\cdot, \boldsymbol{y})f(\boldsymbol{y})\rho(\boldsymbol{y})\,\mathrm{d}\boldsymbol{y}$.*

The R-SVGD is defined as

$$\dot{\rho}_t = -(G_{\rho_t}^{\mathrm{RS}})^{-1}\tfrac{\delta}{\delta\rho_t}\mathrm{KL}(\rho_t \parallel \rho^*) = \nabla \cdot \left( \rho_t \left( (1-\nu)\mathcal{T}_{K,\rho_t} + \nu I \right)^{-1} \mathcal{T}_{K,\rho_t} \nabla \log \tfrac{\rho_t}{\rho^*} \right). \quad (8)$$

**Theorem 3.4** (R-SVGD). *Let $\rho_0$ and $\rho^*$ be two Gaussian densities. Then the solution $\rho_t$ of R-SVGD (8) with the bilinear kernel $K_2$ converges to $\rho^*$ as $t \to \infty$, and $\rho_t$ is the density of $\mathcal{N}(\boldsymbol{b}, \Sigma_t)$ with*

$$\begin{cases} \dot{\boldsymbol{\mu}}_t = -Q^{-1}(\boldsymbol{\mu}_t - \boldsymbol{b}) \\ \dot{\Sigma}_t = 2((1-\nu)\Sigma_t + \nu I)^{-1}\Sigma_t - ((1-\nu)\Sigma_t + \nu I)^{-1}\Sigma_t^2 Q^{-1} - Q^{-1}((1-\nu)\Sigma_t + \nu I)^{-1}\Sigma_t^2 \end{cases}. \quad (9)$$

*If $\Sigma_0 Q = Q\Sigma_0$, we have $\|\Sigma_t - Q\| = \mathcal{O}\left(e^{-2t/((1-\nu)\lambda + \nu)}\right)$, where $\lambda$ is the largest eigenvalue of $Q$.*

From this theorem we see that R-SVGD can take the advantage of both regimes by choosing $\nu$ wisely. Another interesting connection is that on $\Theta$ the induced regularized Stein metric coincides with the Stein metric with a different kernel $K_4(\boldsymbol{x}, \boldsymbol{y}) = (\boldsymbol{x} - \boldsymbol{\mu})^\top((1-\nu)\Sigma + \nu I)^{-1}(\boldsymbol{y} - \boldsymbol{\mu}) + 1$ (see Theorem B.7).

**Stein AIG Flow.** Accelerating methods are widely used in first-order optimization algorithms and have attracted considerable interest in particle-based variational inference [50]. [76, 86] study the accelerated information gradient (AIG) flows as the analogue of Nesterov's accelerated gradient method [63] on the density manifold. Given a probability space $\mathcal{P}(\mathbb{R}^d)$ with a metric tensor $g_\rho(\cdot, \cdot)$, let $G_\rho : T_\rho\mathcal{P}(\mathbb{R}^d) \to T_\rho^*\mathcal{P}(\mathbb{R}^d)$ be the corresponding isomorphism. The Hamiltonian flow in probability space [16] follows from

$$\partial_t \begin{bmatrix} \rho_t \\ \Phi_t \end{bmatrix} = \begin{bmatrix} 0 & 1 \\ -1 & 0 \end{bmatrix} \begin{bmatrix} \frac{\delta}{\delta\rho_t}\mathcal{H}(\rho_t, \Phi_t) \\ \frac{\delta}{\delta\Phi_t}\mathcal{H}(\rho_t, \Phi_t) \end{bmatrix}, \text{ where } \mathcal{H}(\rho_t, \Phi_t) := \tfrac{1}{2}\int \Phi_t G_{\rho_t}^{-1}\Phi_t\,\mathrm{d}\boldsymbol{x} + \mathrm{KL}(\rho \parallel \rho^*)$$

is called the Hamiltonian function, which consists of a kinetic energy $\tfrac{1}{2}\int \Phi G_\rho^{-1}\Phi\,\mathrm{d}\boldsymbol{x}$ and a potential energy $\mathrm{KL}(\rho \parallel \rho^*)$. Following [86] we introduce the accelerated information gradient flow in probability space. Let $\alpha_t \geq 0$ be a scalar function of time $t$. We add a damping term $\alpha_t\Phi_t$ to the Hamiltonian flow:

$$\partial_t \begin{bmatrix} \rho_t \\ \Phi_t \end{bmatrix} = -\begin{bmatrix} 0 \\ \alpha_t\Phi_t \end{bmatrix} + \begin{bmatrix} 0 & 1 \\ -1 & 0 \end{bmatrix} \begin{bmatrix} \frac{\delta}{\delta\rho_t}\mathcal{H}(\rho_t, \Phi_t) \\ \frac{\delta}{\delta\Phi_t}\mathcal{H}(\rho_t, \Phi_t) \end{bmatrix}, \quad (10)$$

By adopting the Stein metric we obtain the Stein AIG flow (S-AIGF):

$$\begin{cases} \dot{\rho}_t = -\nabla \cdot \left( \rho_t(\cdot) \int K(\cdot, \boldsymbol{y})\rho_t(\boldsymbol{y})\nabla\Phi_t(\boldsymbol{y})\,\mathrm{d}\boldsymbol{y} \right) \\ \dot{\Phi}_t = -\alpha_t\Phi_t - \int \nabla\Phi_t(\cdot)^\top\nabla\Phi_t(\boldsymbol{y})K(\cdot, \boldsymbol{y})\rho_t(\boldsymbol{y})\,\mathrm{d}\boldsymbol{y} - \frac{\delta}{\delta\rho_t}\mathrm{KL}(\rho_t \parallel \rho^*) \end{cases}. \quad (11)$$

Again we characterize the dynamics of S-AIGF with the linear kernel for the Gaussian family:

**Theorem 3.5** (S-AIGF). *Let $\rho_0$ and $\rho^*$ be two centered Gaussian densities. Then the solution $\rho_t$ of S-AIGF (11) with the bilinear kernel $K_1$ or $K_2$ is the density of $\mathcal{N}(\boldsymbol{0}, \Sigma_t)$ where $\Sigma_t$ satisfies*

$$\begin{cases} \dot{\Sigma}_t = 2(S_t\Sigma_t^2 + \Sigma_t^2 S_t) \\ \dot{S}_t = -\alpha_t S_t - 2(S_t^2\Sigma_t + \Sigma_t S_t^2) + \tfrac{1}{2}(\Sigma_t^{-1} - Q^{-1}) \end{cases}, \quad (12)$$

*where $S_t \in \mathrm{Sym}(d, \mathbb{R})$ with initial value $S_0 = 0$.*

Note that the convergence properties of S-AIGF still remains open in contrast to the Wasserstein AIG flow (W-AIGF) as [86] shows that the W-AIGF for the centered Gaussian family is

$$\dot{\Sigma}_t = 2(S_t\Sigma_t + \Sigma_t S_t), \qquad \dot{S}_t = -\alpha_t S_t - 2S_t^2 + \tfrac{1}{2}(\Sigma_t^{-1} - Q^{-1}),$$

and that if $\alpha_t$ is well-chosen, the KL divergence in W-AIGF converges at the rate of $\mathcal{O}(e^{-t/\sqrt{\lambda}})$. Thus, when $\lambda$ is large it converges faster than WGF. It is also interesting to point out that the acceleration effect of Nesterov's scheme also comes from time discretization of the ODE system (see [74]) as it moves roughly $\sqrt{\epsilon}$ rather than $\epsilon$ along the gradient path when the step size is $\epsilon$.

## 3.2 Finite-Particle Systems

In this subsection, we consider the case where $N < \infty$ particles evolve in time $t$. We set a Gaussian target $\mathcal{N}(\boldsymbol{b}, Q)$ (i.e., the potential is $V(\boldsymbol{x}) = \frac{1}{2}(\boldsymbol{x} - \boldsymbol{b})^\top Q^{-1}(\boldsymbol{x} - \boldsymbol{b})$) and run the SVGD algorithm with a bilinear kernel, and obtain the dynamics of $\boldsymbol{x}_1^{(t)}, \cdots, \boldsymbol{x}_N^{(t)}$. The continuous-time particle-based SVGD corresponds to the following deterministic interactive system in $\mathbb{R}^d$:

$$\dot{\boldsymbol{x}}_i^{(t)} = \frac{1}{N} \sum_{j=1}^N \nabla_{\boldsymbol{x}_j} K\left(\boldsymbol{x}_i^{(t)}, \boldsymbol{x}_j^{(t)}\right) - \frac{1}{N} \sum_{j=1}^N K\left(\boldsymbol{x}_i^{(t)}, \boldsymbol{x}_j^{(t)}\right) \nabla V\left(\boldsymbol{x}_j^{(t)}\right) \tag{13}$$

with initial particles given by $\boldsymbol{x}_i^{(0)}$ $(i = 1, \cdots, N)$. Now denoting the sample mean and covariance matrix at time $t$ by $\boldsymbol{\mu}_t := \frac{1}{N} \sum_{j=1}^N \boldsymbol{x}_j^{(t)}$ and $C_t := \frac{1}{N} \sum_{j=1}^N \boldsymbol{x}_j^{(t)} \boldsymbol{x}_j^{(t)\top} - \boldsymbol{\mu}_t \boldsymbol{\mu}_t^\top$, we have the following theorem.

**Theorem 3.6** (SVGD). *Suppose the initial particles satisfy that $C_0$ is non-singular. Then SVGD (13) with the bilinear kernel $K_1$ and Gaussian potential $V$ has a unique solution given by*

$$\boldsymbol{x}_i^{(t)} = A_t(\boldsymbol{x}_i^{(0)} - \boldsymbol{\mu}_0) + \boldsymbol{\mu}_t, \tag{14}$$

*where $A_t$ is the unique (matrix) solution of the linear system*

$$\dot{A}_t = \left(I - Q^{-1}(C_t + \boldsymbol{\mu}_t \boldsymbol{\mu}_t^\top) + Q^{-1} \boldsymbol{b} \boldsymbol{\mu}_t^\top\right) A_t, \quad A_0 = I, \tag{15}$$

*and $\boldsymbol{\mu}_t$ and $C_t$ are the unique solution of the ODE system*

$$\begin{cases} \dot{\boldsymbol{\mu}}_t = (I - Q^{-1} C_t) \boldsymbol{\mu}_t - (1 + \boldsymbol{\mu}_t^\top \boldsymbol{\mu}_t) Q^{-1}(\boldsymbol{\mu}_t - \boldsymbol{b}) \\ \dot{C}_t = 2C_t - C_t \left(C_t + \boldsymbol{\mu}_t(\boldsymbol{\mu}_t - \boldsymbol{b})^\top\right) Q^{-1} - Q^{-1} \left(C_t + (\boldsymbol{\mu}_t - \boldsymbol{b}) \boldsymbol{\mu}_t^\top\right) C_t \end{cases}. \tag{16}$$

The ODE system (16) is exactly the same as that in the density flow (4). Thus, we have the the same convergence rates as in Theorem 3.1. Theorem 3.6 can be interpreted as: At each time $t$ the particle positions are a linear transformation of the initialization. On one hand, if we initialize *i.i.d.* from Gaussian, there is uniform in time convergence as shown in the theorem below.

On the other hand, if we initialize *i.i.d.* from a non-Gaussian distribution $\rho_0$. At each time $t$ the mean field limit $\rho_t$ should be a linear transformation of $\rho_0$ and cannot converge to the Gaussian target $\rho^*$ as $t \to \infty$. Note that in general SVGD with the bilinear kernel might not always converge to the target distribution for nonparametric sampling but for GVI there is no such issue. This will be discussed in detail in Appendix C together with general results of well-posedness and mean-field convergence of SVGD with the bilinear kernel, which has not yet been studied in literature.

**Theorem 3.7** (Uniform in time convergence). *Given the same setting as Theorem 3.6, further suppose the initial particles are drawn* i.i.d. *from $\mathcal{N}(\boldsymbol{\mu}_0, \Sigma_0)$. Then there exists a constant $C_{d,Q,\boldsymbol{b},\Sigma_0,\boldsymbol{\mu}_0}$ such that for all $t \in [0, \infty]$, for all $N \geq 2$, with the empirical measure $\zeta_N^{(t)} = \frac{1}{N} \sum_{i=1}^N \delta_{\boldsymbol{x}_i^{(t)}}$, the second moment of Wasserstein-2 distance between $\zeta_N^{(t)}$ and $\rho_t$ converges:*

$$\mathbb{E}\left[\mathcal{W}_2^2\left(\zeta_N^{(t)}, \rho_t\right)\right] \leq C_{d,Q,\boldsymbol{b},\Sigma_0,\boldsymbol{\mu}_0} \times \begin{cases} N^{-1} \log \log N & \text{if } d = 1 \\ N^{-1} (\log N)^2 & \text{if } d = 2 \\ N^{-2/d} & \text{if } d \geq 3 \end{cases}. \tag{17}$$

Similar to Theorem 3.2, we also provide the finite-particle result for a centered Gaussian target.

**Theorem 3.8** (SVGD for centered Gaussian). *Suppose the SVGD particle system (13) with the bilinear kernel $K_1$ or $K_2$ is targeting a centered Gaussian distribution and initialized by $\left(\boldsymbol{x}_i^{(0)}\right)_{i=1}^N$ such that $\boldsymbol{\mu}_0 = \boldsymbol{0}$ and $C_0 Q = Q C_0$. Then we have the following closed-form solution*

$$\boldsymbol{x}_i^{(t)} = \left(e^{-2t} I + (1 - e^{-2t}) Q^{-1} C_0\right)^{-1/2} \boldsymbol{x}_i^{(0)}. \tag{18}$$

**Analogous Result for R-SVGD.** Next we consider the particle dynamics of R-SVGD. As shown in [32], the finite-particle system of R-SVGD is

$$\dot{X}_t = -\left((1 - \nu) \frac{K_t}{N} + \nu I\right)^{-1} \left(\frac{K_t}{N} \mathcal{L}_t \nabla V - \frac{1}{N} \sum_{j=1}^N \mathcal{L}_t \nabla K\left(\boldsymbol{x}_j^{(t)}, \cdot\right)\right),$$

where $X_t := \left(\boldsymbol{x}_1^{(t)}, \cdots, \boldsymbol{x}_N^{(t)}\right)^\top$, $\mathcal{L}_t f := \left(f(\boldsymbol{x}_i^{(t)}), \cdots, f(\boldsymbol{x}_N^{(t)})\right)^\top$ for all $f : \mathbb{R}^d \to \mathbb{R}^d$ and $(K_t)_{ij} := K\left(\boldsymbol{x}_i^{(t)}, \boldsymbol{x}_j^{(t)}\right)$ for all $1 \leq i, j \leq N$. Similar to Theorem 3.6, we have the following result:

**Theorem 3.9** (R-SVGD). *Suppose the R-SVGD system* (13) *with $K_1$ or $K_2$ is targeting a centered Gaussian distribution and initialized by $\left(\boldsymbol{x}_i^{(0)}\right)_{i=1}^{N}$ such that $\boldsymbol{\mu}_0 = \boldsymbol{0}$. Then we have $\boldsymbol{x}_i^{(t)} = A_t \boldsymbol{x}_i^{(0)}$ where $A_t$ is the unique solution of the linear system*

$$\dot{A}_t = (I - Q^{-1}C_t)((1-\nu)C_t + \nu I)^{-1}A_t, \quad A_0 = I, \tag{19}$$

*and the sample covariance matrix $C_t$ is given by*

$$\dot{C}_t = 2((1-\nu)C_t + \nu I)^{-1}C_t - ((1-\nu)C_t + \nu I)^{-1}C_t^2 Q^{-1} - Q^{-1}((1-\nu)C_t + \nu I)^{-1}C_t^2. \tag{20}$$

Again we observe that the particles at time $t$ is a time-changing linear transformation of the initializers.

**Discrete-time Analysis for Finite Particles.** Next we consider the algorithm in discrete time $t$. The SVGD updates according to the following equation:

$$\boldsymbol{x}_i^{(t+1)} = \boldsymbol{x}_i^{(t)} + \frac{\epsilon}{N}\left(\sum_{j=1}^{N}\nabla_{\boldsymbol{x}_j^{(t)}}K\left(\boldsymbol{x}_i^{(t)}, \boldsymbol{x}_j^{(t)}\right) - \sum_{j=1}^{N}K\left(\boldsymbol{x}_i^{(t)}, \boldsymbol{x}_j^{(t)}\right)\nabla_{\boldsymbol{x}_j^{(t)}}V\left(\boldsymbol{x}_j^{(t)}\right)\right). \tag{21}$$

For simplicity, we only consider the case where both the target and initializers are centered, i.e., $\boldsymbol{b} = \boldsymbol{\mu}_0 = \boldsymbol{0}$ and show the convergence:

**Theorem 3.10** (Discrete-time convergence). *For a centered Gaussian target, suppose the particle system* (21) *with $K_1$ or $K_2$ is initialized by $\left(\boldsymbol{x}_i^{(0)}\right)_{i=1}^{N}$ such that $\boldsymbol{\mu}_0 = \boldsymbol{0}$ and $C_0 Q = Q C_0$. For $0 < \epsilon < 0.5$, we have $\boldsymbol{\mu}_t = \boldsymbol{0}$ and $\|C_t - Q\| \to 0$ as long as all the eigenvalues of $Q^{-1}C_0$ lie in the interval $(0, 1 + 1/\epsilon)$.*

*Furthermore, if we set $u_\epsilon$ to be the smaller root of the equation $f'_\epsilon(u) = 1 - \epsilon$ (it has 2 distinct roots) where $f_\epsilon(x) := (1 + \epsilon(1-x))^2 x$, then we have linear convergence, i.e.,*

$$\|C_t - Q\| \le (1-\epsilon)^t \|C_0 - Q\| \le e^{-\epsilon t}\|C_0 - Q\|$$

*as long as all the eigenvalues of $Q^{-1}C_0$ lie in the interval $[u_\epsilon, 1/3 + 1/(3\epsilon)]$.*

The above result illustrates that firstly the step sizes required are restricted by the largest eigenvalue of $Q^{-1}C_0$. In particular if $C_0 = I_d$ then we need smaller step size if the smallest eigenvalue of $Q$ is smaller, which corresponds to the $\beta$-log-smoothness condition of the target distribution. Secondly we can potentially have faster convergence over iteration given larger step sizes. We believe that the commutativity assumption in Theorem 3.10 can be relaxed and similar results can be obtained for general targets. Detailed examinations are left as future work.

## 4 Beyond Gaussian Targets

In this section we consider the Gaussian–SVGD with a general target and have the following dynamics.

**Theorem 4.1.** *Let $\rho^*$ be the density of the target distribution with the potential function $V(\boldsymbol{x})$ that satsifies Assumption C.1 and $\rho_0$ be the density of $\mathcal{N}(\boldsymbol{\mu}_0, \Sigma_0)$. The Gaussian–SVGD with $K_1$ produces a Gaussian density $\rho_t$ with mean $\boldsymbol{\mu}_t$ and covariance matrix $\Sigma_t$ given by*

$$\begin{cases} \dot{\boldsymbol{\mu}}_t = (I - \Gamma_t \Sigma_t)\boldsymbol{\mu}_t - (1 + \boldsymbol{\mu}_t^\top \boldsymbol{\mu}_t)\boldsymbol{m}_t \\ \dot{\Sigma}_t = 2\Sigma_t - \Sigma_t\left(\Sigma_t\Gamma_t + \boldsymbol{\mu}_t\boldsymbol{m}_t^\top\right) - \left(\Gamma_t\Sigma_t + \boldsymbol{m}_t\boldsymbol{\mu}_t^\top\right)\Sigma_t \end{cases}, \tag{22}$$

*where $\Gamma_t = \mathbb{E}_{\boldsymbol{x} \sim \rho_t}[\nabla^2 V(\boldsymbol{x})]$ and $\boldsymbol{m}_t = \mathbb{E}_{\boldsymbol{x} \sim \rho_t}[\nabla V(\boldsymbol{x})]$.*

*Furthermore, suppose that $\theta^*$ is the unique solution of the following optimization problem*

$$\min_{\theta=(\boldsymbol{\mu}, \Sigma)} \mathrm{KL}(\rho_\theta \parallel \rho^*), \text{ where } \rho_\theta \text{ is the Gaussian measure } \mathcal{N}(\boldsymbol{\mu}, \Sigma).$$

*Then we have $\rho_t \to \rho_{\theta^*} \sim \mathcal{N}(\boldsymbol{\mu}^*, \Sigma^*)$ as $t \to \infty$.*

In particular, if the target is strongly log-concave, it gives rise to the following linear convergence:

| **Algorithm 1** Density-based Gaussian–SVGD. | **Algorithm 2** Particle-based Gaussian–SVGD. |
|---|---|
| **for** $t$ in $0:T$ **do** 
 $\quad$ Draw $\big(\boldsymbol{x}_i^{(t)}\big)_{i=1}^N$ from $\mathcal{N}(\boldsymbol{\mu}_t, \Sigma_t)$ 
 $\quad$ Update $\widehat{m}_t$ and $\widehat{\Gamma}_t$ using (24) or (25) 
 $\quad \boldsymbol{\mu}_{t+1} \leftarrow \boldsymbol{\mu}_t + \epsilon_1\, F\big(\boldsymbol{\mu}_t, \Sigma_t, \widehat{m}_t, \widehat{\Gamma}_t\big)$ 
 $\quad M_{t+1} \leftarrow I + \epsilon_2\, G\big(\boldsymbol{\mu}_t, \Sigma_t, \widehat{m}_t, \widehat{\Gamma}_t\big)$ 
 $\quad \Sigma_{t+1} \leftarrow M_{t+1}\Sigma_t M_{t+1}^\top$ 
 **end for** | Draw $(\boldsymbol{x}_i^{(0)})_{i=1}^N$ from $\mathcal{N}(\boldsymbol{\mu}_0, \Sigma_0)$ 
 **for** $t$ in $0:T$ **do** 
 $\quad \boldsymbol{\mu}_t \leftarrow \frac{1}{N}\sum_{k=1}^N \boldsymbol{x}_k^{(t)}$ 
 $\quad \Sigma_t \leftarrow \frac{1}{N}\sum_{k=1}^N \boldsymbol{x}_k^{(t)}\boldsymbol{x}_k^{(t)\top} - \boldsymbol{\mu}_t\boldsymbol{\mu}_t^\top$ 
 $\quad$ Update $\widehat{m}_t$ and $\widehat{\Gamma}_t$ using (24) or (25) 
 $\quad$ Update $(\boldsymbol{x}_i^{(t+1)})_{i=1}^N$ using (26) 
 **end for** |

**Theorem 4.2.** *Assume that the target $\rho^*$ is $\alpha$-strongly log-concave and $\beta$-log-smooth, i.e., $\alpha I \preceq \nabla^2 V(\boldsymbol{x}) \preceq \beta I$. Then $\rho_t$ of Theorem 4.1 converges to $\rho_{\theta*}$ at the following rate*

$$\|\boldsymbol{\mu}_t - \boldsymbol{\mu}^*\| = \mathcal{O}(e^{-(\gamma-\epsilon)t}), \quad \|\Sigma_t - \Sigma^*\| = \mathcal{O}(e^{-(\gamma-\epsilon)t}), \quad \forall \epsilon > 0,$$

*where $\gamma/\alpha$ is the smallest eigenvalue of the matrix*

$$\begin{bmatrix} I_d \otimes \Sigma^* & \boldsymbol{\mu}^* \otimes (\Sigma^*)^{1/2} \\ \boldsymbol{\mu}^{*\top} \otimes (\Sigma^*)^{1/2} & (1 + \boldsymbol{\mu}^{*\top}\boldsymbol{\mu}^*)I_d \end{bmatrix} \text{ with a lower bound } \gamma > \frac{\alpha}{\beta(1 + \boldsymbol{\mu}^{*\top}\boldsymbol{\mu}^*) + 1}.$$

Typically the $\beta$-log-smoothness condition is not required for continuous-time analyses. However, it is required in the above statement, as our proof technique is based on comparing the decay of the energy function of the flow to that for WGF, following [13]. Relaxing this condition is interesting and we leave it as future work.

**Unifying Algorithms.** For general targets, we propose two unifying algorithm frameworks where we can choose any bilinear kernel (e.g., $K_1$–$K_4$) to solve GVI with SVGD. The first framework is density-based where we update $\boldsymbol{\mu}_t$ and $\Sigma_t$ according to the mean-field dynamics. It requires the closed-form of the ODE system

$$\begin{cases} \dot{\boldsymbol{\mu}}_t = F(\boldsymbol{\mu}_t, \Sigma_t, \boldsymbol{m}_t, \Gamma_t) \\ \dot{\Sigma}_t = \Sigma_t\, G(\boldsymbol{\mu}_t, \Sigma_t, \boldsymbol{m}_t, \Gamma_t) + G(\boldsymbol{\mu}_t, \Sigma_t, \boldsymbol{m}_t, \Gamma_t)^\top\, \Sigma_t \end{cases}, \tag{23}$$

where $\boldsymbol{m}_t = \mathbb{E}_{\boldsymbol{x}\sim\rho_t}[\nabla V(\boldsymbol{x})]$ and $\Gamma_t = \mathbb{E}_{\boldsymbol{x}\sim\rho_t}[\nabla^2 V(\boldsymbol{x})]$, and $F$ and $G$ are some closed-form functions. For example $F(\boldsymbol{\mu}_t, \Sigma_t, \boldsymbol{m}_t, \Gamma_t) = (I - \Gamma_t\Sigma_t)\boldsymbol{\mu}_t - (1 + \boldsymbol{\mu}_t^\top\boldsymbol{\mu}_t)\boldsymbol{m}_t$ and $G(\boldsymbol{\mu}_t, \Sigma_t, \boldsymbol{m}_t, \Gamma_t) = I - \Sigma_t\Gamma_t - \boldsymbol{\mu}_t\boldsymbol{m}_t^\top$ for $K_1$ as shown in (22). Note that $\boldsymbol{m}_t$ and $\Gamma_t$ can be estimated from samples using

$$\widehat{\boldsymbol{m}}_t = \frac{1}{N}\sum_{k=1}^N \nabla V(\boldsymbol{x}_k^{(t)}), \quad \widehat{\Gamma}_t = \frac{1}{N}\sum_{k=1}^N \nabla^2 V(\boldsymbol{x}_k^{(t)}), \tag{24}$$

or using the first-order estimator

$$\widehat{\Gamma}_t = \frac{1}{N}\sum_{k=1}^N \nabla V(\boldsymbol{x}_k^{(t)})(\boldsymbol{x}_k^{(t)} - \boldsymbol{\mu}_t)^\top \Sigma_t^{-1}. \tag{25}$$

The second framework is particle-based and does not need the closed-form ODE of the mean and covariance, making it more flexible than Algorithm 1. Here we intially draw $N$ particles and keep updating them over time using

$$\boldsymbol{x}_i^{(t+1)} = \boldsymbol{x}_i^{(t)} + \frac{\epsilon}{N}\left(\sum_{j=1}^N \nabla_{\boldsymbol{x}_j^{(t)}} K(\boldsymbol{x}_i^{(t)}, \boldsymbol{x}_j^{(t)}) - \sum_{j=1}^N K(\boldsymbol{x}_i^{(t)}, \boldsymbol{x}_j^{(t)})\widehat{\nabla V}(\boldsymbol{x}_j^{(t)})\right), \tag{26}$$

where $\widehat{\nabla V}$ is a (time-dependent) linear approximation of $\nabla V$ defined as $\widehat{\nabla V}(\boldsymbol{x}) = \widehat{\Gamma}_t(\boldsymbol{x} - \boldsymbol{\mu}_t) + \widehat{\boldsymbol{m}}_t$. Intuitively this is used instead of $\nabla V$ to ensure the Gaussianity of the particle system.

We now remark on the algorithms proposed in the literature that emerge as instances of the two proposed algorithm frameworks, thereby highlighting the unifying viewpoint offered by our analysis. The use of the kernel $K_1$ for SVGD in variational inference dates back to [54]. Algorithms 1 and 2 with the kernel $K_2$ correspond precisely to the GF and GPF algorithms in [27]. Moreover, if $K_3$ is chosen, Algorithm 1 reproduces the BWGD algorithm in [43] (with $N = 1$) and shares some similarity with the FB-GVI algorithm in [19]. Detailed discussions on these variants, connections and convergence properties are deferred to Appendix D.

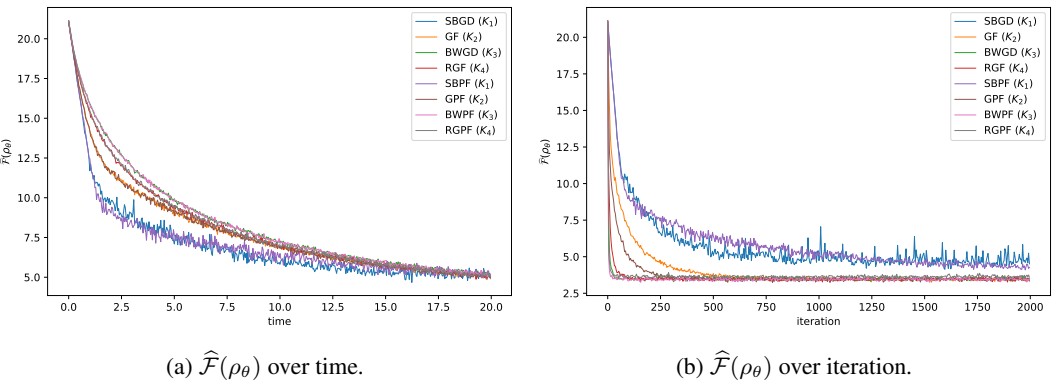

(a) $\widehat{\mathcal{F}}(\rho_\theta)$ over time.

(b) $\widehat{\mathcal{F}}(\rho_\theta)$ over iteration.

Figure 1: Convergence of Algorithms 1 and 2 with bilinear kernels in Bayesian logistic regression.

## 5 Simulations

In this section, we conduct simulations to compare Gaussian–SVGD dynamics with different kernels and the performance of the algorithms mentioned in the previous section. We consider three settings, **Bayesian logistic regression**, and **Gaussian** and **Gaussian mixture targets**. Here we present the results for Bayesian logistic regression as it involves a non-Gaussian but unimodal target and is one of the typical setups such that GVI is preferred in practice. For sake of space we leave the simulations for the other two settings along with further discussions to Appendix E.

**Bayesian Logistic Regression.** The following generative model is considered: Given a parameter $\boldsymbol{\xi} \in \mathbb{R}^d$, draw samples $\{(X_i, Y_i)\}_{i=1}^n \in (\mathbb{R}^d \times \{0, 1\})^n$ such that $X_i \overset{\text{i.i.d.}}{\sim} \mathcal{N}(\mathbf{0}, I_d)$ and $Y_i \mid X_i \sim \text{Bern}(\sigma(\langle \boldsymbol{\xi}, X_i \rangle))$ where $\sigma(\cdot)$ is the logistic function. Given the samples $\{(X_i, Y_i)\}_{i=1}^n$ and a uniform (improper) prior on $\boldsymbol{\xi}$, the potential function of the posterior $\rho^*$ on $\boldsymbol{\xi}$ is given by $V(\boldsymbol{\xi}) = \sum_{i=1}^n \left( \log(1 + \exp(\langle \boldsymbol{\xi}, X_i \rangle)) - Y_i \langle \boldsymbol{\xi}, X_i \rangle \right)$. We run both Algorithms 1 and 2 initialized at $\rho_0 = \mathcal{N}(\mathbf{0}, I_d)$ to find the $\rho_{\theta^*}$ that minimizes $\text{KL}(\rho_\theta \parallel \rho^*)$. In Figure 1, **SBGD** (Simple Bilinear Gradient Descent), **GF** (Gaussian Flow [27]), **BWGD** (Bures–Wasserstein Gradient Descent [43]), and **RGF** (Regularized Gaussian Flow) are density-based algorithms with the bilinear kernels $K_1$, $K_2$, $K_3$, and $K_4$ ($\nu = 0.5$) respectively; **SBPF** (Simple Bilinear Particle Flow), **GPF** (Gaussian Particle Flow [27]), **BWPF** (Bures–Wasserstein Particle Flow), and **RGPF** (Regularized Gaussian Particle Flow) are particle-based algorithms with the bilinear kernels $K_1$, $K_2$, $K_3$, and $K_4$ ($\nu = 0.5$) respectively. We use the sample estimator of $\mathcal{F}(\rho_\theta) = \int (\log \rho_\theta + V) \, \mathrm{d}\rho_\theta = \text{KL}(\rho_\theta \parallel \rho^*) + C$ ($C$ is some constant) to evaluate the learned parameters $\theta = (\boldsymbol{\mu}, \Sigma)$. For a fair comparison in Figure 1b, we draw 2000 particles for particle-based algorithms and run all algorithms for 2000 iterations so that a total of 2000 samples are drawn for the density-based algorithms. The largest safe step sizes are $0.02, 0.1, 2, 0.8, 0.02, 0.2, 4, 4$.

Figure 1 shows the decay of $\widehat{\mathcal{F}}(\rho_\theta)$ over time or iterations. For Figure 1a the same step size $0.01$ is specified for all algorithms while for Figure 1b we choose the largest safe step size for each algorithm. In other words, Figure 1a provides the continuous-time flow of the dynamical system in each algorithm while Figure 1b emphasizes more on the discrete-time algorithmic behaviors. In Figure 1a there are roughly four distinct curves indicating four different kernels. $K_1$ has the most rapid descent, followed by $K_2$, $K_4$, and $K_3$.

From Figure 1b we observe that BWPF and RGPF are the better choices for practical use. The difference in the largest step sizes shows that in terms of stability $K_3$ is the best, $K_4$ is almost as stable, but $K_2$ and $K_1$ are much worse. We also observe that particle-based algorithms are consistently more stable than density-based counterparts (which are essentially stochastic gradient based). The superiority of particle-based algorithms are even more evident in the Gaussian mixture experiment where the target is multi-modal. We further remark that another recently proposed density-based algorithm, the FB-GVI [19] shows comparable performance to BWPF and RGPF with large step sizes. We conduct a comparison of these three algorithms in Appendix E but do not include it here for clarity. It would also be really interesting to study the particle-based analogue of FB-GVI as future works.

## Acknowledgements

We thank Lester Mackey and Jiaxin Shi for several clarifications about their work, [72], and for helpful discussions regarding the larger literature on SVGD. Promit Ghosal was supported in part by NSF grant DMS-2153661. Krishnakumar Balasubramanian was supported in part by NSF grant DMS-2053918. Natesh S. Pillai was supported by ONR grant N00014-21-1-2664.

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

# A    Further Discussion

**Other Related Works.** There exist several other approaches for minimizing KL-divergence over the Wasserstein space (or over appropriately restricted subsets). For example, normalizing flows [68, 37, 11], the blob method [10], variants of gradient descent in Wasserstein spaces [69, 85, 5], natural-gradient based variational inference techniques [48], mean-field methods [88], neural networks based approaches [2] and MCMC methods. This is certainly not a comprehensive list and summarizing the large literature on this topic is impossible in the limited space available. However, we emphasize that a majority of the above methods are nonparametric and do not come with strong theoretical guarantees. Our main focus in this work is on theoretically understanding (Gaussian)-SVGD and its connection to GVI.

**Beyond Gaussian VI.** While we present a comprehensive study of Gaussian–SVGD, it is currently unclear how such relation between bilinear kernels and Gaussian family could be generalized. For example, it would be interesting to ask if there is a class (submanifold) $\mathcal{C}$ of probability densities such that for any initialization-target pair $\rho_0, \rho^* \in \mathcal{C}$, the dynamics of SVGD with a radius-based kernel function (RBF-SVGD) would always remain in $\mathcal{C}$. If such $\mathcal{C}$ is identified, we could similarly carry out the convergence analysis of RBF-SVGD and perform variational inference with respect to the class $\mathcal{C}$. It remains an open question whether the uniform in time propagation of chaos as shown in Theorem 3.7 would hold for general kernels. Moreover, there has been increasing interest on variational inference with respect to other special density classes. Remarkably [43] considered Gaussian mixtures in additional to GVI, and [62] analyzed the family of elliptical distributions, which has applications for performing variational inference under heavy-tails [21].

**SVGD in High Dimensions.** It is well-known in literature [89, 4] that RBF-SVGD suffers from severe particle degeneracy in high dimensions and cannot guarantee good covariance estimation. However, such issue has not been observed for Gaussian–SVGD in our simulations. It would be interesting future works to study whether this undesired phenomenon is tied to specific choices of kernels and to carry out theoretical analysis of (Gaussian)–SVGD in high dimensions.

# B    Details on Gaussian–Stein Metrics

For any region $\Omega \subseteq \mathbb{R}^d$, denote the set of probability densities on $\Omega$ by

$$\mathcal{P}(\Omega) := \left\{ \rho \in \mathcal{F}(\Omega) : \int_\Omega \rho \, \mathrm{d}\boldsymbol{x} = 1, \rho \geq 0 \right\},$$

where $\mathcal{F}(\Omega)$ is the set of $\mathcal{C}^\infty$-smooth functions on $\Omega$. As studied in [42], $\mathcal{P}(\Omega)$ forms a Fréchet manifold called the density manifold. For any "point" $\rho \in \mathcal{P}(\Omega)$, the tangent space is given by

$$T_\rho \mathcal{P}(\Omega) = \left\{ \sigma \in \mathcal{F}(\Omega) : \int \sigma \, \mathrm{d}\boldsymbol{x} = 0 \right\}.$$

And the cotangent space at $\rho$, $T_\rho^* \mathcal{P}(\Omega)$ consists of equivalent classes of $\mathcal{F}(\Omega)$, each containing functions that differ by a constant. A Riemannian metric (tensor) assigns to each $\rho \in \mathcal{P}(\Omega)$ a positive definite inner product $g_\rho : T_\rho \mathcal{P}(\Omega) \times T_\rho \mathcal{P}(\Omega) \to \mathbb{R}$. If we define the pairing between $T_\rho^* \mathcal{P}(\Omega)$ and $T_\rho \mathcal{P}(\Omega)$ by

$$\langle \Phi, \sigma \rangle := \int \Phi \cdot \sigma \, \mathrm{d}\boldsymbol{x},$$

where $\Phi \in T_\rho^* \mathcal{P}(\Omega)$ and $\sigma \in T_\rho \mathcal{P}(\Omega)$. Any Riemannian metric uniquely corresponds to an isomorphism (called the **canonical isomorphism**) between the tangent and cotangent bundles [66], i.e., we have $G_\rho : T_\rho \mathcal{P}(\Omega) \to T_\rho^* \mathcal{P}(\Omega)$ through

$$g_\rho(\sigma_1, \sigma_2) = \langle G_\rho \sigma_1, \sigma_2 \rangle = \langle G_\rho \sigma_2, \sigma_1 \rangle, \quad \sigma_1, \sigma_2 \in T_\rho \mathcal{P}(\Omega).$$

**Definition B.1** (Wasserstein metric)**.** *The Wasserstein metric is induced by the following canonical isomorphism* $G_\rho^{\mathrm{Wass}} : T_\rho \mathcal{P}(\Omega) \to T_\rho^* \mathcal{P}(\Omega)$ *such that*

$$(G_\rho^{\mathrm{Wass}})^{-1} \Phi = -\nabla \cdot (\rho \nabla \Phi), \quad \Phi \in T_\rho^* \mathcal{P}(\Omega).$$

Note that the Wasserstein metric we define here corresponds exactly to the Wasserstein-2 distance studied in the literature of optimal transport [83]. For details of such connection, please refer to [36]. Also we need to clarify that the concepts we introduce follow from the convention in Riemannian geometry, where the manifold strictly speaking is required to be finite-dimensional. For

a mathematically formal formulation of infinite-dimensional calculus on the density manifold, please refer to [42, 41, 65, 57].

The Wasserstein gradient flow can be seen as the natural gradient flow on the density manifold with respect to the Wasserstein metric. In specific, consider any energy functional $E(\rho)$, e.g., the KL divergence from $\rho$ to some fixed target $\rho^*$. The Wasserstein gradient flow for $E(\rho)$ is provided by

$$\dot{\rho}_t = -(G^{\text{Wass}}_{\rho_t})^{-1} \frac{\delta E}{\delta \rho_t} = \nabla \cdot \left( \rho_t \nabla \frac{\delta E}{\delta \rho_t} \right), \tag{27}$$

where $\frac{\delta E}{\delta \rho_t}$ is the variational derivative of the functional $E$ with respect to $\rho_t$. If $E(\rho)$ is the KL divergence mentioned above, (27) gives the linear Fokker-Planck equation [33]

$$\dot{\rho}_t = \nabla \cdot (\nabla \rho_t + \rho_t \nabla V),$$

where $V(\boldsymbol{x}) = -\log \rho^*(\boldsymbol{x})$ is called the potential function. If $E(\rho)$ is the Wasserstein metric between $\rho$ and $\rho^*$, (1) gives the geodesic flow on the density manifold. If $E(\rho)$ is the maximum mean discrepancy (MMD) or kernelized Stein discrepancy (KSD) between $\rho$ and $\rho^*$, it leads to the MMD descent and KSD descent algorithms [3, 38].

Interestingly, the mean field flow of SVGD can also be seen as the gradient flows of the KL divergence on the density manifold but with respect to the Stein metric, where we perform kernelization in the cotangent space before taking the divergence [51, 22].

**Definition B.2** (Stein metric). *The Stein metric is induced by the following canonical isomorphism $G^{\text{Wass}}_\rho : T_\rho \mathcal{P}(\Omega) \to T^*_\rho \mathcal{P}(\Omega)$ such that*

$$(G^{\text{Stein}}_\rho)^{-1} \Phi = -\nabla \cdot \left( \rho(\cdot) \int K(\cdot, \boldsymbol{y}) \rho(\boldsymbol{y}) \nabla \Phi(\boldsymbol{y}) \, \mathrm{d}\boldsymbol{y} \right), \quad \Phi \in T^*_\rho \mathcal{P}(\Omega).$$

In particular, the mean field PDE of the SVGD algorithm can be written as

$$\dot{\rho}_t(\boldsymbol{x}) = -(G^{\text{Stein}}_\rho)^{-1} \frac{\delta \, \mathrm{KL}(\rho_t \parallel \rho^*)}{\delta \rho_t}(\boldsymbol{x}) \tag{28}$$

$$= \nabla \cdot \left( \rho_t(\boldsymbol{x}) \int K(\boldsymbol{x}, \boldsymbol{y}) \rho_t(\boldsymbol{y}) \nabla \frac{\delta \, \mathrm{KL}(\rho_t \parallel \rho^*)}{\delta \rho_t}(\boldsymbol{y}) \, \mathrm{d}\boldsymbol{y} \right)$$

$$= \nabla \cdot \left( \rho_t(\boldsymbol{x}) \int K(\boldsymbol{x}, \boldsymbol{y}) \rho_t(\boldsymbol{y}) \, \nabla \big( \log \rho_t(\boldsymbol{y}) + V + 1 \big) \, \mathrm{d}\boldsymbol{y} \right)$$

$$= \nabla \cdot \left( \rho_t(\boldsymbol{x}) \int K(\boldsymbol{x}, \boldsymbol{y}) \big( \nabla \rho_t(\boldsymbol{y}) + \rho_t(\boldsymbol{y}) \nabla V(\boldsymbol{y}) \big) \, \mathrm{d}\boldsymbol{y} \right),$$

where $V(\boldsymbol{x}) = -\log \rho^*(\boldsymbol{x})$.

We set $\Omega = \mathbb{R}^d$ and consider the multivariate Gaussian densities $\mathcal{N}(\boldsymbol{\mu}, \Sigma)$ in $\mathbb{R}^d$ :

$$\rho(\boldsymbol{x}, \theta) = \frac{1}{\sqrt{\det(2\pi\Sigma)}} \exp \left( -\frac{1}{2} (\boldsymbol{x} - \boldsymbol{\mu})^\top \Sigma^{-1} (\boldsymbol{x} - \boldsymbol{\mu}) \right),$$

where $\theta := (\boldsymbol{\mu}, \Sigma) \in \Theta$ and $\Theta := \mathbb{R}^d \times \mathrm{Sym}^+(d, \mathbb{R})$. Here $\mathrm{Sym}^+(d, \mathbb{R})$ is the set of (symmetric) positive definite $d \times d$ matrices, which is an open subset of the $d \times d$ symmetric matrix space $\mathrm{Sym}(d, \mathbb{R})$. In this way, $\Theta$ can be identified as a Riemannian submanifold of the density manifold $\mathcal{P}(\mathbb{R}^d)$ with the induced Riemannian structure.

We first look into the Stein metric with the bilinear kernel $K_1(\boldsymbol{x}, \boldsymbol{y}) = \boldsymbol{x}^\top \boldsymbol{y} + 1$, and derive the closed form of the induced metric tensor on $\Theta$, which plays an essential role in characterizing the SVGD dynamics.

**Theorem B.3** (Gaussian–Stein metric with the simple bilinear kernel). *Given $\theta = (\boldsymbol{\mu}, \Sigma) \in \Theta$, let $g_\theta(\cdot, \cdot)$ denote the Gaussian–Stein metric tensor for the multivariate Gaussian model with the bilinear kernel $K_1(\boldsymbol{x}, \boldsymbol{y}) = \boldsymbol{x}^\top \boldsymbol{y} + 1$, and $G_\theta$ be the corresponding canonical isomorphism from $T_\theta \Theta \simeq \mathbb{R}^d \times \mathrm{Sym}(d, \mathbb{R})$ to $T^*_\theta \Theta \simeq \mathbb{R}^d \times \mathrm{Sym}(d, \mathbb{R})$.*

*For any $\boldsymbol{\nu} \in \mathbb{R}^n$, and $S \in \mathrm{Sym}(d, \mathbb{R})$, the inverse map $G^{-1}_\theta$ is given by the following automorphism on $\mathbb{R}^d \times \mathrm{Sym}(d, \mathbb{R})$:*

$$G^{-1}_\theta(\boldsymbol{\nu}, S) = \big( 2S\Sigma\boldsymbol{\mu} + (1 + \boldsymbol{\mu}^\top \boldsymbol{\mu})\boldsymbol{\nu}, \; \big( \Sigma(2\Sigma S + \boldsymbol{\mu}\boldsymbol{\nu}^\top) + (2S\Sigma + \boldsymbol{\nu}\boldsymbol{\mu}^\top)\Sigma \big) \big). \tag{29}$$

*And for any $\xi, \eta \in T_\theta \Theta$ the Stein metric tensor can be written as*

$$g_\theta(\xi, \eta) = \operatorname{tr}(S_1 S_2 \Sigma^2) + (\boldsymbol{b}_1^\top S_2 + \boldsymbol{b}_2^\top S_1)\Sigma\boldsymbol{\mu} + (1 + \boldsymbol{\mu}^\top \boldsymbol{\mu})\boldsymbol{b}_1^\top \boldsymbol{b}_2. \tag{30}$$

*Here $\xi = \left(\widetilde{\boldsymbol{\mu}}_1, \widetilde{\Sigma}_1\right)$ and $\eta = \left(\widetilde{\boldsymbol{\mu}}_2, \widetilde{\Sigma}_2\right)$, in which $\widetilde{\boldsymbol{\mu}}_1, \widetilde{\boldsymbol{\mu}}_2 \in \mathbb{R}^d, \widetilde{\Sigma}_1, \widetilde{\Sigma}_2 \in \operatorname{Sym}(d, \mathbb{R})$. And $\boldsymbol{b}_i, S_i$'s $(i = 1, 2)$ are defined as*

$$\left(\boldsymbol{b}_i, \frac{1}{2}S_i\right) = G_\theta(\widetilde{\boldsymbol{\mu}}_i, \widetilde{\Sigma}_i).$$

Then for the Gaussian–Stein metric with kernel $K_2(\boldsymbol{x}, \boldsymbol{y}) = (\boldsymbol{x} - \boldsymbol{\mu})^\top (\boldsymbol{y} - \boldsymbol{\mu}) + 1$ we have the following result:

**Theorem B.4** (Gaussian–Stein metric with the affine-invariant bilinear kernel). *Given $\theta = (\boldsymbol{\mu}, \Sigma) \in \Theta$, let $g_\theta(\cdot, \cdot)$ denote the affine-invariant Gaussian–Stein metric tensor for the multivariate Gaussian model with the affine-invariant bilinear kernel $K_2$, and $G_\theta$ be the corresponding canonical isomorphism from $T_\theta \Theta \simeq \mathbb{R}^d \times \operatorname{Sym}(d, \mathbb{R})$ to $T_\theta^* \Theta \simeq \mathbb{R}^d \times \operatorname{Sym}(d, \mathbb{R})$.*

*For any $\boldsymbol{\nu} \in \mathbb{R}^n$, and $S \in \operatorname{Sym}(d, \mathbb{R})$, the inverse map $G_\theta^{-1}$ is given by the following automorphism on $\mathbb{R}^d \times \operatorname{Sym}(d, \mathbb{R})$:*

$$G_\theta^{-1}(\boldsymbol{\nu}, S) = \left(\boldsymbol{\nu}, \ 2\left(\Sigma^2 S + S\Sigma^2\right)\right). \tag{31}$$

*And for any $\xi, \eta \in T_\theta \Theta$ the Stein metric tensor can be written as*

$$g_\theta(\xi, \eta) = \operatorname{tr}(S_1 S_2 \Sigma^2) + \widetilde{\boldsymbol{\mu}}_1^\top \widetilde{\boldsymbol{\mu}}_2. \tag{32}$$

*Here $\xi = \left(\widetilde{\boldsymbol{\mu}}_1, \widetilde{\Sigma}_1\right)$ and $\eta = \left(\widetilde{\boldsymbol{\mu}}_2, \widetilde{\Sigma}_2\right)$, in which $\widetilde{\boldsymbol{\mu}}_1, \widetilde{\boldsymbol{\mu}}_2 \in \mathbb{R}^d, \widetilde{\Sigma}_1, \widetilde{\Sigma}_2 \in \operatorname{Sym}(d, \mathbb{R})$. And $S_i$'s $(i = 1, 2)$ are defined as the symmetric solution of*

$$\widetilde{\Sigma}_i = \Sigma^2 S_i + S_i \Sigma^2.$$

Now if we further restrict the Gaussian family to have zero mean, it induces the submanifold $\Theta_0 = \operatorname{Sym}^+(d, \mathbb{R})$. We have the following result for $K_1$ or $K_2$ as a direct corollary of Theorem B.3 or Theorem B.4.

**Corollary B.5.** *Given $\Sigma \in \Theta_0$, let $g_\Sigma(\cdot, \cdot)$ denote the Gaussian–Stein metric tensor for the centered Gaussian family with the bilinear kernel $K(\boldsymbol{x}, \boldsymbol{y}) = \boldsymbol{x}^\top \boldsymbol{y} + 1$, and $G_\Sigma$ be the corresponding canonical isomorphism from $T_\Sigma \Theta_0 \simeq \operatorname{Sym}(d, \mathbb{R})$ to $T_\Sigma^* \Theta_0 \simeq \operatorname{Sym}(d, \mathbb{R})$.*

*For any $S \in \operatorname{Sym}(d, \mathbb{R})$, the inverse map $G_\Sigma^{-1}$ is given by the following automorphism on $\operatorname{Sym}(d, \mathbb{R})$:*

$$G_\Sigma^{-1}(S) = 2(\Sigma^2 S + S\Sigma^2). \tag{33}$$

*And for any $\widetilde{\Sigma}_1, \widetilde{\Sigma}_2 \in T_\Sigma \Theta_0$ the Stein metric tensor can be written as*

$$g_\Sigma(\widetilde{\Sigma}_1, \widetilde{\Sigma}_2) = \operatorname{tr}(S_1 S_2 \Sigma^2), \tag{34}$$

*where for $i = 1, 2$, $S_i$ is the unique solution in $\operatorname{Sym}(d, \mathbb{R})$ that satisfies the Lyapunov equation*

$$\widetilde{\Sigma}_i = \Sigma^2 S_i + S_i \Sigma^2.$$

Next we consider the Bures–Wasserstein metric for Gaussian families, which is defined as the Wasserstein metric restricted to the Gaussian family. It has the following elegant expressions from [77, 60]:

**Theorem B.6** (Bures–Wasserstein metric). *Given $\theta = (\boldsymbol{\mu}, \Sigma) \in \Theta$, let $g_\theta(\cdot, \cdot)$ denote the Bures–Wasserstein metric tensor for the multivariate Gaussian model (or equivalently Gaussian–Stein metric with the kernel $K_3$), and $G_\theta$ be the corresponding isomorphism from $T_\theta \Theta \simeq \mathbb{R}^d \times \operatorname{Sym}(d, \mathbb{R})$ to $T_\theta^* \Theta \simeq \mathbb{R}^d \times \operatorname{Sym}(d, \mathbb{R})$. For any $\boldsymbol{\nu} \in \mathbb{R}^n$, and $S \in \operatorname{Sym}(d, \mathbb{R})$, the inverse map $G_\theta^{-1}$ is given by the following automorphism on $\mathbb{R}^d \times \operatorname{Sym}(d, \mathbb{R})$:*

$$(G_\theta^{\text{Wass}})^{-1}(\boldsymbol{\nu}, S) = \left(\boldsymbol{\nu}, \ 2(\Sigma S + S\Sigma)\right). \tag{35}$$

*And for any $\xi, \eta \in T_\theta \Theta$ the Bures–Wasserstein metric tensor can be written as*

$$g_\theta(\xi, \eta) = \operatorname{tr}(S_1 S_2 \Sigma) + \widetilde{\boldsymbol{\mu}}_1^\top \widetilde{\boldsymbol{\mu}}_2. \tag{36}$$

*Here $\xi = \left( \widetilde{\boldsymbol{\mu}}_1, \widetilde{\Sigma}_1 \right)$ and $\eta = \left( \widetilde{\boldsymbol{\mu}}_2, \widetilde{\Sigma}_2 \right)$, in which $\widetilde{\boldsymbol{\mu}}_1, \widetilde{\boldsymbol{\mu}}_2 \in \mathbb{R}^d, \widetilde{\Sigma}_1, \widetilde{\Sigma}_2 \in \mathrm{Sym}(d, \mathbb{R})$. And $S_i$'s $(i = 1, 2)$ are defined as the symmetric solution of*

$$\widetilde{\Sigma}_i = \Sigma S_i + S_i \Sigma.$$

*Notably this metric coincides with Gaussian–Stein metric with the kernel $K_3(\boldsymbol{x}, \boldsymbol{y}) = (\boldsymbol{x} - \boldsymbol{\mu})^\top \Sigma^{-1} (\boldsymbol{y} - \boldsymbol{\mu}) + 1$.*

Note that the only difference between (32) and (36) is the power of $\Sigma$.

**Theorem B.7** (Regularized Gaussian–Stein metric with the affine-invariant bilinear kernel)**.** *Given $\theta = (\boldsymbol{\mu}, \Sigma) \in \Theta$, let $g_\theta(\cdot, \cdot)$ denote the affine-invariant regularized Gaussian–Stein metric tensor for Gaussian families, and $G_\theta$ be the corresponding isomorphism from $T_\theta \Theta \simeq \mathbb{R}^d \times \mathrm{Sym}(d, \mathbb{R})$ to $T_\theta^* \Theta \simeq \mathbb{R}^d \, \mathrm{Sym}(d, \mathbb{R})$. For any $\boldsymbol{\nu} \in \mathbb{R}^d$ and $S \in \mathrm{Sym}(d, \mathbb{R})$, the inverse map $G_\theta^{-1}$ is given by the following automorphism on $\mathbb{R}^d \times \mathrm{Sym}(d, \mathbb{R})$:*

$$(G_\theta^{\mathrm{RS}})^{-1}(\boldsymbol{\nu}, S) = \left( \boldsymbol{\nu}, 2 \left( ((1 - \nu)\Sigma + \nu I)^{-1} \Sigma^2 S + S((1 - \nu)\Sigma + \nu I)^{-1} \Sigma^2) \right) \right). \qquad (37)$$

*Notably this metric coincides with Gaussian–Stein metric with the kernel $K_4(\boldsymbol{x}, \boldsymbol{y}) = (\boldsymbol{x} - \boldsymbol{\mu})^\top ((1 - \nu)\Sigma + \nu I)^{-1} (\boldsymbol{y} - \boldsymbol{\mu}) + 1$.*

## C  Properties of SVGD Solutions with the Simple Bilinear Kernel

[58] showed a few nice properties of the mean field PDE (2) for SVGD with radius-based kernels that can be written as $K(\boldsymbol{x} - \boldsymbol{y})$ which is symmetric and positive definite, meaning that

$$\sum_{i=1}^{m} \sum_{j=1}^{m} K(\boldsymbol{x}_i - \boldsymbol{x}_j)\xi_i \xi_j \geq 0, \quad \forall \boldsymbol{x}_i \in \mathbb{R}^d, \ \xi_i \in \mathbb{R}, \ m \in \mathbb{N}.$$

Although their results covered a large class of kernels commonly used in practice e.g., Gaussian kernels, they do not apply to the bilinear kernel $K(\boldsymbol{x}, \boldsymbol{y}) = \boldsymbol{x}^\top \boldsymbol{y} + 1$. In this subsection we establish similar results for the bilinear kernel. In fact, some of their results for radius-based kernels still hold here while some do not.

Before showing the properties of the mean field PDE, we need the following assumption on the potential function $V$.

**Assumption C.1.** *The potential function $V : \mathbb{R}^d \to \mathbb{R}$ satisfies the conditions below:*

   1. *$V \in \mathcal{C}^\infty(\mathbb{R}^d)$, $V \geq 0$, and $V(\boldsymbol{x}) \to \infty$ as $\|\boldsymbol{x}\| \to \infty$.*

   2. *For any $\alpha, \beta > 0$, there exists a constant $C_{\alpha, \beta} > 0$ such that if $\|\boldsymbol{y}\| \leq \alpha \|\boldsymbol{x}\| + \beta$, then the following inequality always holds that*

   $$(1 + \|\boldsymbol{x}\|)\|\nabla V(\boldsymbol{y})\| + (1 + \|\boldsymbol{x}\|)^2 \|\nabla^2 V(\boldsymbol{y})\| \leq C_{\alpha, \beta}(1 + V(\boldsymbol{x})).$$

To make things precise although all vector norms in $\mathbb{R}^d$ and all matrix norms are equivalent, we always choose the Euclidean norm for vectors and the spectral norm for matrices unless otherwise specified. Note that our assumption here is a little different from Assumption 2.1 in [58]. We do not require their second formula but our second piece of assumption is slightly stricter than their third one. It is straightforward to check that any positive definite quadratic form satisfies all the assumptions above, corresponding to the case where the target is a non-degenerate Gaussian distribution.

We use $\mathcal{P}_V(\mathbb{R}^d)$ and $\mathcal{P}^p(\mathbb{R}^d)$ $(p = 1, 2, \cdots)$ to denote the set of probability measure $\mu$ on $\mathbb{R}^d$ satisfying

$$\|\mu\|_{\mathcal{P}_V} := \int_{\mathbb{R}^d} (1 + V(\boldsymbol{x})) \, \mathrm{d}\mu(\boldsymbol{x}) < \infty \qquad (38)$$

and

$$\|\mu\|_{\mathcal{P}^p} := \int_{\mathbb{R}^d} \|\boldsymbol{x}\|^p \, \mathrm{d}\mu(\boldsymbol{x}) < \infty \qquad (39)$$

respectively.

**Theorem C.2** (Well-posedness and regularity of the mean field solution). *Let $V$ satisfy Assumption C.1. For any $\nu \in \mathcal{P}_V(\mathbb{R}^d)$, there is a unique $\rho_t \in \mathcal{C}\left([0,\infty), \mathcal{P}_V(\mathbb{R}^d)\right)$ which is a weak solution to (2) with initial condition $\rho_0 = \nu$. Moreover, there exists $C_1 > 0$ depending on $V$ such that*

$$\|\rho_t\|_{\mathcal{P}_V} \leq e^{C_1 t} \|\nu\|_{\mathcal{P}_V} \quad t \geq 0. \tag{40}$$

*If $\nu \in \mathcal{P}^p(\mathbb{R}^d) \cap \mathcal{P}_V(\mathbb{R}^d)$, then for any $t \in [0,\infty)$, we have that $\rho_t \in \mathcal{P}^p(\mathbb{R}^d) \cap \mathcal{P}_V(\mathbb{R}^d)$ and there exists $C_2 > 0$ depending on $V$ such that*

$$\|\rho_t\|_{\mathcal{P}^p} \leq e^{C_2 t} \|\nu\|_{\mathcal{P}^p} \quad t \geq 0. \tag{41}$$

*If $\nu$ has a density $\rho_0(\boldsymbol{x}) \geq 0$, then $\rho_t$ also has a density. Furthermore, if $\rho_0 \in \mathcal{H}^k(\mathbb{R}^d)$ for some $k$, then we have $\rho_t \in \mathcal{H}^k(\mathbb{R}^d)$. Here*

$$\mathcal{H}^k(\mathbb{R}^d) = \mathcal{W}^{k,2}(\mathbb{R}^d) := \left\{ u \in \mathcal{L}^p(\mathbb{R}^d) : D^\alpha u \in \mathcal{L}^p(\mathbb{R}^d) \, \forall |\alpha| \leq k \right\}, \quad k \geq 1$$

*denotes the Sobolev (Hilbert) space of order $k$.*

**Theorem C.3** (Well-posedness of the finite-particle solution). *Let $V$ satisfy Assumption C.1. Then for any initial condition $X_0 = \left(\boldsymbol{x}_1^{(0)}, \cdots, \boldsymbol{x}_N^{(0)}\right)^\top \in \mathbb{R}^{dN}$, the system (13) has a unique solution*

$$X_t = \left(\boldsymbol{x}_1^{(t)}, \cdots, \boldsymbol{x}_N^{(t)}\right)^\top \in \mathcal{C}^1\left([0,\infty), \mathbb{R}^{dN}\right),$$

*and the measure $\mu_t^N = \frac{1}{N} \sum_{i=1}^N \delta_{\boldsymbol{x}_i^{(t)}}$ is a weak solution to the PDE (2).*

Finally, we show that if two initial probability measures are close to each other, the solutions of (2) up to time $T$ are also close. We need to further impose the following assumption on $V$.

**Assumption C.4.** *There exists a constant $C_V > 0$ and some index $q > 1$ such that*

$$\|\nabla V(\boldsymbol{x})\|^q \leq C_V(1 + V(\boldsymbol{x})) \quad \text{for every } \boldsymbol{x} \in \mathbb{R}^d \tag{42}$$

*and that*

$$\sup_{\theta \in [0,1]} \|\nabla^2 V(\theta \boldsymbol{x} + (1-\theta)\boldsymbol{y})\|^q \leq C_V \left(1 + \frac{V(\boldsymbol{x}) + V(\boldsymbol{y})}{(\|\boldsymbol{x}\| + \|\boldsymbol{y}\|)^q}\right). \tag{43}$$

Remarkably a Gaussian distribution satisfies both Assumptions C.1 and C.4. Secondly an important observation is that (42) implies that there is $C_0$ such that

$$V(\boldsymbol{x}) \leq C_0(1 + \|\boldsymbol{x}\|^{q'}) \quad \forall \boldsymbol{x} \in \mathbb{R}^d \tag{44}$$

where $q' = q/(q-1)$. Indeed, we note that

$$\partial_t \left(1 + V\left(t\frac{\boldsymbol{x}}{\|\boldsymbol{x}\|}\right)\right)^{\frac{q-1}{q}} = \frac{q-1}{q} \cdot \frac{\frac{\boldsymbol{x}}{\|\boldsymbol{x}\|} \cdot \nabla V\left(t\frac{\boldsymbol{x}}{\|\boldsymbol{x}\|}\right)}{\left(1 + V\left(t\frac{\boldsymbol{x}}{\|\boldsymbol{x}\|}\right)\right)^{1/q}} \leq \left(1 - \frac{1}{q}\right) C_V^{1/q}.$$

Integrating from $t = 0$ to $t = \|\boldsymbol{x}\|$, we get (44), which shows that $\mathcal{P}^p \subset \mathcal{P}_V$ for any $p \geq q' = q/(q-1)$.

**Theorem C.5.** *Let $V$ satisfy Assumptions C.1 and C.4. Let $R > 0$. Assume that $\nu_1, \nu_2$ are two initial probability measures in $\mathcal{P}^p(\mathbb{R}^d)$ satisfying $\|\nu_i\|_{\mathcal{P}^p} \leq R$ ($i = 1, 2$). Let $\mu_{1,t}$ and $\mu_{2,t}$ be the associated weak solutions to (2). Then given any $T > 0$, there exists a constant $C_T > 0$ depending on $V, R$ and $T$ such that*

$$\sup_{t \in [0,T]} \mathcal{W}_p(\mu_{1,t}, \mu_{2,t}) \leq C_T \mathcal{W}_p(\nu_1, \nu_2).$$

**Remark.**  1. *Theorem C.5 implies the convergence of empirical measure to the mean field limit at time $t \in [0,T]$. In fact, if we set $\nu_{1,n}$ to be an empirical measure, which converges to $\nu_2$ as $n$ grows to infinity, then $\mu_{1,n,t}$, the solution of (2) at time $t$ with initializaiton $\nu_{1,n}$, will also converge to $\mu_{2,t}$ for any $t \in [0,T]$.*

   2. *In general there is no guarantee that the density $\rho_t$ will converge to the target density $\rho^*$ in (2) with the bilinear kernel. Some counterexamples will be elaborated in the remarks following Theorem 3.6.*

   3. *We show in Theorem 3.1 that for Gaussian families $\rho_t$ always converges to the target density $\rho^*$ as $t \to \infty$ and the convergence rate is linear in KL divergence. We also establish a uniform in time convergence result (Theorem 3.7) for the empirical measure in this case.*

# D Details of the Gaussian–SVGD Algorithms

**Different Ways of Estimating $\Gamma_t$.** The first-order estimator of $\Gamma_t$ arises from

$$\Gamma_t = \mathbb{E}_{\rho_\theta}[\nabla^2 V(\boldsymbol{x})] = \int \rho_\theta(\boldsymbol{x})\nabla^2 V(\boldsymbol{x})\,\mathrm{d}\boldsymbol{x} = -\int \nabla\rho_\theta(\boldsymbol{x})\nabla V(\boldsymbol{x})^\top\,\mathrm{d}\boldsymbol{x}$$

$$= \int \rho_\theta(\boldsymbol{x})\Sigma^{-1}(\boldsymbol{x}-\boldsymbol{\mu})\nabla V(\boldsymbol{x})^\top\,\mathrm{d}\boldsymbol{x} = \Sigma^{-1}\mathbb{E}_{\rho_\theta}[(\boldsymbol{x}-\boldsymbol{\mu})\nabla V(\boldsymbol{x})^\top].$$

Since $\Gamma_t$ is symmetric we also have

$$\Gamma_t = \mathbb{E}_{\rho_\theta}[\nabla V(\boldsymbol{x})(\boldsymbol{x}-\boldsymbol{\mu})^\top]\Sigma^{-1}.$$

Note that using the first-order estimator also comes at a cost as the inverse of $\Sigma$ is needed. However, for density-based Gaussian–SVGD this $\Sigma^{-1}$ might cancel with $\Sigma$ in computing.

**Previous Algorithms Under the Proposed Frameworks.** The use of $K_1$ for SVGD in variational inference dates back to [54]. Our Algorithm 2 is slightly different from [54] in the sense that $\nabla V$ is replaced by a linear function to ensure Gaussianity. Moreover, Algorithms 1 and 2 with the kernel $K_2$ correspond precisely to the GF and GPF algorithms in [27]. If $K_3$ (Bures–Wasserstein metric) is chosen, Algorithm 1 reproduces the BW-SGD algorithm in [43] (with $N = 1$). [19] also uses $K_3$ (Bures–Wasserstein metric) but their energy function for gradient flow is different from others. Instead of directly performing the gradient descent to minimize KL divergence, they separate the KL divergence into two parts and perform the proximal gradient descent.

**Variants of Gradient Descent in Density-Based Gaussian–SVGD.** For the density-based SVGD, we draw new samples at each step. It is interesting to study the behavioral difference between drawing one sample (stochastic) and efficiently many samples (almost deterministic). For example, in [43] only one sample is drawn at each time step and they study the stochastic properties arising from this design. [19] considers both settings. In general, they do not differ much in convergence rates but there could be huge gaps in the constants of the bounds and will actually impact practical performance. Another choice is to only draw $N$ samples at time 0 and we use a linear transformation of the same $N$ points to serve as new samples at time $t$, which becomes similar to the particle-based Gaussian–SVGD. Moreover, the vanilla gradient descent could also be replaced by accelerated ones or gradient descent with adaptive learning rate, e.g., AdaGrad, RMSProp.

**Resampling Scheme and Particle-Level Convergence.** As presented in the main text the Gaussian–SVGD for a general target is given by

$$\boldsymbol{x}_i^{(t+1)} = \boldsymbol{x}_i^{(t)} + \frac{\epsilon}{N}\left(\sum_{j=1}^N \nabla_{\boldsymbol{x}_j^{(t)}} K\big(\boldsymbol{x}_i^{(t)}, \boldsymbol{x}_j^{(t)}\big) - \sum_{j=1}^N K\big(\boldsymbol{x}_i^{(t)}, \boldsymbol{x}_j^{(t)}\big)\widehat{\nabla V}\big(\boldsymbol{x}_j^{(t)}\big)\right). \tag{45}$$

where $\widehat{\nabla V}(\boldsymbol{x}) = \widehat{\Gamma}_t(\boldsymbol{x} - \boldsymbol{\mu}_t) + \widehat{\boldsymbol{m}}_t$.

The updating rules of Gaussian–SVGD given above is totally deterministic, meaning that at each time step $\boldsymbol{x}_i^{(t)}$ is updated only using deterministic quantities and $\big(\boldsymbol{x}_k^{(t)}\big)_{k=1}^N$ without any external randomness. This is computationally more efficient but imposes difficulty in the analysis. On the other hand, we could also consider slightly modifying the updating rules by applying a resampling scheme to get a better estimation of $\boldsymbol{m}_t$ and $\Gamma_t$. In other words, we resample $(\boldsymbol{y}_k)_{k=1}^M$ i.i.d. from $\mathcal{N}(\boldsymbol{\mu}_t, \Sigma_t)$ and use the following estimators

$$\widehat{\boldsymbol{m}}_t = \frac{1}{M}\sum_{k=1}^M \nabla V(\boldsymbol{y}_k^{(t)}), \quad \widehat{\Gamma}_t = \frac{1}{M}\sum_{k=1}^M \nabla^2 V(\boldsymbol{y}_k^{(t)}), \tag{46}$$

or first-order estimators

$$\widehat{\Gamma}_t = \frac{1}{M}\sum_{k=1}^M \nabla V(\boldsymbol{y}_k^{(t)})(\boldsymbol{y}_k^{(t)} - \boldsymbol{\mu}_t)^\top \Sigma_t^{-1} = \frac{1}{M}\sum_{k=1}^M \Sigma_t^{-1}(\boldsymbol{y}_k^{(t)} - \boldsymbol{\mu}_t)\nabla V(\boldsymbol{y}_k^{(t)})^\top. \tag{47}$$

In this way, when $M$ is large enough $\widehat{\nabla V}(\boldsymbol{x})$ will be sufficiently close to $\Gamma_t(\boldsymbol{x} - \boldsymbol{\mu}_t) + \boldsymbol{m}_t$.

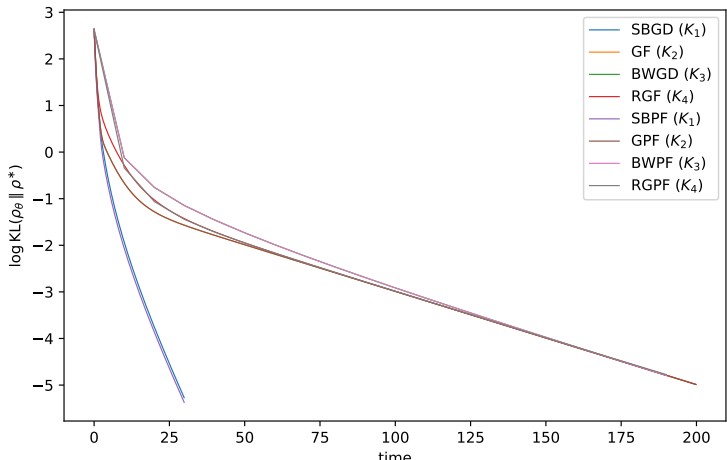

Figure 2: Convergence of Algorithms 1 and 2 with bilinear kernels for a Gaussian target.

Now we replace $\widehat{\boldsymbol{m}}_t$ and $\widehat{\Gamma}_t$ by $\boldsymbol{m}_t$ and $\Gamma_t$ and consider the continuous-time dynamics

$$\dot{\boldsymbol{x}}_i^{(t)} = \frac{1}{N}\left(\sum_{j=1}^N \nabla_{\boldsymbol{x}_j^{(t)}} K\big(\boldsymbol{x}_i^{(t)}, \boldsymbol{x}_j^{(t)}\big) - \sum_{j=1}^N K\big(\boldsymbol{x}_i^{(t)}, \boldsymbol{x}_j^{(t)}\big)\big(\Gamma_t(\boldsymbol{x}_j^{(t)} - \boldsymbol{\mu}_t) + \boldsymbol{m}_t\big)\right), \qquad (48)$$

where $\boldsymbol{\mu}_t$ and $\Sigma_t$ are sample mean and variance and

$$\boldsymbol{m}_t := \mathbb{E}_{\boldsymbol{x} \sim \mathcal{N}(\boldsymbol{\mu}_t, \Sigma_t)}[\nabla V(\boldsymbol{x})], \quad \Gamma_t := \mathbb{E}_{\boldsymbol{x} \sim \mathcal{N}(\boldsymbol{\mu}_t, \Sigma_t)}[\nabla^2 V(\boldsymbol{x})].$$

**Theorem D.1** (Equivalence of density-based and particle-based algorithms). *The solution of the finite-particle system* (48) *with $K_1$ is given by $\boldsymbol{x}_i^{(t)} = A_t(\boldsymbol{x}_i^{(0)} - \boldsymbol{\mu}_0) + \boldsymbol{\mu}_t$ where $A_t$ is the unique solution of*

$$\dot{A}_t = (I - \Gamma_t C_t - \boldsymbol{m}_t \boldsymbol{\mu}_t^\top) A_t, \quad A_0 = I,$$

*where $\boldsymbol{\mu}_t$ and $\Sigma_t$ are the unique solution of the ODE system*

$$\begin{cases} \dot{\boldsymbol{\mu}}_t = (I - \Gamma_t \Sigma_t)\,\boldsymbol{\mu}_t - (1 + \boldsymbol{\mu}_t^\top \boldsymbol{\mu}_t)\boldsymbol{m}_t \\ \dot{\Sigma}_t = 2\Sigma_t - \Sigma_t\left(\Sigma_t \Gamma_t + \boldsymbol{\mu}_t \boldsymbol{m}_t^\top\right) - \left(\Gamma_t \Sigma_t + \boldsymbol{m}_t \boldsymbol{\mu}_t^\top\right) \Sigma_t \end{cases}. \qquad (49)$$

This can be proved using exactly the same technique as in the proof of Theorem 3.6. Here (49) is the same as (22), and hence by Theorem 4.1 $\boldsymbol{\mu}_t$ and $\Sigma_t$ converges to $\boldsymbol{\mu}^*$ and $\Sigma^*$ that solves the GVI and the convergence rate is given in Theorem 4.2 when the target is strongly log-concave. We also conjecture that there is still uniform in time convergence to the mean-field limit for this particle system and leave it to future works. .

## E   Details of the Simulations

**Gaussian Targets.** Following [19], we consider a scenario where the target is Gaussian $\mathcal{N}(\boldsymbol{\mu}, \Sigma)$ where $\boldsymbol{\mu} \sim \mathrm{Unif}([0,1]^{10})$ and $\Sigma^{-1} = U\,\mathrm{diag}\{\lambda_1, \lambda_2, \cdots, \lambda_{10}\}U^\top$ with $U \in \mathbb{R}^{10 \times 10}$ drawn from the Haar measure of the orthogonal matrices $O(10)$ and $\lambda_1, \cdots, \lambda_{10}$ being a geometric sequence such that $\lambda_1 = 0.01$ and $\lambda_{10} = 1$. We run the eight different algorithms as introduced in Section 5 and show the decay of $\log \mathrm{KL}(\rho_\theta \parallel \rho^*)$ over time in Figure 2. Clearly the algorithms with $K_1$ show a faster rate over time compared to the others while the other algorithms eventually all converge at the same rate. This is actually confirmed from our theoretical analysis as in all the other dynamics except SBGD and SBPF, the mean converges at the rate of $\mathcal{O}(e^{-t/\lambda})$ while the covariance converges at a faster rate, resulting in the fact that the KL divergence converges at $\mathcal{O}(e^{-2t/\lambda})$. But for $K_1$ the rate is different and given in Theorem 3.1.

**Gaussian Mixture Targets.** Next we consider the 1-dimensional Gaussian mixture targets given by $w_1 \mathcal{N}(\mu_1, \sigma_1^2) + (1 - w_1)\mathcal{N}(\mu_2, \sigma_2^2)$. We run the aforementioned eight algorithms with initial

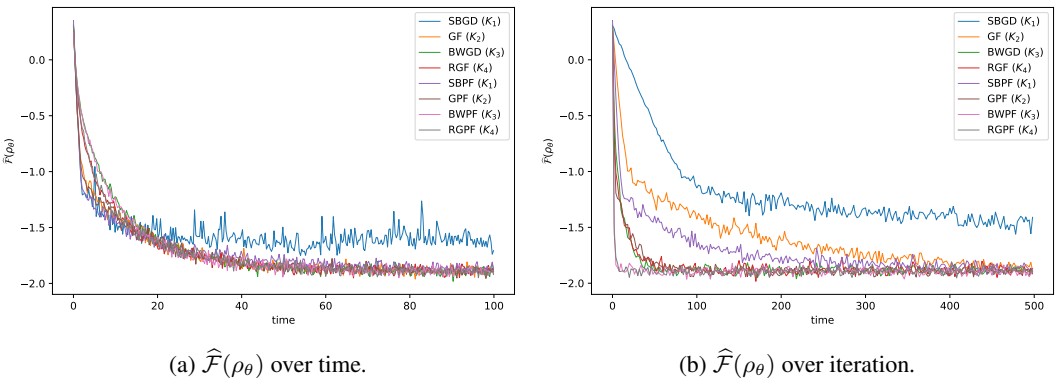

(a) $\widehat{\mathcal{F}}(\rho_\theta)$ over time.          (b) $\widehat{\mathcal{F}}(\rho_\theta)$ over iteration.

Figure 3: Convergence of Algorithms 1 and 2 with bilinear kernels for a Gaussian mixture target.

$\mu = 0$ and $\sigma = 1$ or particles drawn from $\mathcal{N}(0,1)$. Here again we plot the decay of $\widehat{\mathcal{F}}(\rho_\theta)$ over time or iteration as shown in Figure 3. In particular, the plots correspond to the specific setting of $\mu_1 = 5, \mu_2 = 10, \sigma_1 = 5, \sigma_2 = 2$ and $\rho^*(x) \propto 0.3 \exp(-(x-5)^2/50) + 0.7 \exp(-(x-10)^2/8)$. These parameters are arbitrarily chosen. For the decay of $\widehat{\mathcal{F}}(\rho_\theta)$ over time, we fix the step size to be $0.1$ and run 1000 iterations. For $\widehat{\mathcal{F}}(\rho_\theta)$ over time, we draw 500 particles for particle-based algorithms and run all algorithms for 500 iterations so that a total of 500 samples are drawn for the density-based algorithms. The step sizes are chosen to be the largest ones that still allow convergence. Specifically for these eight algorithms the step sizes are $0.02, 0.1, 1, 1, 0.2, 0.8, 8, 8$. Consistent with the results of Bayesian logistic regression, the particle-based ones are more stable and allow larger step sizes, with BWPF and RGPF clearly outperforming all the others. In fact, the constrast between particle-based and density-based algorithms is particularly significant in this problem probably because of the non-log-concave target.

**More on the Bayesian Logistic Regression.** We compare three so-far best performed algorithms, BWPF, RGPF, and FB-GVI [19] for the same problem with different step sizes. From Figure 4 we see that BWPF outperforms the other two. FB-GVI is better than RGPF with a larger learning rate but fluctuate a bit more when $\eta = 2$. This is probably attributed to the stochastic gradients. Furthermore, it is interesting to compare to ordinary gradient descent (OGD) on the variational parameters (mean and covariance) and SVGD with a radius-based kernel function (RBF-SVGD) $K_h(\boldsymbol{x}, \boldsymbol{y}) = \exp\left(-\frac{\|\boldsymbol{x}-\boldsymbol{y}\|^2}{2h^2}\right)$. We show this comparison in Figure 5 with the same step size $\eta = 2$. Firstly, OGD does not converge as fast as BWPF and it is not as stable. Secondly, RBF-SVGD is quite sensitive to the choice of bandwidth and it does not converge as fast as BWPF in general. We also notice that RBF-SVGD is significantly slower in computation compared to Gaussian–SVGD. However, as a particle-based algorithm, RBF-SVGD does have the advantage of being stable even when the step size is large.

# F   Analogous Results for the Affine-Invariant Bilinear Kernels

For the Bures–Wasserstein metric (Gaussian–Stein metric with $K_3$), the dynamics of natural gradient descent has already been studied in literature. See [60, 86, 13] for proofs for the following theorem.

**Theorem F.1** (Wasserstein gradient flow for the Gaussian family). *Let $\rho_0 \sim \mathcal{N}(\boldsymbol{\mu}_0, \Sigma_0)$ and $\rho^* \sim \mathcal{N}(\boldsymbol{b}, Q)$ be two Gaussian measures. Then the solution of* (1) *converges to $\rho^*$ as $t \to \infty$. In particular, $\rho_t$ is the density of $\mathcal{N}(\boldsymbol{\mu}_t, \Sigma_t)$ where the mean $\boldsymbol{\mu}_t$ and covariance matrix $\Sigma_t$ satisfies*

$$\begin{cases} \dot{\boldsymbol{\mu}}_t = -Q^{-1}(\boldsymbol{\mu}_t - \boldsymbol{b}) \\ \dot{\Sigma}_t = 2I - \Sigma_t Q^{-1} - Q^{-1}\Sigma_t \end{cases}. \tag{50}$$

*If $\Sigma_0 Q = Q\Sigma_0$, we have $\|\boldsymbol{\mu}_t - \boldsymbol{b}\| = \mathcal{O}(e^{-t/\lambda})$ and $\|\Sigma_t - Q\| = \mathcal{O}(e^{-2t/\lambda})$, where $\lambda$ is the largest eigenvalue of $Q$.*

Now we focus on the results for Gaussian–SVGD with the kernel $K_2$. First we remark that for $K_2$ the previous results on the well-posedness of mean-field PDE and finite-particle solutions in Appendix C still hold and can be proved using similar techniques to Appendix H but for simplicity they are

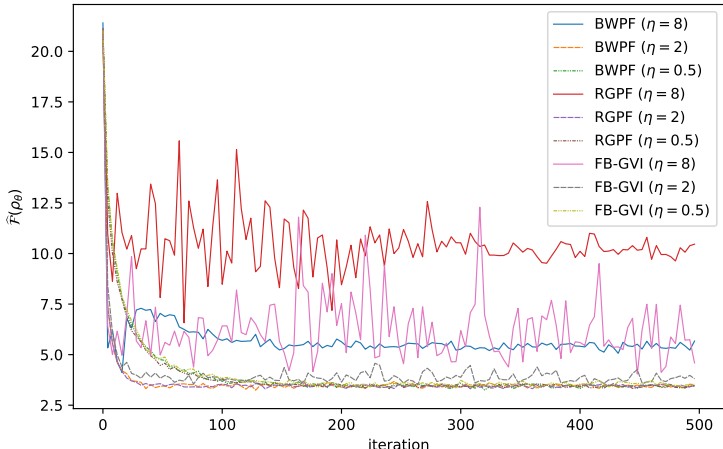

Figure 4: Performance of BWPF, RGPF, and FB-GVI with different step sizes $\eta$.

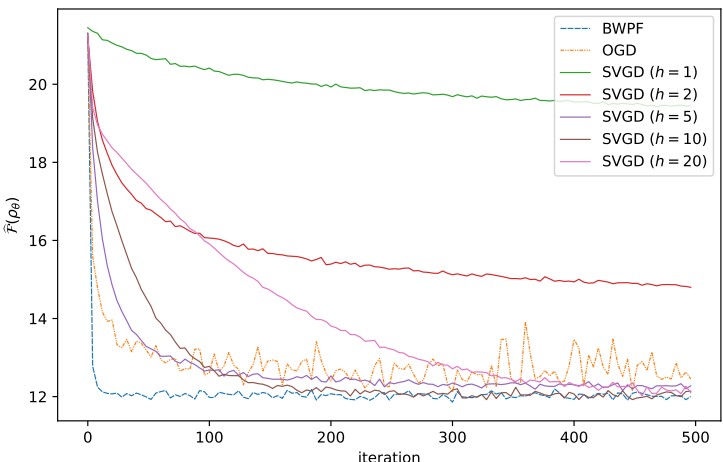

Figure 5: Performance of BWPF, OGD, and RBF-SVGD with bandwidth $h$.

omitted. The proofs of the following theorems will also be omitted unless a different proof technique from the analogous results needs to be applied.

**Theorem F.2** (Analogue of Theorem 3.1). *For any $t \geq 0$ the solution $\rho_t$ of SVGD (2) with the bilinear kernel remains a Gaussian density with mean $\boldsymbol{\mu}_t$ and covariance matrix $\Sigma_t$ given by*

$$\begin{cases} \dot{\boldsymbol{\mu}}_t = -Q^{-1}(\boldsymbol{\mu}_t - \boldsymbol{b}) \\ \dot{\Sigma}_t = 2\Sigma_t - \Sigma_t^2 Q^{-1} - Q^{-1}\Sigma_t^2 \end{cases}, \tag{51}$$

*which has a unique global solution on $[0, \infty)$ given any $\boldsymbol{\mu}_0 \in \mathbb{R}^d$ and $\Sigma_0 \in \mathrm{Sym}^+(d, \mathbb{R})$. And $\rho_t$ converges weakly to $\rho^*$ as $t \to \infty$. If $\Sigma_0 Q = Q\Sigma_0$ then we have the following rates*

$$\|\boldsymbol{\mu}_t - \boldsymbol{b}\| = \mathcal{O}(e^{-t/\lambda}), \quad \|\Sigma_t - Q\| = \mathcal{O}(e^{-2t}), \quad \forall \epsilon > 0,$$

*where $\lambda$ is the largest eigenvalue of $Q$.*

The proof of the dynamics is similar to that of Theorem 3.1. The rate $\mathcal{O}(e^{-t/\lambda})$ is trivially true from the theory of linear ODEs and the rate $\mathcal{O}(e^{-2t})$ is given by Theorem 3.2.

**Theorem F.3** (Analogue of Theorem 3.6). *Suppose the initial particles satisfy that $C_0$ is non-singular. There exists a unique solution of the finite particle system (13) given by*

$$\boldsymbol{x}_i^{(t)} = A_t(\boldsymbol{x}_i^{(0)} - \boldsymbol{\mu}_0) + \boldsymbol{\mu}_t, \tag{52}$$

where $A_t$ is the unique (matrix) solution of the linear system

$$\dot{A}_t = \left(I - Q^{-1}C_t\right) A_t, \quad A_0 = I, \tag{53}$$

and $\boldsymbol{\mu}_t$ and $C_t$ are the unique solution of the ODE system

$$\begin{cases} \dot{\boldsymbol{\mu}}_t = -Q^{-1}(\boldsymbol{\mu}_t - \boldsymbol{b}) \\ \dot{C}_t = 2C_t - C_t^2 Q^{-1} - Q^{-1}C_t^2 \end{cases}. \tag{54}$$

The proof is similar to that of Theorem 3.6.

**Theorem F.4** (Analogue of Theorem 3.7). *Given the same setting as Theorem F.3, further suppose the initial particles are drawn* i.i.d. *from* $\mathcal{N}(\boldsymbol{\mu}_0, \Sigma_0)$. *Then there exists a constant* $C_{d,Q,\boldsymbol{b},\Sigma_0,\boldsymbol{\mu}_0}$ *such that for all t, for all* $N \geq 2$, *with the empirical measure* $\zeta_N^{(t)} = \frac{1}{N}\sum_{i=1}^{N}\delta_{\boldsymbol{x}_i^{(t)}}$, *the second moment of Wasserstein-2 distance between* $\zeta_N^{(t)}$ *and* $\rho_t$ *converges:*

$$\mathbb{E}\left[\mathcal{W}_2^2\left(\zeta_N^{(t)}, \rho_t\right)\right] \leq C_{d,Q,\boldsymbol{b},\Sigma_0,\boldsymbol{\mu}_0} \times \begin{cases} N^{-1}\log\log N & \textit{if } d = 1 \\ N^{-1}(\log N)^2 & \textit{if } d = 2 \\ N^{-2/d} & \textit{if } d \geq 3 \end{cases}. \tag{55}$$

The proof is similar to that of Theorem 3.7. But for sake of completeness we provide more proof details in Appendix L.

**Theorem F.5** (Analogue of Theorem 4.1). *Let* $\rho^*$ *be the density of a target distribution with the potential function* $V(\boldsymbol{x}) = -\log\rho^*(\boldsymbol{x})$ *and* $\rho_0$ *be the density of* $\mathcal{N}(\boldsymbol{\mu}_0, \Sigma_0)$. *Then for any* $t \geq 0$ *the Gaussian–SVGD produces a Gaussian density* $\rho_t$ *with mean* $\boldsymbol{\mu}_t$ *and covariance matrix* $\Sigma_t$ *given by the following ODE system:*

$$\begin{cases} \dot{\boldsymbol{\mu}}_t = -\mathbb{E}_{\boldsymbol{x}\sim\rho_t}[\nabla V(\boldsymbol{x})] \\ \dot{\Sigma}_t = 2\Sigma_t - \Sigma_t^2\mathbb{E}_{\boldsymbol{x}\sim\rho_t}[\nabla^2 V(\boldsymbol{x})] - \mathbb{E}_{\boldsymbol{x}\sim\rho_t}[\nabla^2 V(\boldsymbol{x})]\Sigma_t^2 \end{cases}. \tag{56}$$

*Furthermore, suppose that* $\theta^*$ *is the unique solution of the following optimization problem*

$$\min_{\theta=(\boldsymbol{\mu},\Sigma)} \mathrm{KL}(\rho_\theta \parallel \rho^*), \text{ where } \rho_\theta \text{ is the Gaussian measure } \mathcal{N}(\boldsymbol{\mu},\Sigma).$$

*Then we have* $\rho_t \to \rho_{\theta^*} \sim \mathcal{N}(\boldsymbol{\mu}^*, \Sigma^*)$ *as* $t \to \infty$.

The proof is similar to that of Theorem 4.1. In particular, if the target is strongly log-concave, it gives rise to the following linear convergence rate.

**Theorem F.6** (Analogue of Theorem 4.2). *Assume that the target* $\rho^*$ *is* $\alpha$-*strongly log-concave and* $\beta$-*log-smooth, i.e.,* $\alpha I \preceq \nabla^2 V(\boldsymbol{x}) \preceq \beta I$. *Then* $\rho_t$ *converges to* $\rho_{\theta^*}$ *at the following rate:*

$$\|\boldsymbol{\mu}_t - \boldsymbol{\mu}^*\| = \mathcal{O}(e^{-\alpha t/\max\{\beta,1\}}), \quad \|\Sigma_t - \Sigma^*\| = \mathcal{O}(e^{-\alpha t/\max\{\beta,1\}}).$$

The proof is similar to that of Theorem 4.2.

**Theorem F.7** (Analogue of Theorem D.1). *The solution of the finite-particle system* (48) *with* $K_2$ *is given by* $\boldsymbol{x}_i^{(t)} = A_t(\boldsymbol{x}_i^{(0)} - \boldsymbol{\mu}_0) + \boldsymbol{\mu}_t$ *where* $A_t$ *is the unique solution of*

$$\dot{A}_t = (I - \Gamma_t C_t)A_t, \quad A_0 = I,$$

*where* $\boldsymbol{\mu}_t$ *and* $\Sigma_t$ *are the unique solution of the ODE system*

$$\begin{cases} \dot{\boldsymbol{\mu}}_t = -\boldsymbol{m}_t \\ \dot{\Sigma}_t = 2\Sigma_t - \Sigma_t^2\Gamma_t - \Gamma_t\Sigma_t^2 \end{cases}.$$

The proof is similar to that of Theorem 3.6 or Theorem D.1.

# G Proofs for Section B

**Lemma G.1.** *Let $\rho(\boldsymbol{x}) = (2\pi)^{-d/2}\big(\det(\Sigma)\big)^{-1/2}\exp\left(-\frac{1}{2}\boldsymbol{x}^\top\Sigma^{-1}\boldsymbol{x}\right)$ be the density of a d-dimensional normal random vector where $\Sigma$ is a positive definite matrix. Then for any $d \times d$ real matrix $A$ and $d$-dimensional real vector $\boldsymbol{b}$, we have*

$$\int \boldsymbol{x}^\top A\boldsymbol{x}\rho(\boldsymbol{x})\,\mathrm{d}\boldsymbol{x} = \mathrm{tr}(A\Sigma), \quad \int \boldsymbol{b}^\top\boldsymbol{x}A\boldsymbol{x}\rho(\boldsymbol{x})\,\mathrm{d}\boldsymbol{x} = A\Sigma\boldsymbol{b}.$$

*Proof.* Since $\Sigma^{-1}$ is a positive definite matrix, we can find its positive definite root $\Sigma^{-1/2}$. Let $\boldsymbol{y} = \Sigma^{-1/2}\boldsymbol{x}$ and $\rho_0(\boldsymbol{x}) = (2\pi)^{-d/2}\exp\left(-\frac{1}{2}\boldsymbol{x}^\top\boldsymbol{x}\right)$. Then we have

$$\int \boldsymbol{x}^\top A\boldsymbol{x}\rho(\boldsymbol{x})\,\mathrm{d}\boldsymbol{x}$$

$$= \int \boldsymbol{y}^\top\Sigma^{1/2}A\Sigma^{1/2}\boldsymbol{y}\,(\det(\Sigma))^{-1/2}\,\rho_0(\boldsymbol{y})\,(\det(\Sigma))^{1/2}\,\mathrm{d}\boldsymbol{y}$$

$$= \int \left(\sum_{j=1}^d (\Sigma^{1/2}A\Sigma^{1/2})_{jj}y_j^2\right)\rho_0(\boldsymbol{y})\,\mathrm{d}\boldsymbol{y}$$

$$= \sum_{j=1}^d (\Sigma^{1/2}A\Sigma^{1/2})_{jj}$$

$$= \mathrm{tr}(\Sigma^{1/2}A\Sigma^{1/2}) = \mathrm{tr}(A\Sigma).$$

The second equation is given by

$$\int \boldsymbol{b}^\top\boldsymbol{x}A\boldsymbol{x}\rho(\boldsymbol{x})\,\mathrm{d}\boldsymbol{x} = \int A\boldsymbol{x}\boldsymbol{x}^\top\boldsymbol{b}\rho(\boldsymbol{x})\,\mathrm{d}\boldsymbol{x}$$

$$= A\left(\int \boldsymbol{x}\boldsymbol{x}^\top\rho(\boldsymbol{x})\,\mathrm{d}\boldsymbol{x}\right)\boldsymbol{b} = A\Sigma\boldsymbol{b}.$$

$\square$

**Lemma G.2** (Lyapunov equation). *The Lyapunov equation*

$$PX + XP = Q$$

*has a unique solution. If $P, Q \in \mathrm{Sym}(d, \mathbb{R})$, then the solution $X \in \mathrm{Sym}(d, \mathbb{R})$.*

*Proof.* By Sylvester-Rosenblum theorem in control theory [7], $PX + XP = Q$ has a unique solution $X$. Note that if $P, Q \in \mathrm{Sym}(d, \mathbb{R})$ and $X$ is a solution, then $X^\top$ is also a solution. Thus, we have $X = X^\top$ which implies that $X \in \mathrm{Sym}(d, \mathbb{R})$. $\square$

*Proof of Theorem B.3.* Note that the tangent space at each $\theta \in \Theta = \mathbb{R}^d \times \mathrm{Sym}^+(d, \mathbb{R})$ is $T_\theta\Theta \simeq \mathbb{R}^d \times \mathrm{Sym}(d, \mathbb{R})$. If we define the inner product (pairing) on $T_\theta\Theta$ by

$$\langle \xi, \eta \rangle := \mathrm{tr}(\Sigma_1\Sigma_2) + \boldsymbol{\mu}_1^\top\boldsymbol{\mu}_2, \tag{57}$$

for any $\xi, \eta \in T_\theta\Theta$, where $\xi = \big(\widetilde{\boldsymbol{\mu}}_1, \widetilde{\Sigma}_1\big)$ and $\eta = \big(\widetilde{\boldsymbol{\mu}}_2, \widetilde{\Sigma}_2\big)$, then the tangent bundle $T\Theta$ is trivial. Since

$$\phi: \quad \Theta \quad \to \quad \mathcal{P}(\mathbb{R}^d)$$
$$\theta \quad \mapsto \quad \rho(\cdot, \theta)$$

provides an immersion from $\Theta$ to $\mathcal{P}(\mathbb{R}^d)$, we consider its pushforward $\mathrm{d}\phi_\theta$ given by

$$\mathrm{d}\phi_\theta: \quad T_\theta\Theta \quad \to \quad T_\rho\mathcal{P}(\mathbb{R}^d)$$
$$\xi \quad \mapsto \quad \langle\nabla_\theta\rho(\cdot, \theta), \xi\rangle.$$

On the other hand, for any $\Phi \in T^*_\rho \mathcal{P}(\mathbb{R}^d)$, the inverse canonical isomorphism of Stein metric maps it to

$$G_\rho^{-1}\Phi = -\nabla \cdot \rho(\cdot, \theta) \int K(\cdot, \boldsymbol{y})\rho(\boldsymbol{y}, \theta)\nabla\Phi(\boldsymbol{y})\,\mathrm{d}\boldsymbol{y} \in T_\rho \mathcal{P}(\mathbb{R}^d).$$

Thus, we obtain that

$$\langle \nabla_\theta \rho(\boldsymbol{x}, \theta), \xi \rangle = -\nabla_{\boldsymbol{x}} \cdot \rho(\boldsymbol{x}, \theta) \int K(\boldsymbol{x}, \boldsymbol{y})\rho(\boldsymbol{y}, \theta)\nabla\Phi(\boldsymbol{y})\,\mathrm{d}\boldsymbol{y} \tag{58}$$

for any $\boldsymbol{x} \in \mathbb{R}^d$.

Now we try to find a suitable function $\nabla\Phi_\xi$ that satisfies the equation above. We compute that

$$\begin{aligned}
&\langle \nabla_\theta \rho(\boldsymbol{x}, \theta), \xi \rangle \\
&= \mathrm{tr}\big(\nabla_\Sigma \rho(\boldsymbol{x}, \theta)\widetilde{\Sigma}_1\big) + \nabla_{\boldsymbol{\mu}}\rho(\boldsymbol{x}, \theta)^\top \widetilde{\boldsymbol{\mu}}_1 \\
&= \left( -\frac{1}{2}\left( \mathrm{tr}\left( \Sigma^{-1}\widetilde{\Sigma}_1 \right) - (\boldsymbol{x} - \boldsymbol{\mu})^T\Sigma^{-1}\widetilde{\Sigma}_1\Sigma^{-1}(\boldsymbol{x} - \boldsymbol{\mu}) \right) + \widetilde{\boldsymbol{\mu}}_1^\top \Sigma^{-1}(\boldsymbol{x} - \boldsymbol{\mu}) \right)\rho(\boldsymbol{x}, \theta).
\end{aligned}$$

Letting $\Psi(\boldsymbol{x}) = \int K(\boldsymbol{x}, \boldsymbol{y})\rho(\boldsymbol{y}, \theta)\nabla\Phi(\boldsymbol{y})\,\mathrm{d}\boldsymbol{y}$, we get

$$\begin{aligned}
&-\nabla_{\boldsymbol{x}} \cdot \rho(\boldsymbol{x}, \theta) \int K(\boldsymbol{x}, \boldsymbol{y})\rho(\boldsymbol{y}, \theta)\nabla\Phi(\boldsymbol{y})\,\mathrm{d}\boldsymbol{y} \\
&= -\nabla_{\boldsymbol{x}} \cdot \rho(\boldsymbol{x}, \theta)\Psi(\boldsymbol{x}) \\
&= \left( \Psi(\boldsymbol{x})^\top \Sigma^{-1}(\boldsymbol{x} - \boldsymbol{\mu}) - \nabla \cdot \Psi(\boldsymbol{x}) \right)\rho(\boldsymbol{x}, \theta).
\end{aligned}$$

We choose $\nabla\Phi_\xi(\boldsymbol{x}) = S_1(\boldsymbol{x} - \boldsymbol{\mu}) + \boldsymbol{b}_1$, where $S_1 \in \mathrm{Sym}(d, \mathbb{R})$ and $\boldsymbol{b}_1 \in \mathbb{R}^d$ will be determined later. Note that $S_1$ needs to be symmetric because the gradient field is curl-free. We derive that

$$\begin{aligned}
\Psi(\boldsymbol{x}) &= \int (\boldsymbol{x}^\top \boldsymbol{y} + 1)\rho(\boldsymbol{y}, \theta)\nabla\Phi_\xi(\boldsymbol{y})\,\mathrm{d}\boldsymbol{y} \\
&= \int (\boldsymbol{x}^\top \boldsymbol{y} + 1)\rho(\boldsymbol{y}, \theta)(S_1(\boldsymbol{y} - \boldsymbol{\mu}) + \boldsymbol{b}_1)\,\mathrm{d}\boldsymbol{y} \\
&= \int \big(\boldsymbol{x}^\top (\boldsymbol{y} + \boldsymbol{\mu}) + 1\big)\rho(\boldsymbol{y} + \boldsymbol{\mu}, \theta)\big(S_1\boldsymbol{y} + \boldsymbol{b}_1\big)\,\mathrm{d}\boldsymbol{y} \\
&= S_1\Sigma\boldsymbol{x} + (\boldsymbol{x}^\top \boldsymbol{\mu} + 1)\boldsymbol{b}_1 \\
&= (S_1\Sigma + \boldsymbol{b}_1\boldsymbol{\mu}^\top)\boldsymbol{x} + \boldsymbol{b}_1.
\end{aligned}$$

By comparison of the coefficients, we need

$$\begin{cases}
(S_1\Sigma + \boldsymbol{b}_1\boldsymbol{\mu}^\top)^\top\Sigma^{-1} + \Sigma^{-1}(S_1\Sigma + \boldsymbol{b}_1\boldsymbol{\mu}^\top) = \Sigma^{-1}\widetilde{\Sigma}_1\Sigma^{-1}, \\
\mathrm{tr}(S_1\Sigma + \boldsymbol{b}_1\boldsymbol{\mu}^\top) = \dfrac{1}{2}\mathrm{tr}(\Sigma^{-1}\widetilde{\Sigma}_1), \\
S_1\Sigma\boldsymbol{\mu} + (1 + \boldsymbol{\mu}^\top\boldsymbol{\mu})\boldsymbol{b}_1 = \widetilde{\boldsymbol{\mu}}_1.
\end{cases} \tag{59}$$

Note that the Lyapunov equation

$$PX + XP = Q,$$

where

$$P = \Sigma\left( I - \frac{1}{1 + \boldsymbol{\mu}^\top\boldsymbol{\mu}}\boldsymbol{\mu}\boldsymbol{\mu}^\top \right)\Sigma,$$

$$Q = \widetilde{\Sigma}_1 - \frac{1}{1 + \boldsymbol{\mu}^\top\boldsymbol{\mu}}(\Sigma\boldsymbol{\mu}\widetilde{\boldsymbol{\mu}}_1^\top + \widetilde{\boldsymbol{\mu}}_1\boldsymbol{\mu}^\top\Sigma),$$

has a unique solution $X = S_1 \in \mathrm{Sym}(d, \mathbb{R})$. Together with

$$\boldsymbol{b}_1 = \frac{1}{1 + \boldsymbol{\mu}^\top\boldsymbol{\mu}}(\widetilde{\boldsymbol{\mu}}_1 - S_1\Sigma\boldsymbol{\mu}),$$

we find that (59) holds with the unique solution described above. Now from the calculations above it is straightforward to check that (59) is equivalent to the following equation

$$G_\theta^{-1}\left(\boldsymbol{b}_1, \frac{1}{2}S_1\right) = \left(\widetilde{\boldsymbol{\mu}}_1, \widetilde{\Sigma}_1\right),$$

where $G_\theta^{-1}$ is the map defined in (29). Thus, the existence and uniqueness of the solution indicates that $G_\theta$ is an isomorphism.

Similarly, we let

$$\nabla\Phi_\eta(\boldsymbol{x}) = S_2(\boldsymbol{x} - \boldsymbol{\mu}) + \boldsymbol{b}_2,$$

where $X = S_2 \in \mathrm{Sym}(d, \mathbb{R})$ is the unique solution of the Lyapunov equation

$$PX + XP = Q,$$

where

$$P = \Sigma\left(I - \frac{1}{1 + \boldsymbol{\mu}^\top\boldsymbol{\mu}}\boldsymbol{\mu}\boldsymbol{\mu}^\top\right)\Sigma,$$

$$Q = \widetilde{\Sigma}_2 - \frac{1}{1 + \boldsymbol{\mu}^\top\boldsymbol{\mu}}(\Sigma\boldsymbol{\mu}\widetilde{\boldsymbol{\mu}}_2^\top + \widetilde{\boldsymbol{\mu}}_2\boldsymbol{\mu}^\top\Sigma),$$

and

$$\boldsymbol{b}_2 = \frac{1}{1 + \boldsymbol{\mu}^\top\boldsymbol{\mu}}(\widetilde{\boldsymbol{\mu}}_2 - S_2\Sigma\boldsymbol{\mu}).$$

Now we compute the Riemannian tensor

$$g_\theta(\xi, \eta) = g_\rho(\xi, \eta) = \int \Phi_\xi(\boldsymbol{x})G_\rho^{-1}\Phi_\eta(\boldsymbol{x})\,\mathrm{d}\boldsymbol{x}$$

$$= \int (\nabla\Phi_\xi(\boldsymbol{x}))^\top\rho(\boldsymbol{x},\theta)\int(\boldsymbol{x}^\top\boldsymbol{y} + 1)\rho(\boldsymbol{y},\theta)\nabla\Phi_\eta(\boldsymbol{y})\,\mathrm{d}\boldsymbol{y}\,\mathrm{d}\boldsymbol{x}$$

$$= \int \rho(\boldsymbol{x},\theta)\left(S_1(\boldsymbol{x} - \boldsymbol{\mu}) + \boldsymbol{b}_1\right)^\top\left((S_2\Sigma + \boldsymbol{b}_2\boldsymbol{\mu}^\top)\boldsymbol{x} + \boldsymbol{b}_2\right)\,\mathrm{d}\boldsymbol{x}$$

$$= \mathrm{tr}\left(S_1(S_2\Sigma + \boldsymbol{b}_2\boldsymbol{\mu}^\top)\Sigma\right) + \boldsymbol{b}_1^\top(S_2\Sigma\boldsymbol{\mu} + (1 + \boldsymbol{\mu}^\top\boldsymbol{\mu})\boldsymbol{b}_2)$$

$$= \mathrm{tr}(S_1S_2\Sigma^2) + (\boldsymbol{b}_1^\top S_2 + \boldsymbol{b}_2^\top S_1)\Sigma\boldsymbol{\mu} + (1 + \boldsymbol{\mu}^\top\boldsymbol{\mu})\boldsymbol{b}_1^\top\boldsymbol{b}_2.$$

Finally, we show that $G_\theta$ is indeed the canonical isomorphism corresponding to $g_\theta(\cdot, \cdot)$. We check that

$$g_\theta(\xi, \eta) = \mathrm{tr}\left(S_1(S_2\Sigma + \boldsymbol{b}_2\boldsymbol{\mu}^\top)\Sigma\right) + \boldsymbol{b}_1^\top(S_2\Sigma\boldsymbol{\mu} + (1 + \boldsymbol{\mu}^\top\boldsymbol{\mu})\boldsymbol{b}_2)$$

$$= \frac{1}{2}\mathrm{tr}(S_1\widetilde{\Sigma}_2) + \boldsymbol{b}_1^\top\widetilde{\boldsymbol{\mu}}_2 = \langle G_\theta(\xi), \eta\rangle.$$

$\square$

*Proof of Theorem B.7.* Similar to the proof of Theorem B.3, we define the inner product on $T_\theta\Theta$ by

$$\langle \xi, \eta\rangle := \mathrm{tr}(\Sigma_1\Sigma_2) + \boldsymbol{\mu}_1^\top\boldsymbol{\mu}_2,$$

for any $\xi, \eta \in T_\theta\Theta$, where $\xi = \left(\widetilde{\boldsymbol{\mu}}_1, \widetilde{\Sigma}_1\right)$ and $\eta = \left(\widetilde{\boldsymbol{\mu}}_2, \widetilde{\Sigma}_2\right)$. The map

$$\begin{aligned}\phi: \quad \Theta &\to \quad \mathcal{P}(\mathbb{R}^d)\\ \theta &\mapsto \quad \rho(\cdot, \theta),\end{aligned}$$

where $\rho(\cdot, \Sigma)$ denotes the density of $\mathcal{N}(\boldsymbol{\mu}, \Sigma)$, provides an immersion from $\Theta$ to $\mathcal{P}(\mathbb{R}^d)$. We consider its pushforward $\mathrm{d}\phi_\theta$ given by

$$\begin{aligned}\mathrm{d}\phi_\theta: \quad T_\theta\Theta &\to \quad T_\rho\mathcal{P}(\mathbb{R}^d)\\ \xi &\mapsto \quad \langle\nabla_\theta\rho(\cdot, \theta), \xi\rangle.\end{aligned}$$

On the other hand, for any $\Phi \in T_\rho^*\mathcal{P}(\mathbb{R}^d)$, the inverse canonical isomorphism of regularized Stein metric maps it to

$$G_\rho^{-1}\Phi = -\nabla \cdot \left(\rho((1 - \nu)\mathcal{T}_{K,\rho} + \nu I)^{-1}\mathcal{T}_{K,\rho}(\nabla\Phi)\right) \in T_\rho\mathcal{P}(\mathbb{R}^d).$$

Thus, we obtain that

$$\langle \nabla_\theta \rho(\boldsymbol{x}, \theta), \xi \rangle = -\nabla_{\boldsymbol{x}} \cdot \left( \rho(\boldsymbol{x}, \theta)((1-\nu)\mathcal{T}_{K,\rho} + \nu I)^{-1} \mathcal{T}_{K,\rho}(\nabla \Phi(\boldsymbol{x})) \right) \tag{60}$$

for any $\boldsymbol{x} \in \mathbb{R}^d$.

Now we try to find a suitable function $\nabla \Phi_\xi$ that satisfies the equation above. We compute that

$$\langle \nabla_\theta \rho(\boldsymbol{x}, \theta), \xi \rangle$$
$$= \operatorname{tr}\left( \nabla_\Sigma \rho(\boldsymbol{x}, \theta) \widetilde{\Sigma}_1 \right) + \nabla_{\boldsymbol{\mu}} \rho(\boldsymbol{x}, \theta)^\top \widetilde{\boldsymbol{\mu}}_1$$
$$= \left( -\frac{1}{2} \left( \operatorname{tr}\left( \Sigma^{-1} \widetilde{\Sigma}_1 \right) - (\boldsymbol{x} - \boldsymbol{\mu})^T \Sigma^{-1} \widetilde{\Sigma}_1 \Sigma^{-1} (\boldsymbol{x} - \boldsymbol{\mu}) \right) + \widetilde{\boldsymbol{\mu}}_1^\top \Sigma^{-1} (\boldsymbol{x} - \boldsymbol{\mu}) \right) \rho(\boldsymbol{x}, \theta).$$

Letting $\Psi(\boldsymbol{x}) = ((1-\nu)\mathcal{T}_{K,\rho} + \nu I)^{-1} \mathcal{T}_{K,\rho}(\nabla \Phi(\boldsymbol{x}))$, we get that the RHS of (60) is equal to

$$-\nabla_{\boldsymbol{x}} \cdot \rho(\boldsymbol{x}, \Sigma) \Psi(\boldsymbol{x}) = \left( \Psi(\boldsymbol{x})^\top \Sigma^{-1} (\boldsymbol{x} - \boldsymbol{\mu}) - \nabla \cdot \Psi(\boldsymbol{x}) \right) \rho(\boldsymbol{x}, \Sigma).$$

We choose $\nabla \Phi_\xi(\boldsymbol{x}) = S_1(\boldsymbol{x} - \boldsymbol{\mu}) + \boldsymbol{b}_1$, where $S_1 \in \operatorname{Sym}(d, \mathbb{R})$ and $\boldsymbol{b}_1 \in \mathbb{R}^d$ will be determined later. Note that $S_1$ needs to be symmetric because the gradient field is curl-free. We derive that

$$\mathcal{T}_{K,\rho}(\nabla \Phi_\xi(\boldsymbol{x})) = \int ((\boldsymbol{x} - \boldsymbol{\mu})^\top (\boldsymbol{y} - \boldsymbol{\mu}) + 1) \rho(\boldsymbol{y}, \theta) \nabla \Phi_\xi(\boldsymbol{y}) \, \mathrm{d}\boldsymbol{y}$$
$$= \int ((\boldsymbol{x} - \boldsymbol{\mu})^\top (\boldsymbol{y} - \boldsymbol{\mu}) + 1) \rho(\boldsymbol{y}, \theta)(S_1(\boldsymbol{y} - \boldsymbol{\mu}) + \boldsymbol{b}_1) \, \mathrm{d}\boldsymbol{y}$$
$$= S_1 \Sigma(\boldsymbol{x} - \boldsymbol{\mu}) + \boldsymbol{b}_1.$$

Thus, we know $\Psi(\boldsymbol{x}) = S_1 \Sigma ((1-\nu)\Sigma + \nu I)^{-1} \boldsymbol{x} + \boldsymbol{b}_1$. By comparison of the coefficients, we need $\boldsymbol{b}_1 = \widetilde{\boldsymbol{\mu}}_1$ and

$$\begin{cases} \Sigma ((1-\nu)\Sigma + \nu I)^{-1} S_1 \Sigma^{-1} + \Sigma^{-1} S_1 \Sigma ((1-\nu)\Sigma + \nu I)^{-1} = \Sigma^{-1} \widetilde{\Sigma}_1 \Sigma^{-1}, \\ \operatorname{tr}\left( S_1 \Sigma ((1-\nu)\Sigma + \nu I)^{-1} \right) = \frac{1}{2} \operatorname{tr}\left( \Sigma^{-1} \widetilde{\Sigma}_1 \right). \end{cases} \tag{61}$$

Note that the first equation is equivalent to the following Lyapunov equation

$$\Sigma^2 ((1-\nu)\Sigma + \nu I)^{-1} X + X \Sigma^2 ((1-\nu)\Sigma + \nu I)^{-1} = \widetilde{\Sigma}_1,$$

which has a unique solution $X = S_1 \in \operatorname{Sym}(d, \mathbb{R})$, and the second equation is automatically satisfied once we have the first one. Now from the calculations above it is straightforward to see that

$$G_\theta^{-1}(\boldsymbol{b}_1, S_1) = \left( \boldsymbol{b}_1, 2 \left( ((1-\nu)\Sigma + \nu I)^{-1} \Sigma^2 S_1 + S_1((1-\nu)\Sigma + \nu I)^{-1} \Sigma^2 \right) \right).$$

$\square$

The proofs of Theorems B.4 and B.6 are similar to that of Theorems B.3 and B.7. In particular, other proofs of Theorem B.6 can also be found in literature, for example, see [77, 60]. Finally, Corollary B.5 is the direct corollary of Theorem B.3 or Theorem B.4.

## H   Proofs for Section C

Given a probability measure $\mu$ and a Borel-measurable map $f$, we denote by $f_{\#}\mu$ the pushforward of the measure $\mu$ under the map $f$.

**Definition H.1** (Mean field characteristic flow). *Given a probability measure $\nu$, we say that the map*

$$X(t, \boldsymbol{x}, \nu) : [0, \infty) \times \mathbb{R}^d \to \mathbb{R}^d$$

*is a mean field characteristic flow associated to the particle system* (13) *or to the mean field PDE* (2) *if $X$ is $\mathcal{C}^1$ in time and solves the following problem*

$$\begin{aligned} \dot{X}(t, \boldsymbol{x}, \nu) &= -\left( \nabla K * \mu_t \right)(X(t, \boldsymbol{x}, \nu)) - \left( K * (\mu_t \nabla V) \right)(X(t, \boldsymbol{x}, \nu)) \\ \mu_t &= X(t, \cdot, \nu)_{\#}\nu \\ X(0, \boldsymbol{x}, \nu) &= \boldsymbol{x} \end{aligned} \tag{62}$$

Note that here $X(t, \cdot, \nu)_{\#}\nu$ is the push-forward of $\nu$ under the map $\boldsymbol{x} \mapsto X(t, \cdot, \nu)$, and $\{X(t, \cdot, \nu)\}_{t \geq 0, \nu}$ can be regarded as a family of maps from $\mathbb{R}^d$ to $\mathbb{R}^d$, parameterized by $t$ and $\nu$. In the lemma below we show that the mean field characteristic flow (62) is well-defined.

**Lemma H.2** (Solution of the mean field characteristic flow). *Assume the conditions of Assumption C.1 hold, and $\nu \in \mathcal{P}_V(\mathbb{R}^d)$. For any $T > 0$, there exists a unique solution $X(\cdot, \cdot, \nu) \in \mathcal{C}^1([0, T], Y)$ to the problem* (62), *where $Y$ is the function space given by*

$$Y := \left\{ u \in \mathcal{C}(\mathbb{R}^d, \mathbb{R}^d) : \sup_{\boldsymbol{x} \in \mathbb{R}^d} \frac{\|u(\boldsymbol{x}) - \boldsymbol{x}\|}{1 + \|\boldsymbol{x}\|} < \infty \right\}.$$

*Moreover, the measure $\mu_t = X(t, \cdot, \nu)_{\#}\nu$ satisfies*

$$\|\mu_t\|_{\mathcal{P}_V} \leq e^{Ct} \|\nu\|_{\mathcal{P}_V},$$

*for some constant $C$ that is independent of $\nu$.*

*Proof.* We prove the lemma in two steps. First we show local well-posedness of the mean field characteristic flow. Second we extend the local solution to $t \in [0, \infty)$.

Fix $r > 0$, and we define

$$Y_r := \left\{ u \in Y : \sup_{\boldsymbol{x} \in \mathbb{R}^d} \frac{\|u(\boldsymbol{x}) - \boldsymbol{x}\|}{1 + \|\boldsymbol{x}\|} \leq r \right\}.$$

We show that there exists $T_0 > 0$ such that (62) has a unique solution $X(t, \boldsymbol{x}, \nu)$ on $t \in [0, T_0]$, and the solution is in the following function class

$$S_r := \mathcal{C}\left([0, T_0], Y_r\right),$$

which is a complete metric space equipped with the uniform metric

$$d_S(u, v) := \sup_{t \in [0, T_0]} d_Y(u(t, \cdot), v(t, \cdot)), \quad d_Y(u, v) := \sup_{\boldsymbol{x} \in \mathbb{R}^d} \frac{\|u(\boldsymbol{x}) - v(\boldsymbol{x})\|}{1 + \|\boldsymbol{x}\|}.$$

Now we check the integral formulation of (62) given by

$$
\begin{aligned}
X(t, \boldsymbol{x}, \nu) =& \boldsymbol{x} - \int_0^t \int_{\mathbb{R}^d} \nabla_2 K\left(X(s, \boldsymbol{x}, \nu), X(s, \boldsymbol{x}', \nu)\right) \nu(\mathrm{d}\boldsymbol{x}') \, \mathrm{d}s \\
&- \int_0^t \int_{\mathbb{R}^d} K\left(X(s, \boldsymbol{x}, \nu), X(s, \boldsymbol{x}', \nu)\right) \nabla V\left(X(s, \boldsymbol{x}', \nu)\right) \nu(\mathrm{d}\boldsymbol{x}') \, \mathrm{d}s \\
=& \boldsymbol{x} - \int_0^t X(s, \boldsymbol{x}, \nu) \, \mathrm{d}s - \int_0^t \int_{\mathbb{R}^d} \nabla V\left(X(s, \boldsymbol{x}', \nu)\right) X(s, \boldsymbol{x}', \nu)^\top X(s, \boldsymbol{x}, \nu) \nu(\mathrm{d}\boldsymbol{x}') \, \mathrm{d}s.
\end{aligned}
$$

$$(63)$$

Let us define the operator $\mathcal{F} : u(t, \cdot) \mapsto \mathcal{F}(u)(t, \cdot)$ by

$$\mathcal{F}(u)(t, \boldsymbol{x}) := \boldsymbol{x} - \int_0^t u(s, \boldsymbol{x}) \, \mathrm{d}s - \int_0^t \int_{\mathbb{R}^d} \nabla V\left(u(s, \boldsymbol{x}')\right) u(s, \boldsymbol{x}')^\top u(s, \boldsymbol{x}) \nu\left(\mathrm{d}\boldsymbol{x}'\right) \mathrm{d}s.$$

We aim to show that $\mathcal{F}$ is a contraction map in $S_r$, and thus, it has a unique fixed point. For this purpose we first prove that $\mathcal{F}$ maps $S_r$ into $S_r$. It is straightforward to check that $(t, \boldsymbol{x}) \mapsto \mathcal{F}(u)(t, \boldsymbol{x})$. We now need to establish a bound on $\left\|\mathcal{F}(u)(t, \boldsymbol{x}) - \boldsymbol{x}\right\|$. If $u \in S_r$, then for any $s \in [0, T_0]$ and $\boldsymbol{x} \in \mathbb{R}^d$

$$\|u(s, \boldsymbol{x})\| \leq \|\boldsymbol{x}\| + \|u(s, \boldsymbol{x}) - \boldsymbol{x}\| \leq (r + 1)\|\boldsymbol{x}\| + r.$$

By Assumption C.1, we have that

$$
\begin{aligned}
\left\|\nabla V\left(u(s, \boldsymbol{x}')\right) u(s, \boldsymbol{x}')^\top\right\| &= \left\|\nabla V\left(u(s, \boldsymbol{x}')\right)\right\| \cdot \|u(s, \boldsymbol{x}')\| \\
&\leq (r + 1)(1 + \|\boldsymbol{x}'\|) \left\|\nabla V\left(u(s, \boldsymbol{x}')\right)\right\| \\
&\leq (r + 1) C_{r+1, r}(1 + V(\boldsymbol{x}')).
\end{aligned}
$$

As a consequence, we have

$$\big\|\mathcal{F}(u)(t,\boldsymbol{x}) - \boldsymbol{x}\big\|$$

$$\leq t\big((r+1)\|\boldsymbol{x}\| + r\big) + t\big((r+1)\|\boldsymbol{x}\| + r\big)(r+1)C_{r+1,r}\int(1 + V(\boldsymbol{x}'))\nu(\mathrm{d}\boldsymbol{x}')$$

$$\leq \widetilde{C}_r t(1 + \|\boldsymbol{x}\|),$$

for some constant $\widetilde{C}_r$, where we used the assumption that $\nu \in \mathcal{P}_V(\mathbb{R}^d)$. Therefore, choosing $T_0 \leq r/\widetilde{C}_r$ we get

$$\sup_{t\in[0,T_0]} \sup_{\boldsymbol{x}\in\mathbb{R}^d} \frac{\|\mathcal{F}(u)(t,\boldsymbol{x}) - \boldsymbol{x}\|}{1 + \|\boldsymbol{x}\|} \leq \widetilde{C}_r T_0 \leq r,$$

which shows that $\mathcal{F}$ maps from $S_r$ to $S_r$ for sufficiently small $T_0$. Next, we prove that $\mathcal{F}$ is indeed a contraction map. If $u, v \in S_r$, then for any $t \in [0, T_0]$ and $\boldsymbol{x} \in \mathbb{R}^d$

$$\big\|\mathcal{F}(u)(t,\boldsymbol{x}) - \mathcal{F}(v)(t,\boldsymbol{x})\big\|$$

$$\leq \int_0^t \big\|u(s,\boldsymbol{x}) - v(s,\boldsymbol{x})\big\|\,\mathrm{d}s + \int_0^t \int_{\mathbb{R}^d} \big\|\nabla V\left(u(s,\boldsymbol{x}')\right) u(s,\boldsymbol{x}')^\top\big\|\,\nu(\mathrm{d}\boldsymbol{x}')\,\|u(s,\boldsymbol{x}) - v(s,\boldsymbol{x})\|\,\mathrm{d}s$$

$$+ \int_0^t \int_{\mathbb{R}^d} \big\|\nabla V\left(u(s,\boldsymbol{x}')\right)\big\| \cdot \|u(s,\boldsymbol{x}') - v(s,\boldsymbol{x}')\|\,\nu(\mathrm{d}\boldsymbol{x}')\,\|v(s,\boldsymbol{x})\|\,\mathrm{d}s$$

$$+ \int_0^t \int_{\mathbb{R}^d} \big\|\nabla V\left(u(s,\boldsymbol{x}')\right) - \nabla V\left(v(s,\boldsymbol{x}')\right)\big\| \cdot \|v(s,\boldsymbol{x}')\|\,\nu(\mathrm{d}\boldsymbol{x}')\,\|v(s,\boldsymbol{x})\|\,\mathrm{d}s =: \mathrm{I} + \mathrm{II} + \mathrm{III} + \mathrm{IV}.$$

Term I can be upper-bounded by

$$\mathrm{I}/(1 + \|\boldsymbol{x}\|) \leq \int_0^t \frac{\big\|u(s,\boldsymbol{x}) - v(s,\boldsymbol{x})\big\|}{1 + \|\boldsymbol{x}\|}\,\mathrm{d}s \leq t d_S(u,v). \tag{64}$$

Similarly we have

$$\mathrm{II}/(1 + \|\boldsymbol{x}\|) \leq t d_S(u,v)(r+1)C_{r+1,r}\int(1 + V(\boldsymbol{x}'))\nu(\mathrm{d}\boldsymbol{x}'). \tag{65}$$

For the third term, we apply Assumption C.1 and get

$$\big\|\nabla V\left(u(s,\boldsymbol{x}')\right)\big\| \cdot \big\|u(s,\boldsymbol{x}') - v(s,\boldsymbol{x}')\big\|$$

$$= (1 + \|\boldsymbol{x}'\|)\big\|\nabla V\left(u(s,\boldsymbol{x}')\right)\big\| \cdot \frac{\big\|u(s,\boldsymbol{x}') - v(s,\boldsymbol{x}')\big\|}{1 + \|\boldsymbol{x}'\|}$$

$$\leq (r+1)C_{r+1,r}(1 + V(\boldsymbol{x}'))d_S(u,v).$$

Thus, we have

$$\mathrm{III}/(1 + \|\boldsymbol{x}\|) \leq (r+1)C_{r+1,r}\int(1 + V(\boldsymbol{x}'))\nu(\mathrm{d}\boldsymbol{x}')\int_0^t \frac{\|v(s,\boldsymbol{x})\|}{1 + \|\boldsymbol{x}\|}\,\mathrm{d}s$$

$$\leq t d_S(u,v)(r+1)^2 C_{r+1,r}\int(1 + V(\boldsymbol{x}'))\nu(\mathrm{d}\boldsymbol{x}'). \tag{66}$$

Finally applying Assumption C.1 once again, we have

$$\big\|\nabla V\left(u(s,\boldsymbol{x}')\right) - \nabla V\left(v(s,\boldsymbol{x}')\right)\big\| \cdot \|v(s,\boldsymbol{x}')\|$$

$$= (1 + \|\boldsymbol{x}'\|)\big\|\nabla V\left(u(s,\boldsymbol{x}')\right) - \nabla V\left(v(s,\boldsymbol{x}')\right)\big\| \cdot \frac{\|v(s,\boldsymbol{x}')\|}{1 + \|\boldsymbol{x}'\|}$$

$$\leq (r+1)(1 + \|\boldsymbol{x}'\|)\max_{\lambda\in[0,1]}\big\|\nabla^2 V(\lambda u(s,\boldsymbol{x}') + (1-\lambda)v(s,\boldsymbol{x}'))\big\| \cdot \big\|u(s,\boldsymbol{x}') - v(s,\boldsymbol{x}')\big\|$$

$$= (r+1)(1 + \|\boldsymbol{x}'\|)^2 \max_{\lambda\in[0,1]}\big\|\nabla^2 V(\lambda u(s,\boldsymbol{x}') + (1-\lambda)v(s,\boldsymbol{x}'))\big\| \frac{\big\|u(s,\boldsymbol{x}') - v(s,\boldsymbol{x}')\big\|}{1 + \|\boldsymbol{x}'\|}$$

$$\leq (r+1)C_{r+1,r}(1 + V(\boldsymbol{x}'))d_S(u,v),$$

where in $(*)$ we have used the fact that $\lambda u + (1-\lambda)v \in S_r$, and thus, $\lambda u + (1-\lambda)v$ also satisfies the inequality (3.3), which enables us to apply Assumption C.1.

Thus, Term IV is also bounded from above by (the same as the upper bound of Term III)

$$\text{IV}/(1+\|\boldsymbol{x}\|) \leq t d_S(u,v)(r+1)^2 C_{r+1,r} \int (1+V(\boldsymbol{x}'))\nu(\mathrm{d}\boldsymbol{x}'). \tag{67}$$

Now combining (64)–(67), we conclude that $\mathcal{F}$ is a contraction on $S_r$ when $T_0$ is small enough. By the contraction mapping theorem, $\mathcal{F}$ has a unique fixed point $X(\cdot,\cdot,\nu) \in S_r$, which solves (63). After defining $\mu_t = X(t,\cdot,\nu)_\#\nu$, one sees that $X(t,x,\nu)$ solves (62) in the small time interval $[0, T_0]$.

Now we proceed to the second step of extending the local solution. Define

$$\tau := \sup\left\{t \in \mathbb{R}^+ \cup \{\infty\} : \text{(62) has a (unique) solution on } [0,t)\right\}.$$

If $\tau = \infty$, then we have a global solution. Otherwise suppose $\tau < \infty$. After examining the bounds we have established in the previous step, we can see that supposing the local solution exists at some time $T_0$, it may be extended beyond $T_0$ as long as the quantity

$$\|\mu_t\|_{\mathcal{P}_V(\mathbb{R}^d)} = \int_{\mathbb{R}^d} (1+V(X(t,\boldsymbol{x},\nu)))\nu(\mathrm{d}\boldsymbol{x})$$

is finite at time $T_0$. We therefore establish an upper bound on this quantity.

$$\partial_t \int_{\mathbb{R}^d} (1+V(X(t,\boldsymbol{x},\nu)))\nu(\mathrm{d}\boldsymbol{x})$$

$$= -\int_{\mathbb{R}^d} \nabla V(X(t,\boldsymbol{x},\nu))^\top X(t,\boldsymbol{x},\nu)\nu(\mathrm{d}\boldsymbol{x})$$

$$\quad - \int_{\mathbb{R}^d} \int_{\mathbb{R}^d} \nabla V(X(s,\boldsymbol{x},\nu))^\top \nabla V(X(s,\boldsymbol{x}',\nu))X(s,\boldsymbol{x}',\nu)^\top X(s,\boldsymbol{x},\nu)\nu(\mathrm{d}\boldsymbol{x}')\nu(\mathrm{d}\boldsymbol{x})$$

$$\leq -\int_{\mathbb{R}^d} \nabla V(X(t,\boldsymbol{x},\nu))^\top X(t,\boldsymbol{x},\nu)\nu(\mathrm{d}\boldsymbol{x})$$

$$\leq C_{1,0} \int_{\mathbb{R}^d} (1+V(X(t,\boldsymbol{x},\nu)))\nu(\mathrm{d}\boldsymbol{x}).$$

The last inequality follows from Assumption C.1. Therefore, by Grönwall's inequality we get

$$\|\mu_t\|_{\mathcal{P}_V(\mathbb{R}^d)} = \int_{\mathbb{R}^d} (1+V(X(t,\boldsymbol{x},\nu)))\nu(\mathrm{d}\boldsymbol{x}) \leq e^{C_{1,0}t} \int_{\mathbb{R}^d} (1+V(\boldsymbol{x}))\nu(\mathrm{d}\boldsymbol{x}) = e^{C_{1,0}t}\|\nu\|_{\mathcal{P}_V(\mathbb{R}^d)},$$

holds for all $t \in [0,\tau)$. Next we show an upper bound on $\|X(t,\boldsymbol{x},\nu)\|$. We derive that

$$\partial_t \|X(t,\boldsymbol{x},\nu)\|^2 = 2X(t,\boldsymbol{x},\nu)^\top \dot{X}(t,\boldsymbol{x},\nu)$$

$$= -2X(t,\boldsymbol{x},\nu)^\top \left(I + \int_{\mathbb{R}^d} \nabla V(X(t,\boldsymbol{x}',\nu)) X(t,\boldsymbol{x}',\nu)^\top \nu(\mathrm{d}\boldsymbol{x}')\right) X(t,\boldsymbol{x},\nu)$$

$$\leq 2\|X(t,\boldsymbol{x},\nu)\|^2 \left(1 + C_{1,0} \int_{\mathbb{R}^d} (1+V(X(t,\boldsymbol{x}',\nu)))\nu(\mathrm{d}\boldsymbol{x}')\right)$$

$$\leq 2\|X(t,\boldsymbol{x},\nu)\|^2 \left(1 + C_{1,0}e^{C_{1,0}t}\|\nu\|_{\mathcal{P}_V(\mathbb{R}^d)}\right).$$

Again the first inequality follows from Assumption C.1. Thus, by Grönwall's inequality we have

$$\|X(t,\boldsymbol{x},\nu)\| \leq \exp\left(t + (e^{C_{1,0}t}-1)\|\nu\|_{\mathcal{P}_\Omega(\mathbb{R}^d)}\right)\|\boldsymbol{x}\|,$$

for all $t \in [0,\tau)$. This also implies that

$$\|\dot{X}(t,\boldsymbol{x},\nu)\|$$

$$\leq \left(1 + C_{1,0}e^{C_{1,0}t}\|\nu\|_{\mathcal{P}_V(\mathbb{R}^d)}\right)\|X(t,\boldsymbol{x},\nu)\|$$

$$\leq \left(1 + C_{1,0}e^{C_{1,0}t}\|\nu\|_{\mathcal{P}_V(\mathbb{R}^d)}\right)\exp\left(t + (e^{C_{1,0}t}-1)\|\nu\|_{\mathcal{P}_\Omega(\mathbb{R}^d)}\right)\|\boldsymbol{x}\|,$$

for all $t \in [0, \tau)$.

Next we extend the solution to $t \in [0, \tau]$. For this purpose, we first prove that for any sequence $\{t_i\}_{i=1}^{\infty}$ such that $0 < t_1 < t_2 < \cdots < \tau$ and $\lim_{i \to \infty} t_i = \tau$, the sequence of functions $\{X(t_i, \cdot, \nu)\}$ is Cauchy in $Y$. Then by completeness of $Y$, there is a limiting function for the sequence, and it is straightforward to see that such function is unique (does not rely on the choice of the sequence $\{t_i\}_{i=1}^{\infty}$).

In fact, we check that for any $i < j$

$$\|X(t_j, \boldsymbol{x}, \nu) - X(t_i, \boldsymbol{x}, \nu)\|$$

$$= (t_j - t_i) \int_0^1 \dot{X}(\lambda t_j + (1-\lambda)t_i, \boldsymbol{x}, \nu) \, \mathrm{d}\lambda$$

$$\leq (t_j - t_i) \sup_{t \in [t_i, t_j]} \|\dot{X}(t, \boldsymbol{x}, \nu)\|$$

$$\leq (t_j - t_i) \left(1 + C_{1,0} e^{C_{1,0}\tau} \|\nu\|_{\mathcal{P}_V(\mathbb{R}^d)}\right) \exp\left(\tau + (e^{C_{1,0}\tau} - 1)\|\nu\|_{\mathcal{P}_\Omega(\mathbb{R}^d)}\right) \|\boldsymbol{x}\|.$$

Thus, for any $\epsilon > 0$ there exists $N > 0$ such that for any $j > i > N$, we have

$$d_Y(X(t_j, \boldsymbol{x}, \nu), X(t_i, \boldsymbol{x}, \nu)) = \sup_{\boldsymbol{x}} \frac{\|X(t_j, \boldsymbol{x}, \nu) - X(t_i, \boldsymbol{x}, \nu)\|}{1 + \|\boldsymbol{x}\|} < \epsilon.$$

In other words, $\{X(t_i, \cdot, \nu)\}$ is a Cauchy sequence in $Y$.

Now we know that (62) has a unique solution on $[0, \tau]$. Since $\|\mu_\tau\|_{\mathcal{P}_V(\mathbb{R}^d)} < \infty$, we can further find a unique solution of (62) on $[\tau, \tau + T_0]$ for some $T_0$ small enough, which contradicts the definition of $\tau$. Therefore, we conclude that $\tau = \infty$, and (62) has a unique global solution. Finally, thanks to the integral formulation (63) $\dot{X}$ is continuous on $[0, \infty) \times \mathbb{R}^d$. The proof is complete. $\qquad\square$

*Proof of Theorem C.2.* Given $\nu$, let $X(t, x, \nu)$ be the mean field characteristic flow defined in (62), and let $\rho_t = X(t, \cdot, \nu)_{\#}\nu$. Note that (2) can be rewritten as

$$\dot{\rho}_t + \nabla \cdot (\rho_t U[\rho_t]) = 0,$$

where $U[\rho]$ is the vector field given by

$$U[\rho](\boldsymbol{x}) = -\boldsymbol{x} - \int_{\mathbb{R}^d} \nabla V(\boldsymbol{y}) \boldsymbol{y}^\top \boldsymbol{x} \nu(\mathrm{d}\boldsymbol{y}) = -\boldsymbol{x} - \int_{\mathbb{R}^d} \rho(\boldsymbol{y}) \nabla V(\boldsymbol{y}) \boldsymbol{y}^\top \boldsymbol{x} \, \mathrm{d}\boldsymbol{y}.$$

Then $\rho_t$ is a weak solution to (2) in the sense that

$$\sup_{t \in [0,T]} \|\rho_t\|_{\mathcal{P}_V} < \infty, \quad \forall T > 0,$$

and

$$\int_0^\infty \int_{\mathbb{R}^d} \left(\dot{\phi}(t, \boldsymbol{x}) + \nabla\phi(t, \boldsymbol{x})^\top U[\rho_t](\boldsymbol{x})\right) \rho_t(\mathrm{d}\boldsymbol{x}) \, \mathrm{d}t + \int_{\mathbb{R}^d} \phi(0, \boldsymbol{x})\nu(\mathrm{d}\boldsymbol{x}) = 0$$

holds for all $\phi \in \mathcal{C}_0^\infty([0, \infty) \times \mathbb{R}^d)$. This is either directly checked or follows immediately from Theorem 5.34 in [83].

By Lemma H.2 there exists some constant $C_1$ such that

$$\|\rho_t\|_{\mathcal{P}_V} \leq \|\nu\|_{\mathcal{P}_V}.$$

Suppose that $\nu \in \mathcal{P}^p(\mathbb{R}^d) \cap \mathcal{P}_V(\mathbb{R}^d)$. As shown in the proof of Lemma H.2, the map $X(t, \boldsymbol{x}, \nu)$ is an element of the space $Y$ with $d_Y(X(t, \boldsymbol{x}, \nu), \boldsymbol{x}) \leq C_1 e^{C_1 t}$. Therefore, since

$$\|X(t, \boldsymbol{x}, \nu)\|^p \leq 2^p \|\boldsymbol{x}\|^p + 2^p (1 + \|\boldsymbol{x}\|)^p d_Y(X(t, \boldsymbol{x}, \nu), \boldsymbol{x})^p,$$

we have

$$\|\rho_t\|_{\mathcal{P}^p} = \int_{\mathbb{R}^d} \|\boldsymbol{y}\|^p \rho_t(\mathrm{d}\boldsymbol{y}) = \int_{\mathbb{R}^d} \|X(t, \boldsymbol{x}, \nu)\|^p \nu(\mathrm{d}\boldsymbol{x}) \leq e^{C_2 t} \|\nu\|_{\mathcal{P}^p}$$

for some constant $C_2 > 0$ and $\rho_t \in \mathcal{P}^p(\mathbb{R}^d) \cap \mathcal{P}_V(\mathbb{R}^d)$.

We now explain that the uniqueness of the weak solution follows from the uniqueness of the mean field characteristic flow. Suppose $q \in \mathcal{C}\left([0,T], \mathcal{P}_V(\mathbb{R}^d)\right)$ is another weak solution to (2). By definition of the weak solution, the vector field $(t, \boldsymbol{x}) \mapsto U\left[q_t\right](\boldsymbol{x})$ is bounded over $[0,T] \times \mathbb{R}^d$, continuous in $t$ and Lipschitz continuous in $\boldsymbol{x}$. Then we can define a continuous family of maps $\widetilde{X}(t, \cdot, \nu)$ by

$$
\begin{aligned}
\dot{\widetilde{X}} &= U\left[q_t\right]\left(\widetilde{X}\right) \\
\widetilde{X}(0, \boldsymbol{x}, \nu) &= \boldsymbol{x}
\end{aligned}
$$

And the measure $\widetilde{q}_t = \widetilde{X}(t, \cdot, \nu)_{\#}\nu$ is a weak solution to the transport equation

$$
\dot{\widetilde{q}}_t + \nabla \cdot (\widetilde{q}_t U[q_t](\boldsymbol{x})) = 0
$$

with initial condition $\widetilde{q}_0 = \nu = q_0$. Uniqueness of the solution to this linear equation implies that $\widetilde{q}_t = q_t$. Thus, we have $\widetilde{X}(t, \cdot, \nu)_{\#}\nu = q_t$. In other words, $\widetilde{X}(t, \boldsymbol{x}, \nu)$ is the mean field characteristic flow for $\nu$. Uniqueness of the characteristic flow implies that $\widetilde{X} = X$, and hence $q_t = \rho_t$. Thus, we conclude that the weak solution is unique.

Lastly, we show the regularity result: If $\nu$ has a density $\rho_0(\boldsymbol{x}) \geq 0$, then $\rho_t$ also has a density. Furthermore, if $\rho_0 \in \mathcal{H}^k(\mathbb{R}^d)$ for some $k$, then we have $\rho_t \in \mathcal{H}^k(\mathbb{R}^d)$.

Note that we have already proven that $\rho_t \in \mathcal{C}\left([0,T], \mathcal{P}_V\right)$ and

$$
\|\rho_t\|_{\mathcal{P}_V} \leq e^{Ct}\|\rho_0\|_{\mathcal{P}_V}, \quad t \geq 0.
$$

Noting that

$$
\begin{aligned}
U[\rho](\boldsymbol{x}) &= -\boldsymbol{x} - \int_{\mathbb{R}^d} \nabla V(\boldsymbol{y})\boldsymbol{y}^\top \boldsymbol{x} \, \mathrm{d}\mu(\boldsymbol{y}) \\
&= -\boldsymbol{x} + \int_{\mathbb{R}^d} V(\boldsymbol{y})\boldsymbol{x} \, \mathrm{d}\mu(\boldsymbol{y}) \\
&= \int_{\mathbb{R}^d} (V(\boldsymbol{y}) - 1) \, \mathrm{d}\mu(\boldsymbol{y}) \, \boldsymbol{x},
\end{aligned}
$$

we have

$$
\begin{aligned}
&|U[\rho_t](\boldsymbol{x})| \leq e^{Ct}\|\rho_0\|_{\mathcal{P}_V}\|\boldsymbol{x}\|, \\
&\|\nabla U[\rho_t](\boldsymbol{x})\| \leq e^{Ct}\|\rho_0\|_{\mathcal{P}_V}, \\
&D_{\boldsymbol{x}}^j U[\rho_t](\boldsymbol{x}) = 0 \text{ for } j = 2, \cdots, k+1.
\end{aligned}
$$

Thus, $U(t, \boldsymbol{x}) := U[\rho_t](\boldsymbol{x}) \in \mathcal{C}\left([0,T], \mathcal{C}_b^{k+1}(\mathbb{R}^d)\right)$ where $\mathcal{C}_b^{k+1}(\mathbb{R}^d)$ is the space of continuous functions with bounded $(k+1)$-th order derivatives. Let $\Phi_t(\boldsymbol{x}) = X(t, \boldsymbol{x}, \nu)$ denote the characteristic flow. Since $\Phi_t$ satisfies the ODE system

$$
\partial_t \Phi_t(\boldsymbol{x}) = U[\rho_t](\Phi_t(\boldsymbol{x})),
$$

from the regularity theory of ODE systems (see Chapter 2 of [79]) we know that both the map $\boldsymbol{x} \mapsto \Phi_t$ and its inverse $\Phi_t^{-1}$ are $\mathcal{C}^k$. Therefore, if $\rho_0$ has a density, then $\rho_t$ also has a density and it is given by

$$
\rho_t(\boldsymbol{x}) = (\Phi_t)_{\#}\rho_0 = \rho_0(\Phi_t^{-1}(\boldsymbol{x})) \exp\left(-\int_0^t (\nabla_{\boldsymbol{x}} \cdot U[\rho_s])(\Phi_s \circ \Phi_t^{-1}(\boldsymbol{x})) \, \mathrm{d}s\right).
$$

Moreover, since $\rho$ satisfies

$$
\dot{\rho}_t = -\nabla \cdot (\rho_t U[\rho_t])
$$

with the vector field $U(t, \boldsymbol{x}) \in \mathcal{C}\left([0,T], \mathcal{C}_b^{k+1}(\mathbb{R}^d)\right)$, it follows from Lemma 2.8 of [44] that

$$
\rho \in \mathcal{C}\left([0,T], \mathcal{H}^k(\mathbb{R}^d)\right)
$$

for any $T \geq 0$. $\qquad\square$

*Proof of Theorem C.3.* We show that the particle system (13) is well-posed and that the empirical measure is a weak solution to the mean field PDE. We introduce the function

$$H_N(X_t) = \frac{1}{N} \sum_{i=1}^{N} V\left(\boldsymbol{x}_i^{(t)}\right) + 1.$$

Since $V$ is $\mathcal{C}^1$ hence locally Lipschitz, by Picard-Lindelöf theorem the problem (13) has a unique solution up to some time $T_0 > 0$. Intuitively we only need to show that the solution does not blow up at any finite time. We claim that for some constant $C$,

$$H_N(X_t) \leq H_N(X_0) \cdot e^{Ct}. \tag{68}$$

To establish this, we first differentiate $V(x_i(t))$ with respect to $t$ and sum over $i$:

$$\partial_t \left( \frac{1}{N} \sum_{i=1}^{N} V\left(\boldsymbol{x}_i^{(t)}\right) \right)$$

$$= -\frac{1}{N} \sum_{i=1}^{N} \nabla V\left(\boldsymbol{x}_i^{(t)}\right)^\top \boldsymbol{x}_i^{(t)} - \frac{1}{N^2} \sum_{i,j=1}^{N} \boldsymbol{x}_i^{(t)\top} \boldsymbol{x}_j^{(t)} \nabla V\left(\boldsymbol{x}_i^{(t)}\right)^\top \nabla V\left(\boldsymbol{x}_j^{(t)}\right)$$

$$\leq C_{1,0} \left( \frac{1}{N} \sum_{i=1}^{N} V\left(\boldsymbol{x}_i^{(t)}\right) + 1 \right).$$

Note that here we have used Assumption C.1. By Grönwall's inequality, (68) holds.

Now to be rigorous, once again we define

$$\tau := \sup\left\{ t \in \mathbb{R}^+ \cup \{\infty\} : (13) \text{ has a (unique) solution on } [0, t) \right\}.$$

If $\tau < \infty$, we define

$$\boldsymbol{x}_i^{(\tau)} := \lim_{t \nearrow \tau^-} \boldsymbol{x}_i^{(t)}.$$

Then (13) has a unique solution on $[0, \tau]$. Again by Picard–Lindelöf theorem, there exists some $\epsilon > 0$ such that (13) has a unique solution on $[\tau, \tau + \epsilon]$, which contradicts the definition of $\tau$. Thus, we conclude that $\tau = \infty$, which means that there is a global unique solution to (13).

Having established the well-posedness of the finite particle system, it now follows from the definition of the characteristic flow $X\left(t, \boldsymbol{x}, \mu_0^N\right)$ that

$$\boldsymbol{x}_i^{(t)} = X\left(t, \boldsymbol{x}_i^{(t)}, \mu_0^N\right)$$

and

$$\mu_t^N(\mathrm{d}\boldsymbol{x}) = \left(X\left(t, \cdot, \mu_0^N\right)\right)_\# \mu_0^N.$$

Similar to the proof of Theorem C.2, we conclude that $\mu_t^N$ is a weak solution to the mean field PDE (2). $\qquad \square$

Finally, we show Theorem C.5.

*Proof of Theorem C.5.* Recall that $p = q' = q/(q-1)$. By assumption $\|\nu_i\|_{\mathcal{P}^p} \leq R < \infty$ and the fact that $\mathcal{P}^p(\mathbb{R}^d) \subset \mathcal{P}_V(\mathbb{R}^d)$, we know that there exists $C > 0$ depending on $R$ such that

$$\|\nu_i\|_{\mathcal{P}_V} \leq C < \infty.$$

From the proof of Theorem C.2 and Definition H.1, we know that the weak solutions $\mu_{i,t}$ take the form

$$\mu_{i,t} = (X(t, \cdot, \nu_i))_\#, \quad i = 1, 2.$$

Now we bound $\mathcal{W}_p^p(\mu_{1,t}, \mu_{2,t})$ using $\mathcal{W}_p^p(\nu_1, \nu_2)$. Let $\pi^0$ be a coupling between $\nu_1$ and $\nu_2$. For $\delta > 0$ define $\phi_\delta(\boldsymbol{x}) := \frac{1}{p}(\|\boldsymbol{x}\| + \delta)^{p/2}$ to be an approximation to $\frac{1}{p}\|\boldsymbol{x}\|^p$. Given any two points $\boldsymbol{x}_1, \boldsymbol{x}_2 \in \mathbb{R}^d$, we have that

$$\partial_t \phi_\delta \left( X(t, \boldsymbol{x}_1, \nu_1) - X(t, \boldsymbol{x}_2, \nu_2) \right)$$

$$= - \nabla \phi_\delta \left( X(t, \boldsymbol{x}_1, \nu_1) - X(t, \boldsymbol{x}_2, \nu_2) \right)^\top$$

$$\left\{ \left( X(t, \boldsymbol{x}_1, \nu_1) - X(t, \boldsymbol{x}_2, \nu_2) \right) \right.$$

$$+ \left( \int_{\mathbb{R}^{2d}} \nabla V(X(t, \boldsymbol{x}_1', \nu_1)) X(t, \boldsymbol{x}_1', \nu_1)^\top X(t, \boldsymbol{x}_1, \nu_1) \nu_1(\mathrm{d}\boldsymbol{x}_1') \right.$$

$$\left. \left. - \int_{\mathbb{R}^{2d}} \nabla V(X(t, \boldsymbol{x}_2', \nu_2)) X(t, \boldsymbol{x}_2', \nu_2)^\top X(t, \boldsymbol{x}_2, \nu_2) \nu_2(\mathrm{d}\boldsymbol{x}_2') \right) \right\}$$

$$= - \nabla \phi_\delta \left( X(t, \boldsymbol{x}_1, \nu_1) - X(t, \boldsymbol{x}_2, \nu_2) \right)^\top$$

$$\left\{ \left( X(t, \boldsymbol{x}_1, \nu_1) - X(t, \boldsymbol{x}_2, \nu_2) \right) \right.$$

$$+ \int_{\mathbb{R}^{2d}} \nabla V(X(t, \boldsymbol{x}_1', \nu_1)) X(t, \boldsymbol{x}_1', \nu_1)^\top (X(t, \boldsymbol{x}_1, \nu_1) - X(t, \boldsymbol{x}_2, \nu_2)) \pi^0(\mathrm{d}\boldsymbol{x}_1' \, \mathrm{d}\boldsymbol{x}_2')$$

$$+ \int_{\mathbb{R}^{2d}} \nabla V(X(t, \boldsymbol{x}_1', \nu_1)) (X(t, \boldsymbol{x}_1', \nu_1) - X(t, \boldsymbol{x}_2', \nu_2))^\top X(t, \boldsymbol{x}_2, \nu_2) \pi^0(\mathrm{d}\boldsymbol{x}_1' \, \mathrm{d}\boldsymbol{x}_2')$$

$$\left. + \int_{\mathbb{R}^{2d}} (\nabla V(X(t, \boldsymbol{x}_1', \nu_1)) - \nabla V(X(t, \boldsymbol{x}_2', \nu_2))) X(t, \boldsymbol{x}_2', \nu_2)^\top X(t, \boldsymbol{x}_2, \nu_2) \pi^0(\mathrm{d}\boldsymbol{x}_1' \, \mathrm{d}\boldsymbol{x}_2') \right\}$$

$$=: I_1 + I_2 + I_3 + I_4.$$

Below we bound $I_i$ individually. First, noticing that

$$\|\nabla \phi_\delta(\boldsymbol{x})\| = \left\| (\|\boldsymbol{x}\|^2 + \delta)^{p/2-1} \boldsymbol{x} \right\| \leq \|\boldsymbol{x}\|^{p-1}, \tag{69}$$

we obtain

$$I_1 \leq \|X(t, \boldsymbol{x}_1, \nu_1) - X(t, \boldsymbol{x}_2, \nu_2)\|^p. \tag{70}$$

Next we bound $I_2$:

$$I_2 \leq \|X(t, \boldsymbol{x}_1, \nu_1) - X(t, \boldsymbol{x}_2, \nu_2)\|^p \left\| \int_{\mathbb{R}^{2d}} \nabla V(X(t, \boldsymbol{x}_1', \nu_1)) X(t, \boldsymbol{x}_1', \nu_1)^\top \nu_1(\mathrm{d}\boldsymbol{x}_1') \right\|$$

$$\overset{a}{\leq} \|X(t, \boldsymbol{x}_1, \nu_1) - X(t, \boldsymbol{x}_2, \nu_2)\|^p \cdot \tag{71}$$

$$\left( \int_{\mathbb{R}^{2d}} \|\nabla V(X(t, \boldsymbol{x}_1', \nu_1))\|^q \nu_1(\mathrm{d}\boldsymbol{x}_1') \right)^{1/q} \left( \int_{\mathbb{R}^{2d}} \|X(t, \boldsymbol{x}_1', \nu_1)\|^p \nu_1(\mathrm{d}\boldsymbol{x}_1') \right)^{1/p}$$

$$\overset{b}{\leq} \|X(t, \boldsymbol{x}_1, \nu_1) - X(t, \boldsymbol{x}_2, \nu_2)\|^p \, C_V^{1/q} \|\mu_{1,t}\|_{\mathcal{P}_V}^{1/q} \cdot \|\mu_{1,t}\|_{\mathcal{P}^p}$$

$$\overset{c}{\leq} C_V^{1/q} e^{(C_1/q + C_2)t} \|\nu_1\|_{\mathcal{P}_V}^{1/q} \cdot \|\nu_1\|_{\mathcal{P}^p} \|X(t, \boldsymbol{x}_1, \nu_1) - X(t, \boldsymbol{x}_2, \nu_2)\|^p. \tag{72}$$

Here we have applied Hölder's inequality in $a$ and Theorem C.2 in $c$. The inequality $b$ is due to Assumption C.4 and the definitions of the $\mathcal{P}_V$-norm and $\mathcal{P}^p$-norm.

Similarly for $I_3$ we use Hölder's inequality again and get

$$I_3 \leq \|X(t, \boldsymbol{x}_1, \nu_1) - X(t, \boldsymbol{x}_2, \nu_2)\|^{p-1} \cdot \|X(t, \boldsymbol{x}_2, \nu_2)\| \cdot$$
$$\int_{\mathbb{R}^{2d}} \|\nabla V(X(t, \boldsymbol{x}_1', \nu_1))\| \cdot \|X(t, \boldsymbol{x}_1', \nu_1) - X(t, \boldsymbol{x}_2', \nu_2)\| \, \pi^0(\mathrm{d}\boldsymbol{x}_1' \, \mathrm{d}\boldsymbol{x}_2')$$
$$\leq \|X(t, \boldsymbol{x}_1, \nu_1) - X(t, \boldsymbol{x}_2, \nu_2)\|^{p-1} \cdot \|X(t, \boldsymbol{x}_2, \nu_2)\| \cdot$$
$$\left( \int_{\mathbb{R}^{2d}} \|\nabla V(X(t, \boldsymbol{x}_1', \nu_1))\|^q \, \nu_1(\mathrm{d}\boldsymbol{x}_1') \right)^{1/q} \left( \int_{\mathbb{R}^{2d}} \|X(t, \boldsymbol{x}_1', \nu_1) - X(t, \boldsymbol{x}_2', \nu_2)\|^p \, \pi^0(\mathrm{d}\boldsymbol{x}_1' \, \mathrm{d}\boldsymbol{x}_2') \right)^{1/p}$$
$$\leq C_V^{1/q} e^{(C_1/q + C_2)t} \|\nu_1\|_{\mathcal{P}_V}^{1/q} \|X(t, \boldsymbol{x}_1, \nu_1) - X(t, \boldsymbol{x}_2, \nu_2)\|^{p-1} \|X(t, \boldsymbol{x}_2, \nu_2)\| \cdot$$
$$\left( \int_{\mathbb{R}^{2d}} \|X(t, \boldsymbol{x}_1', \nu_1) - X(t, \boldsymbol{x}_2', \nu_2)\|^p \, \pi^0(\mathrm{d}\boldsymbol{x}_1' \, \mathrm{d}\boldsymbol{x}_2') \right)^{1/p}. \tag{73}$$

Finally we proceed to bound $I_4$. An application of the intermediate value theorem to the difference of $\nabla V$ yields that

$$I_4 \leq \|X(t, \boldsymbol{x}_1, \nu_1) - X(t, \boldsymbol{x}_2, \nu_2)\|^{p-1} \cdot \|X(t, \boldsymbol{x}_2, \nu_2)\| \cdot$$
$$\int_{\mathbb{R}^{2d}} \sup_{\theta \in [0,1]} \left\| \nabla^2 V(\theta X(t, \boldsymbol{x}_1', \nu_1) + (1 - \theta) X(t, \boldsymbol{x}_2', \nu_2)) \right\| \cdot$$
$$\|X(t, \boldsymbol{x}_1', \nu_1) - X(t, \boldsymbol{x}_2', \nu_2)\| \cdot \|X(t, \boldsymbol{x}_2', \nu_2)\| \, \pi^0(\mathrm{d}\boldsymbol{x}_1' \, \mathrm{d}\boldsymbol{x}_2')$$
$$\leq \|X(t, \boldsymbol{x}_1, \nu_1) - X(t, \boldsymbol{x}_2, \nu_2)\|^{p-1} \|X(t, \boldsymbol{x}_2, \nu_2)\| \cdot$$
$$\left( \int_{\mathbb{R}^{2d}} \sup_{\theta \in [0,1]} \left\| \nabla^2 V(\theta X(t, \boldsymbol{x}_1', \nu_1) + (1 - \theta) X(t, \boldsymbol{x}_2', \nu_2)) \right\|^q \|X(t, \boldsymbol{x}_2', \nu_2)\|^q \, \pi^0(\mathrm{d}\boldsymbol{x}_1' \, \mathrm{d}\boldsymbol{x}_2') \right)^{1/q} \cdot$$
$$\left( \int_{\mathbb{R}^{2d}} \|X(t, \boldsymbol{x}_1', \nu_1) - X(t, \boldsymbol{x}_2', \nu_2)\|^p \, \pi^0(\mathrm{d}\boldsymbol{x}_1' \, \mathrm{d}\boldsymbol{x}_2') \right)^{1/p}$$
$$\overset{a}{\leq} C_V^{1/q} \|X(t, \boldsymbol{x}_1, \nu_1) - X(t, \boldsymbol{x}_2, \nu_2)\|^{p-1} \|X(t, \boldsymbol{x}_2, \nu_2)\| \cdot$$
$$\left( \int_{\mathbb{R}^{2d}} \left( \|V(X(t, \boldsymbol{x}_1', \nu_1))\| + \|V(X(t, \boldsymbol{x}_2', \nu_2))\| + \|X(t, \boldsymbol{x}_2', \nu_2)\|^q \right) \pi^0(\mathrm{d}\boldsymbol{x}_1' \, \mathrm{d}\boldsymbol{x}_2') \right)^{1/q} \cdot$$
$$\left( \int_{\mathbb{R}^{2d}} \|X(t, \boldsymbol{x}_1', \nu_1) - X(t, \boldsymbol{x}_2', \nu_2)\|^p \, \pi^0(\mathrm{d}\boldsymbol{x}_1' \, \mathrm{d}\boldsymbol{x}_2') \right)^{1/p}$$
$$\leq C_V^{1/q} \left( \|\mu_{1,t}\|_{\mathcal{P}_V}^{1/q} + \|\mu_{2,t}\|_{\mathcal{P}_V}^{1/q} + \|\mu_{2,t}\|_{\mathcal{P}^q} \right) \|X(t, \boldsymbol{x}_1, \nu_1) - X(t, \boldsymbol{x}_2, \nu_2)\|^{p-1} \|X(t, \boldsymbol{x}_2, \nu_2)\| \cdot$$
$$\left( \int_{\mathbb{R}^{2d}} \|X(t, \boldsymbol{x}_1', \nu_1) - X(t, \boldsymbol{x}_2', \nu_2)\|^p \, \pi^0(\mathrm{d}\boldsymbol{x}_1' \, \mathrm{d}\boldsymbol{x}_2') \right)^{1/p}$$
$$\overset{b}{\leq} C_V^{1/q} \left( e^{C_1 t/q} \|\nu_1\|_{\mathcal{P}_V}^{1/q} + e^{C_1 t/q} \|\nu_2\|_{\mathcal{P}_V}^{1/q} + e^{C_2 t} \|\nu_2\|_{\mathcal{P}^p} \right) \|X(t, \boldsymbol{x}_1, \nu_1) - X(t, \boldsymbol{x}_2, \nu_2)\|^{p-1} \cdot$$
$$\|X(t, \boldsymbol{x}_2, \nu_2)\| \left( \int_{\mathbb{R}^{2d}} \|X(t, \boldsymbol{x}_1', \nu_1) - X(t, \boldsymbol{x}_2', \nu_2)\|^p \, \pi^0(\mathrm{d}\boldsymbol{x}_1' \, \mathrm{d}\boldsymbol{x}_2') \right)^{1/p} \tag{74}$$

Note that to get $a$ we have applied (43) of Assumption C.4 and $b$ is implied by Theorem C.2.

If we define
$$D_p(\pi)(s) := \left( \int_{\mathbb{R}^{2d}} \|X(s, \boldsymbol{x}_1', \nu_1) - X(s, \boldsymbol{x}_2', \nu_2)\|^p \, \pi(\mathrm{d}\boldsymbol{x}_1' \, \mathrm{d}\boldsymbol{x}_2') \right)^{1/p},$$

combining (70), (71), (73) and (74) we obtain that
$$\partial_t \phi_\delta \left( X(t, \boldsymbol{x}_1, \nu_1) - X(t, \boldsymbol{x}_2, \nu_2) \right)$$
$$\leq 4 C_V^{1/q} e^{(C_1/q + C_2)t} R^{(q+1)/q} \|X(t, \boldsymbol{x}_1, \nu_1) - X(t, \boldsymbol{x}_2, \nu_2)\|^{p-1} \cdot$$
$$\left( \|X(t, \boldsymbol{x}_1, \nu_1) - X(t, \boldsymbol{x}_2, \nu_2)\| + D_p(\pi^0)(t) \|X(t, \boldsymbol{x}_2, \nu_2)\| \right).$$

Now integrating the inequality above with respect to the coupling $\pi^0(\mathrm{d}\boldsymbol{x}_1, \mathrm{d}\boldsymbol{x}_2)$ using the fact that

$$\int_{\mathbb{R}^{2d}} \|X(s, \boldsymbol{x}_1, \nu_1) - X(s, \boldsymbol{x}_2, \nu_2)\|^{p-1} \|X(s, \boldsymbol{x}_2, \nu_2)\| \pi^0(\mathrm{d}\boldsymbol{x}_1 \, \mathrm{d}\boldsymbol{x}_2)$$

$$\leq \left( \int_{\mathbb{R}^{2d}} \|X(s, \boldsymbol{x}_1, \nu_1) - X(s, \boldsymbol{x}_2, \nu_2)\|^p \, \pi^0(\mathrm{d}\boldsymbol{x}_1 \, \mathrm{d}\boldsymbol{x}_2) \right)^{(p-1)/p} \left( \int_{\mathbb{R}^{2d}} \|X(s, \boldsymbol{x}_2, \nu_2)\|^p \nu_2(\boldsymbol{x}_2) \right)^{1/p}$$

$$\leq e^{C_2 t} R D_p^{p-1}(\pi^0)(s).$$

Thus, we obtain that

$$\partial_t \phi_\delta\left(X(t, \boldsymbol{x}_1, \nu_1) - X(t, \boldsymbol{x}_2, \nu_2)\right) \leq 8 C_V^{1/q} e^{(C_1/q + 2C_2)t} R^{(2q+1)/q} D_p^p(\pi^0)(t) \leq C_{V,R} e^{CT} D_p^p(\pi^0)(t).$$

Integrating $t$ we get

$$\phi_\delta\left(X(t, \boldsymbol{x}_1, \nu_1) - X(t, \boldsymbol{x}_2, \nu_2)\right) \leq \phi_\delta(\boldsymbol{x}_1 - \boldsymbol{x}_2) + C_{V,R} e^{CT} \int_0^t D_p^p(\pi^0)(s) \, \mathrm{d}s.$$

Finally letting $\delta \to 0$ yields

$$D_p^p(\pi^0)(t) \leq D_p^p(\pi^0)(0) + C_{V,R} e^{CT} \int_0^t D_p^p(\pi^0)(s) \, \mathrm{d}s.$$

By Grönwall's inequality, we obtain that

$$D_p^p(\pi^0)(t) \leq D_p^p(0) \exp\left(C_{V,R} e^{CT} t\right).$$

Now since $\pi^0 \in \Gamma(\nu_1, \nu_2)$ and $\mu_{i,t} = (X(t, \cdot, \nu_i))_{\#}\nu_i$, the mapping

$$\Xi_t : (\boldsymbol{x}_1, \boldsymbol{x}_2) \in \mathbb{R}^{2d} \mapsto (X(t, \boldsymbol{x}_1, \nu_1), X(t, \boldsymbol{x}_2, \nu_2)) \in \mathbb{R}^{2d}$$

satisties that $(\Xi_t)_{\#}\pi^0 \in \Gamma(\mu_{1,t}, \mu_{2,t})$. As a consequence we have that

$$\begin{aligned}
\mathcal{W}_p^p(\mu_{1,t}, \mu_{2,t}) &= \inf_{\pi \in \Gamma(\mu_{1,t}, \mu_{2,t})} \int_{\mathbb{R}^{2d}} \|\boldsymbol{x}_1 - \boldsymbol{x}_2\|^p \pi(\mathrm{d}\boldsymbol{x}_1 \, \mathrm{d}\boldsymbol{x}_2) \\
&\leq \inf_{\pi^0 \in \Gamma(\nu_1, \nu_2)} D_p^p(\pi^0)(t) \\
&\leq \exp\left(C_{V,R} e^{CT} T\right) \inf_{\pi^0 \in \Gamma(\nu_1, \nu_2)} D_p^p(\pi^0)(0) \\
&= \exp\left(C_{V,R} e^{CT} T\right) \cdot \mathcal{W}_p^p(\nu_1, \nu_2).
\end{aligned}$$

$\square$

# I   Proofs for Section 3.1

*Proof of Theorem 3.1.* For any $\theta = (\boldsymbol{\mu}, \Sigma) \in \Theta$, define $\widetilde{E}(\boldsymbol{\mu}, \Sigma) := E(\rho)$. Then we have

$$\widetilde{E}(\boldsymbol{\mu}, \Sigma) = \mathrm{KL}(\rho \,\|\, \rho^*) = \frac{1}{2}\left(\mathrm{tr}(Q^{-1}\Sigma) - \log\det(Q^{-1}\Sigma) - d + (\boldsymbol{\mu} - \boldsymbol{b})^\top Q^{-1}(\boldsymbol{\mu} - \boldsymbol{b})\right),$$

where $\rho$ is the density of $\mathcal{N}(\boldsymbol{\mu}, \Sigma)$ and $\rho^*$ is the density of $\mathcal{N}(\boldsymbol{b}, Q)$.

Now we consider the gradient flow on the submanifold $\Theta$

$$\dot{\theta}_t = -G_{\theta_t}^{-1}(\nabla_{\theta_t}\widetilde{E}(\theta_t)).$$

For clarity note that here

$$\nabla_{\theta_t}\widetilde{E}(\theta_t) := \left(\nabla_{\boldsymbol{\mu}_t}\widetilde{E}(\boldsymbol{\mu}_t, \Sigma_t), \ \nabla_{\Sigma_t}\widetilde{E}(\boldsymbol{\mu}_t, \Sigma_t)\right),$$

where $\nabla_\Sigma \widetilde{E}(\boldsymbol{\mu}, \Sigma)$ denotes the standard matrix derivative (not the covariant derivative or affine connection in the contexts of Riemannian geometry). We calculate that

$$\nabla_{\boldsymbol{\mu}}\widetilde{E}(\boldsymbol{\mu}, \Sigma) = Q^{-1}(\boldsymbol{\mu} - \boldsymbol{b}), \quad \nabla_\Sigma \widetilde{E}(\boldsymbol{\mu}, \Sigma) = \frac{1}{2}(Q^{-1} - \Sigma^{-1}).$$

Thus, the gradient flow on $\Theta$ is equivalent to

$$\Leftrightarrow \begin{cases} \dot{\boldsymbol{\mu}}_t = -\left(2\nabla_{\Sigma_t}\widetilde{E}(\boldsymbol{\mu}_t,\Sigma_t)\Sigma_t\boldsymbol{\mu}_t + (1+\boldsymbol{\mu}_t^\top\boldsymbol{\mu}_t)\nabla_{\boldsymbol{\mu}_t}\widetilde{E}(\boldsymbol{\mu}_t,\Sigma_t)\right) \\ \dot{\Sigma}_t = -\Sigma_t\left(2\Sigma_t\nabla_{\Sigma_t}\widetilde{E}(\boldsymbol{\mu}_t,\Sigma_t) + \boldsymbol{\mu}_t\nabla_{\boldsymbol{\mu}_t}^\top\widetilde{E}(\boldsymbol{\mu}_t,\Sigma_t)\right) \\ \qquad -\left(2\nabla_{\Sigma_t}\widetilde{E}(\boldsymbol{\mu}_t,\Sigma_t)\Sigma_t + \nabla_{\boldsymbol{\mu}_t}\widetilde{E}(\boldsymbol{\mu}_t,\Sigma_t)\boldsymbol{\mu}_t^\top\right)\Sigma_t \end{cases}$$

$$\Leftrightarrow \begin{cases} \dot{\boldsymbol{\mu}}_t = (I - Q^{-1}\Sigma_t)\boldsymbol{\mu}_t - (1+\boldsymbol{\mu}_t^\top\boldsymbol{\mu}_t)Q^{-1}(\boldsymbol{\mu}_t - \boldsymbol{b}) \\ \dot{\Sigma}_t = 2\Sigma_t - \Sigma_t\left(\Sigma_t + \boldsymbol{\mu}_t(\boldsymbol{\mu}_t - \boldsymbol{b})^\top\right)Q^{-1} - Q^{-1}\left(\Sigma_t + (\boldsymbol{\mu}_t - \boldsymbol{b})\boldsymbol{\mu}_t^\top\right)\Sigma_t \end{cases}. \tag{75}$$

Note that it is trivial to check that the functions on the right-hand-side of (3.1) are locally Lipschitz with respect to $\boldsymbol{\mu}_t$ and $\Sigma_t$ (continuously differentiable hence locally Lipschitz). By Picard-Lindelöf theorem, this ODE system given $\boldsymbol{\mu}_0 \in \mathbb{R}^d$, $\Sigma_0 \in \mathrm{Sym}^+(d,\mathbb{R})$ has a unique solution on $t \in [0,\epsilon)$ for some $\epsilon > 0$. Let

$$s := \sup\left\{t \in \mathbb{R}^+ \cup \{\infty\} : (75) \text{ has a (unique) solution on } [0,s)\right\}.$$

For convenience we define the curve on $\Theta = \mathbb{R}^d \times \mathrm{Sym}^+(d,\mathbb{R})$ by

$$\gamma : [0,s) \to \mathbb{R}^d \times \mathrm{Sym}(d,\mathbb{R}), \quad \gamma(t) := (\boldsymbol{\mu}_t, \Sigma_t).$$

Next we consider the density flow given by

$$\dot{\rho}_t = -G_{\rho_t}^{-1}\frac{\delta E(\rho_t)}{\delta\rho_t} = \nabla\cdot\left(\rho_t(\cdot)\int K(\cdot,\boldsymbol{y})\big(\nabla\rho_t(\boldsymbol{y}) + \rho_t(\boldsymbol{y})\nabla V(\boldsymbol{y})\big)\,\mathrm{d}\boldsymbol{y}\right). \tag{76}$$

By Theorem C.2 we know that there is a unique solution $\rho_t$ in $\mathcal{P}(\mathbb{R}^d)$ for $t \in [0,\infty)$. We claim that

**Claim.** $\rho_t$ *is a Gaussian density for* $t \in [0,s)$.

In fact, if we let $\widetilde{\rho}_t := \phi(\gamma(t))$, where $\phi$ is the immersion

$$\begin{aligned} \phi: \quad \Theta \quad &\to \quad \mathcal{P}(\mathbb{R}^d) \\ \theta \quad &\mapsto \quad \rho(\cdot,\theta). \end{aligned}$$

By uniqueness of the solution it suffices to prove that (76) holds for $\rho_t = \widetilde{\rho}_t$. This can be checked by direct calculation of course. But there is also a more elegant way to show it. We consider the following commutative diagram.

$$\begin{array}{ccc} T_\theta\Theta & \xrightarrow{\mathrm{d}\phi_\theta} & T_{\widetilde{\rho}}\mathcal{P}(\mathbb{R}^d) \\ {\scriptstyle G_\theta}\downarrow & & \downarrow{\scriptstyle G_{\widetilde{\rho}}} \\ T_\theta^*\Theta & \xrightarrow{\psi_\theta} & T_{\widetilde{\rho}}^*\mathcal{P}(\mathbb{R}^d) \end{array}$$

Here $\mathrm{d}\phi_\theta$ is the pushforward of the immersion $\phi$ at point $\theta$ and $\psi_\theta$ is the inverse of the pullback map $\phi^* : T_{\widetilde{\rho}}^*\mathcal{P}(\mathbb{R}^d) \to T_\theta^*\Theta$ restricted on $\mathrm{Im}\,G_{\widetilde{\rho}_t}\circ\mathrm{d}\phi_\theta(\simeq T_\theta^*\Theta)$. The diagram is commutative due to the fact that $G_\theta$ is the canonical isomorphism on the submanifold $\Theta$ induced from $\mathcal{P}(\mathbb{R}^d)$.

Now we show that $\frac{\delta E}{\delta\widetilde{\rho}_t} \in \mathrm{Im}\,G_{\widetilde{\rho}_t}\circ\mathrm{d}\phi_\theta$. In the proof of Theorem B.3, we have shown that $\psi_\theta$ maps $\left(\boldsymbol{b},\frac{1}{2}S\right)$ to some $\Phi$ such that $\nabla\Phi(\boldsymbol{x}) = S(\boldsymbol{x}-\boldsymbol{\mu})+\boldsymbol{b}$, where $\boldsymbol{\mu}$ is obtained from $G_\theta^{-1}\left(\boldsymbol{b},\frac{1}{2}S\right)$. Thus, $\mathrm{Im}\,\psi_\theta$ contains all functions (or more precisely the equivalent classes of functions that differ by a constant) with the form

$$\Phi(\boldsymbol{x}) = \frac{1}{2}(\boldsymbol{x}-\boldsymbol{\mu})^\top S(\boldsymbol{x}-\boldsymbol{\mu}) + \boldsymbol{b}\boldsymbol{x} + C, \quad S \in \mathrm{Sym}(d,\mathbb{R}),\ \boldsymbol{b} \in \mathbb{R}^d,\ C \text{ is any constant}.$$

In other words, $\mathrm{Im}\,\psi_\theta$ contains all quadratic forms on $\mathbb{R}^d$. Note that we can derive that

$$\frac{\delta E(\widetilde{\rho}_t)}{\delta\widetilde{\rho}_t} = \log\widetilde{\rho}_t + V$$

is exactly a quadratic form. Thus,

$$\frac{\delta E}{\delta \widetilde{\rho}_t} \in \operatorname{Im} \psi_{\theta_t} = \operatorname{Im} \psi_{\theta_t} \circ G_{\theta_t} = \operatorname{Im} G_{\widetilde{\rho}_t} \circ \mathrm{d}\phi_\theta.$$

Next since $\mathrm{d}\phi_{\theta_t}$ maps the tangent vector $\frac{\partial}{\partial \theta_t} \in T_{\theta_t}\Theta$ to $\frac{\delta}{\delta \widetilde{\rho}_t} \in T_{\widetilde{\rho}_t}\mathcal{P}(\mathbb{R}^d)$, we have that

$$\phi^* \frac{\delta E}{\delta \widetilde{\rho}_t} = \nabla_{\theta_t}\widetilde{E}.$$

Combining this with the fact that $\frac{\delta E}{\delta \widetilde{\rho}_t} \in \operatorname{Im} G_{\widetilde{\rho}_t} \circ \mathrm{d}\phi_\theta$, we get

$$\frac{\delta E}{\delta \widetilde{\rho}_t} = \psi_\theta(\nabla_{\theta_t}\widetilde{E}).$$

Thus, we have

$$G_{\widetilde{\rho}_t}^{-1}\frac{\delta E}{\delta \widetilde{\rho}_t} = G_{\widetilde{\rho}_t}^{-1}\psi_\theta(\nabla_{\theta_t}\widetilde{E}) = \mathrm{d}\phi_{\theta_t} G_{\theta_t}^{-1}(\nabla_{\theta_t}\widetilde{E}).$$

And we conclude

$$\dot{\widetilde{\rho}}_t = \mathrm{d}\phi_{\theta_t}\dot{\theta}_t = -\mathrm{d}\phi_{\theta_t} G_{\theta_t}^{-1}(\nabla_{\theta_t}\widetilde{E}) = -G_{\widetilde{\rho}_t}^{-1}\frac{\delta E}{\delta \widetilde{\rho}_t}.$$

The claim is proven.

Now back to the original problem. Suppose $s < \infty$. Since we know that the mean field PDE (76) has a unique solution on $[0, \infty)$, in particular, it exists on $[0, s]$. Note that the weak limit of Gaussian distributions is Gaussian and since $\rho_s \in \mathcal{P}(\mathbb{R}^d)$ it does not degenerate. By definition of $\phi$ we have that $\phi^{-1}(\rho_s) \in \Theta$. By letting $(\boldsymbol{\mu}_s, \Sigma_s) = \gamma(s) := \phi^{-1}(\rho_s)$, we obtain the solution of (75) on $[0, s]$. Again by Picard-Lindelöf theorem there exists a small neighborhood $[s, s+\epsilon')$ such that (75) has a unique solution. This together with the solution on $[0, s]$ contradicts the definition of $s$. Therefore, we conclude that $s = \infty$. (75) has a unique global solution corresponding to the mean and covariance matrix of $\rho_t$.

Next, we prove that $\rho_t$ converges weakly to $\rho^*$ as $t \to \infty$. We calculate the quantity $\dot{\widetilde{E}}(\boldsymbol{\mu}_t, \Sigma_t)$. By Jacobi's formula in matrix calculus (Theorem 8.1 in [59]), we have

$$\partial_t \det \Sigma_t = \det \Sigma_t \operatorname{tr}(\Sigma_t^{-1}\dot{\Sigma}_t).$$

Thus, we derive that

$$\begin{aligned}
\dot{\widetilde{E}}(\boldsymbol{\mu}_t, \Sigma_t) &= \frac{1}{2}\operatorname{tr}((Q^{-1} - \Sigma_t^{-1})\dot{\Sigma}_t) + (\boldsymbol{\mu}_t - \boldsymbol{b})^\top Q^{-1}\dot{\boldsymbol{\mu}}_t \\
&= -\operatorname{tr}\left((Q^{-1} - \Sigma_t^{-1})^2\Sigma_t^2\right) - 2\operatorname{tr}\left((Q^{-1} - \Sigma_t^{-1})\Sigma_t\boldsymbol{\mu}_t(\boldsymbol{\mu}_t - \boldsymbol{b})^\top Q^{-1}\right) \\
&\quad - (1 + \boldsymbol{\mu}_t^\top\boldsymbol{\mu}_t)(\boldsymbol{\mu}_t - \boldsymbol{b})^\top Q^{-2}(\boldsymbol{\mu}_t - \boldsymbol{b}) \\
&= -\operatorname{tr}\left(((Q^{-1} - \Sigma_t^{-1})\Sigma_t + Q^{-1}(\boldsymbol{\mu}_t - \boldsymbol{b})\boldsymbol{\mu}_t^\top)^\top ((Q^{-1} - \Sigma_t^{-1})\Sigma_t + Q^{-1}(\boldsymbol{\mu}_t - \boldsymbol{b})\boldsymbol{\mu}_t^\top)\right) \\
&\quad - (\boldsymbol{\mu}_t - \boldsymbol{b})^\top Q^{-2}(\boldsymbol{\mu}_t - \boldsymbol{b}) \leq 0.
\end{aligned}$$

Noticing that

$$0 \leq -\int_0^t \dot{\widetilde{E}}(\boldsymbol{\mu}_s, \Sigma_s)\,\mathrm{d}s = \widetilde{E}(\boldsymbol{\mu}_0, \Sigma_0) - \widetilde{E}(\boldsymbol{\mu}_t, \Sigma_t) < \infty,$$

we obtain that $\dot{\widetilde{E}}(\boldsymbol{\mu}_t, \Sigma_t) \to 0$ as $t \to \infty$, which is equivalent to $\boldsymbol{\mu}_t \to \boldsymbol{b}$ and $\Sigma_t \to Q$ by checking the expression above. Thus, we have shown that $\rho_t$ converges weakly to $\rho^*$ which is the density function of $\mathcal{N}(\boldsymbol{b}, Q)$.

Finally, we show the convergence rates of $\boldsymbol{\mu}_t$ and $\Sigma_t$. Since we have already proven that $\rho_t \to \rho^*$, it implies that

$$\boldsymbol{\mu}_t - \boldsymbol{b} = o(1), \quad \Sigma_t - Q = o(1), \quad \Sigma_t^{-1} - Q^{-1} = o(1).$$

If we set $\boldsymbol{\eta}_t = Q^{-1}(\boldsymbol{\mu}_t - \boldsymbol{b})$ and $S_t = (Q^{-1} - \Sigma_t^{-1})\Sigma_t$, then

$$
\begin{aligned}
&- \dot{\widetilde{E}}(\boldsymbol{\mu}_t, \Sigma_t) \\
&= \mathrm{tr}\left( \left((Q^{-1} - \Sigma_t^{-1})\Sigma_t + Q^{-1}(\boldsymbol{\mu}_t - \boldsymbol{b})\boldsymbol{\mu}_t^\top\right)^\top \left((Q^{-1} - \Sigma_t^{-1})\Sigma_t + Q^{-1}(\boldsymbol{\mu}_t - \boldsymbol{b})\boldsymbol{\mu}_t^\top\right)\right) \\
&\quad + (\boldsymbol{\mu}_t - \boldsymbol{b})^\top Q^{-2}(\boldsymbol{\mu}_t - \boldsymbol{b}) \\
&= \mathrm{tr}\left((S_t + \boldsymbol{\eta}_t\boldsymbol{\mu}_t^\top)^\top(S_t + \boldsymbol{\eta}_t\boldsymbol{\mu}_t^\top)\right) + \boldsymbol{\eta}_t^\top\boldsymbol{\eta}_t \\
&= \begin{bmatrix} \mathrm{vec}^\top(S_t) & \boldsymbol{\eta}_t^\top \end{bmatrix} \begin{bmatrix} I_{d^2} & \boldsymbol{\mu}_t \otimes I_d \\ \boldsymbol{\mu}_t^\top \otimes I_d & (1 + \boldsymbol{\mu}_t^\top \boldsymbol{\mu}_t)I_d \end{bmatrix} \begin{bmatrix} \mathrm{vec}(S_t) \\ \boldsymbol{\eta}_t \end{bmatrix} \\
&= \begin{bmatrix} \mathrm{vec}^\top(S_t) & \boldsymbol{\eta}_t^\top \end{bmatrix} \begin{bmatrix} I_{d^2} & \boldsymbol{b}_t \otimes I_d \\ \boldsymbol{b}_t^\top \otimes I_d & (1 + \boldsymbol{b}_t^\top \boldsymbol{b}_t)I_d \end{bmatrix} \begin{bmatrix} \mathrm{vec}(S_t) \\ \boldsymbol{\eta}_t \end{bmatrix} + o(\|S_t\| + \|\boldsymbol{\eta}_t\|).
\end{aligned}
$$

On the other hand, $\widetilde{E}(\boldsymbol{\mu}_t, \Sigma_t)$ can be written as

$$
\begin{aligned}
\widetilde{E}(\boldsymbol{\mu}_t, \Sigma_t) &= \frac{1}{2}\left(\mathrm{tr}(Q^{-1}\Sigma) - \log\det(Q^{-1}\Sigma) - d + (\boldsymbol{\mu} - \boldsymbol{b})^\top Q^{-1}(\boldsymbol{\mu} - \boldsymbol{b})\right) \\
&= \frac{1}{2}\left(\mathrm{tr}(S_t) - \log\det(I_d + S_t) + \boldsymbol{\eta}_t^\top Q\boldsymbol{\eta}_t\right) \\
&= \frac{1}{4}\mathrm{tr}(S_t^\top S_t) + \frac{1}{2}\boldsymbol{\eta}_t^\top Q\boldsymbol{\eta}_t + o(\|S_t\|^2) \\
&= \frac{1}{4}\begin{bmatrix} \mathrm{vec}^\top(S_t) & \boldsymbol{\eta}_t^\top \end{bmatrix} \begin{bmatrix} I_{d^2} & \\ & 2Q \end{bmatrix} \begin{bmatrix} \mathrm{vec}(S_t) \\ \boldsymbol{\eta}_t \end{bmatrix} + o(\|S_t\|^2)
\end{aligned}
$$

Now we prove that $\forall \epsilon > 0$ there exists $T > 0$ such that $-\dot{\widetilde{E}}(\boldsymbol{\mu}_t, \Sigma_t) \geq 4(\gamma - \epsilon)\widetilde{E}(\boldsymbol{\mu}_t, \Sigma_t)$ for $t \geq T$. It suffices to show that

$$
\begin{bmatrix} I_{d^2} & \boldsymbol{b}_t \otimes I_d \\ \boldsymbol{b}_t^\top \otimes I_d & (1 + \boldsymbol{b}_t^\top \boldsymbol{b}_t)I_d \end{bmatrix} \succeq \gamma \begin{bmatrix} I_{d^2} & \\ & 2Q \end{bmatrix},
$$

which is equivalent to

$$
\begin{bmatrix} I_{d^2} & \frac{1}{\sqrt{2}}\boldsymbol{b} \otimes Q^{-1/2} \\ \frac{1}{\sqrt{2}}\boldsymbol{b}^\top \otimes Q^{-1/2} & \frac{1}{2}(1 + \boldsymbol{b}^\top \boldsymbol{b})Q^{-1} \end{bmatrix} \succeq \gamma I_{d^2 + d}.
$$

This is true because by definition $\gamma$ is the smallest eigenvalue of the matrix.

By Grönwall's inequality, we know $\widetilde{E}(\boldsymbol{\mu}_t, \Sigma_t) = \mathcal{O}(e^{-4(\gamma - \epsilon)t})$. Thus, we conclude

$$
\|\boldsymbol{\mu}_t - \boldsymbol{b}\| = \mathcal{O}(e^{-2(\gamma - \epsilon)t}), \quad \|\Sigma_t - Q\| = \mathcal{O}(e^{-2(\gamma - \epsilon)t}), \quad \forall \epsilon > 0.
$$

Finally, we provide a lower bound on $\gamma$. Note that for any $u > 0$ if $\boldsymbol{b} \neq \boldsymbol{0}$ we have

$$
\begin{bmatrix} \frac{1}{1 + u}I_{d^2} & \frac{1}{\sqrt{2}}\boldsymbol{b} \otimes Q^{-1/2} \\ \frac{1}{\sqrt{2}}\boldsymbol{b}^\top \otimes Q^{-1/2} & \frac{1 + u}{2}\boldsymbol{b}^\top \boldsymbol{b}Q^{-1} \end{bmatrix} \succeq 0.
$$

Thus,

$$
\begin{bmatrix} I_{d^2} & \frac{1}{\sqrt{2}}\boldsymbol{b} \otimes Q^{-1/2} \\ \frac{1}{\sqrt{2}}\boldsymbol{b}^\top \otimes Q^{-1/2} & \frac{1}{2}(1 + \boldsymbol{b}^\top \boldsymbol{b})Q^{-1} \end{bmatrix} \succeq \begin{bmatrix} \frac{u}{1 + u}I_{d^2} & \\ & \frac{1}{2}(1 - u\boldsymbol{b}^\top \boldsymbol{b})Q^{-1} \end{bmatrix} =: \Omega_u.
$$

Since $\lambda_{\max}$ is the largest eigenvalue of $Q$, we know the smallest eigenvalue of $\Omega_u$ is given by

$$
\min\left\{\frac{u}{1 + u}, \frac{1 - u\boldsymbol{b}^\top \boldsymbol{b}}{2\lambda_{\max}}\right\}, \quad \text{where } u > 0.
$$

We find $u$ such that this quantity is maximized and get

$$\gamma \geq \max_{u>0} \min \left\{ \frac{u}{1+u}, \frac{1 - u\boldsymbol{b}^\top \boldsymbol{b}}{2\lambda_{\max}} \right\} = \frac{2}{1 + \boldsymbol{b}^\top \boldsymbol{b} + 2\lambda_{\max} + \sqrt{(1 + \boldsymbol{b}^\top \boldsymbol{b} + 2\lambda_{\max})^2 - 8\lambda_{\max}}}$$

$$> \frac{1}{1 + \boldsymbol{b}^\top \boldsymbol{b} + 2\lambda_{\max}}.$$

If $\boldsymbol{b} = 0$, then the smallest eigenvalue is given by

$$\gamma = \min \left\{ 1, \frac{1}{2\lambda_{\max}} \right\} > \frac{1}{1 + 2\lambda_{\max}}.$$

$\square$

*Proof of Theorem 3.2.* Equation (5) is a direct corollary of Theorem 3.1.

By Theorem C.2 there is a unique global solution. Thus, we only need to check that the $\Sigma_t$ given by (6) and (7) satisfy the algebraic Riccati equation (5).

For

$$\Sigma_t^{-1} = e^{-2t}\Sigma_0^{-1} + (1 - e^{-2t})Q^{-1}, \tag{77}$$

we take the derivative with respect to $t$ and get

$$-\Sigma_t^{-1}\dot{\Sigma}_t\Sigma_t^{-1} = 2e^{-2t}(Q^{-1} - \Sigma_0^{-1}). \tag{78}$$

Substituting (78) into (77), we get

$$2\Sigma_t^{-1} = 2Q^{-1} + \Sigma_t^{-1}\dot{\Sigma}_t\Sigma_t^{-1}.$$

Multiplying by $\Sigma_t^2$ and using the fact that $\Sigma_t$ and $Q$ commute, we can see that the algebraic Riccati equation (5) holds.

For

$$\Sigma_t = I + \frac{\eta(1 - e^{-2t})}{1 + \eta e^{-2t}}\boldsymbol{v}\boldsymbol{v}^\top,$$

we apply the Sherman–Morrison formula:

$$(\Sigma_t)^{-1} = I - \frac{\frac{\eta(1-e^{-2t})}{1+\eta e^{-2t}}}{1 + \frac{\eta(1-e^{-2t})}{1+\eta e^{-2t}}}\boldsymbol{v}\boldsymbol{v}^\top = I - \frac{\eta(1 - e^{-2t})}{1 + \eta}\boldsymbol{v}\boldsymbol{v}^\top.$$

Thus,

$$2\Sigma_t^{-1}(Q^{-1} - \Sigma_t^{-1})\Sigma_t = 2\left(I - \frac{\eta(1 - e^{-2t})}{1 + \eta}\boldsymbol{v}\boldsymbol{v}^\top\right)\left(-\frac{\eta e^{-2t}}{1 + \eta}\boldsymbol{v}\boldsymbol{v}^\top\right)\left(I + \frac{\eta(1 - e^{-2t})}{1 + \eta e^{-2t}}\boldsymbol{v}\boldsymbol{v}^\top\right)$$

$$= -\frac{2\eta e^{-2t}}{1 + \eta}\boldsymbol{v}\boldsymbol{v}^\top = \partial_t(\Sigma_t^{-1}).$$

Moreover,

$$(\Sigma_t^{-1} - Q^{-1})\Sigma_t = \frac{\eta e^{-2t}}{1 + \eta}\boldsymbol{v}\boldsymbol{v}^\top\left(I + \frac{\eta(1 - e^{-2t})}{1 + \eta e^{-2t}}\boldsymbol{v}\boldsymbol{v}^\top\right) = \frac{\eta e^{-2t}}{1 + \eta e^{-2t}}\boldsymbol{v}\boldsymbol{v}^\top.$$

Thus,

$$2\operatorname{tr}((\Sigma_t^{-1} - Q^{-1})\Sigma_t) = \frac{2\eta e^{-2t}}{1 + \eta e^{-2t}}\operatorname{tr}(\boldsymbol{v}\boldsymbol{v}^\top) = \frac{\eta e^{-2t}}{1 + \eta e^{-2t}}\operatorname{tr}(\boldsymbol{v}^\top \boldsymbol{v}) = \frac{\eta e^{-2t}}{1 + \eta e^{-2t}}.$$

On the other hand,

$$\det(\Sigma_t) = 1 + \frac{\eta(1 - e^{-2t})}{1 + \eta e^{-2t}}\boldsymbol{v}^\top \boldsymbol{v} = \frac{1 + \eta}{1 + \eta e^{-2t}}.$$

Thus,

$$\partial_t \log \det(\Sigma_t) = -\frac{-2\eta e^{-2t}}{1 + \eta e^{-2t}} = \frac{2\eta e^{-2t}}{1 + \eta e^{-2t}}.$$

Therefore,

$$\rho_t = (2\pi)^{-d/2} \big(\det(\Sigma_t)\big)^{-1/2} \exp\left(-\frac{1}{2}\boldsymbol{x}^\top \Sigma_t^{-1} \boldsymbol{x}\right)$$

with

$$\Sigma_t = I + \frac{\eta(1 - e^{-2t})}{1 + \eta e^{-2t}} \boldsymbol{v}\boldsymbol{v}^\top,$$

is a solution with the initial condition $\Sigma_0 = I_d$. The theorem follows by the uniqueness of the solution of the mean field PDE (Theorem C.2). $\qquad\square$

*Proof of Theorem 3.4.* Similar to the proof of Theorem 3.1, we have

$$\dot{\boldsymbol{\mu}}_t = -\nabla_{\boldsymbol{\mu}_t} \widetilde{E}(\boldsymbol{\mu}_t, \Sigma_t) = -Q^{-1}(\boldsymbol{\mu}_t - \boldsymbol{b})$$

and

$$\dot{\Sigma}_t = -2\Sigma_t^2((1 - \nu)\Sigma_t + \nu I)^{-1} \nabla_{\Sigma_t} \widetilde{E}(\Sigma_t) - 2\nabla_{\Sigma_t} \widetilde{E}(\Sigma_t)\Sigma_t^2((1 - \nu)\Sigma_t + \nu I)^{-1}$$
$$\Leftrightarrow \dot{\Sigma}_t = 2((1 - \nu)\Sigma_t + \nu I)^{-1}\Sigma_t - ((1 - \nu)\Sigma_t + \nu I)^{-1}\Sigma_t^2 Q^{-1} - Q^{-1}((1 - \nu)\Sigma_t + \nu I)^{-1}\Sigma_t^2.$$

Following the arguments similar to the proof of Theorem 3.1, we can show

- (8) has a unique global solution,

- $\rho_t$ is the density of $\mathcal{N}(\boldsymbol{\mu}_t, \Sigma_t)$ given by (8),

- $\dot{\widetilde{E}}(\theta_t) \leq 0$ and $\dot{\widetilde{E}}(\boldsymbol{\mu}_t, \Sigma_t) \to 0$ as $t \to \infty$,

- $\rho_t$ converges weakly to $\rho^*$.

Finally suppose $\Sigma_0 Q = Q\Sigma_0$, then $\Sigma_t$ also commutes with $Q$ since $0$ is a solution of the ODE satisfied by $\Sigma_t Q - Q\Sigma_t$ and the solution is unique. Thus, we can diagonalize $\Sigma_t$ and $Q$ simutaneously. Then there exists orthogonal matrix $P$ such that

$$\Sigma_t = P^\top \operatorname{diag}\left\{\sigma_1^{(t)}, \cdots, \sigma_d^{(t)}\right\} P, \quad Q = P^\top \operatorname{diag}\{\lambda_1, \cdots, \lambda_d\} P.$$

And (9) reduces to

$$\dot{\sigma}_i^{(t)} = \frac{2\sigma_i^{(t)}(\lambda_i - \sigma_i^{(t)})}{\lambda_i\left((1 - \nu)\sigma_i^{(t)} + \nu\right)}.$$

Solving this ODE, we get

$$\frac{(\sigma_i^{(t)} - \lambda_i)^{(1-\nu)\lambda_i + \nu}}{(\sigma_i^{(t)})^\nu} = \frac{(\sigma_i^{(0)} - \lambda_i)^{(1-\nu)\lambda_i + \nu}}{(\sigma_i^{(0)})^\nu} e^{-2t}.$$

Thus, we have $\sigma_i^{(t)} \to \lambda_i$ as $t \to \infty$ and

$$\left|\sigma_i^{(t)} - \lambda_i\right| = \mathcal{O}\left(e^{-2t/((1-\nu)\lambda_i + \nu)}\right).$$

In particular, we conclude $\|\Sigma_t - Q\| = \mathcal{O}\left(e^{-2t/((1-\nu)\lambda + \nu)}\right)$ where $\lambda$ is the largest eigenvalue of $Q$. $\qquad\square$

*Proof of Theorem 3.5.* First, we define the Hamiltonian on the centered Gaussian submanifold by

$$\mathcal{H}(\Sigma_t, S_t) := \frac{1}{2} \operatorname{tr}\big(S_t(G_{\Sigma_t}^{-1} S_t)\big) + E(\Sigma_t).$$

By Corollary B.5 we have

$$G_\Sigma^{-1} S = 2(\Sigma^2 S + S\Sigma^2).$$

Thus, the Hamiltonian is reduced to

$$\mathcal{H}(\Sigma_t, S_t) = 2\operatorname{tr}\big(\Sigma_t^2 S_t^2\big) + \frac{1}{2}\big(\operatorname{tr}(Q^{-1}\Sigma_t) - \log\det(Q^{-1}\Sigma_t) - d\big).$$

Therefore, we have

$$\nabla_{\Sigma_t}\mathcal{H}(\Sigma_t, S_t) = 2\left(\Sigma_t S_t^2 + S_t^2\Sigma_t\right) + \frac{1}{2}(Q^{-1} - \Sigma_t^{-1}), \quad \nabla_{S_t}\mathcal{H}(\Sigma_t, S_t) = 2\left(\Sigma_t^2 S_t + S_t\Sigma_t^2\right),$$

and thus the Hamiltonian or AIG flow on the Gaussian submanifold is given by (12).

Next we show $\Sigma_t$ is well-defined and remains positive definite. We check that $\mathcal{H}_t := \mathcal{H}(\Sigma_t, S_t)$ is decreasing with respect to $t$.

$$\begin{aligned}
\frac{\mathrm{d}\mathcal{H}_t}{\mathrm{d}t} &= \mathrm{tr}\left(\nabla_{S_t}\mathcal{H}_t\dot{S}_t + \nabla_{\Sigma_t}\mathcal{H}_t\dot{\Sigma}_t\right)\\
&= \mathrm{tr}\left(\nabla_{S_t}\mathcal{H}_t\left(-\alpha_t S_t - \nabla_{\Sigma_t}\mathcal{H}_t\right) + \nabla_{\Sigma_t}\mathcal{H}_t\nabla_{S_t}\mathcal{H}_t\right)\\
&= -4\alpha_t\,\mathrm{tr}\left(\Sigma_t^2 S_t^2\right) \le 0.
\end{aligned}$$

Let $\sigma_t$ be the smallest eigenvalue of $\Sigma_t$. Then

$$\log\det(\Sigma_t Q^{-1}) = \log\det\Sigma_t - \log\det Q \ge d\log\sigma_t - \log\det Q.$$

Therefore, we have

$$-\frac{d}{2}(\log\sigma_t + 1) + \frac{1}{2}\log\det Q \le -\frac{1}{2}\left(\log\det(\Sigma_t Q^{-1}) + d\right)$$
$$\le E(\Sigma_t) \le H_t \le H_0,$$

which yields that

$$\sigma_t \ge \exp\left(\log\det Q/d - 2H_0/d - 1\right).$$

This means that the smallest eigenvalue of $\Sigma_t$ has a positive lower bound. Thus, $\Sigma_t \in \mathrm{Sym}^+(d, \mathbb{R})$ for any $t \ge 0$.

Finally we show that the AIG flow on the centered Gaussian submanifold coincides with the one on the density manifold. Similar to the proof of Theorem 3.1, we have a commutative graph

$$\begin{array}{ccc}
T_\Sigma\Theta_0 & \xrightarrow{\mathrm{d}\phi_\Sigma} & T_\rho\mathcal{P}(\mathbb{R}^d)\\
G_\Sigma\downarrow & & \downarrow G_\rho\\
T_\Sigma^*\Theta_0 & \xrightarrow{\psi_\Sigma} & T_\rho^*\mathcal{P}(\mathbb{R}^d)
\end{array}.$$

Here $\psi_\Sigma : S \to \Phi(\boldsymbol{x}) = \boldsymbol{x}^\top S\boldsymbol{x} + C$, i.e., it maps a symmetric matrix to the quadratic function $\Phi(\boldsymbol{x})$ (or more precisely the equivalent classes of quadratic functions that differ by a constant). Now the only things we need to show are that $\frac{\delta\mathcal{H}}{\delta\rho_t} \in \mathrm{Im}\,\psi_\Sigma$ and that $\frac{\delta\mathcal{H}}{\delta\Phi_t} \in \mathrm{Im}\,\mathrm{d}\phi_\Sigma$.

$\frac{\delta\mathcal{H}}{\delta\Phi_t} = G_\rho^{-1}\Phi_t \in \mathrm{Im}\,\mathrm{d}\phi_\Sigma$ is trivially true from the commutative graph. Now since $\frac{\delta E}{\delta\rho_t} = \log\rho_t + V \in \mathrm{Im}\,\psi_\Sigma$ ($\rho_t$ is centered Gaussian density and check the definition of $\psi_\Sigma$), it suffices to prove

$$\frac{\delta}{\delta\rho_t}\int \Phi G_\rho^{-1}\Phi\,\mathrm{d}\boldsymbol{x} \in \mathrm{Im}\,\psi_\Sigma.$$

Note that we have

$$\begin{aligned}
\frac{\delta}{\delta\rho_t}\int \Phi_t G_\rho^{-1}\Phi_t\,\mathrm{d}\boldsymbol{x} &= -\frac{\delta}{\delta\rho_t}\int \Phi_t\nabla\cdot\left(\rho_t(\cdot)\int K(\cdot,\boldsymbol{y})\rho_t(\boldsymbol{y})\nabla\Phi_t(\boldsymbol{y})\,\mathrm{d}\boldsymbol{y}\right)\\
&= \frac{\delta}{\delta\rho_t}\int \nabla\Phi_t\cdot\left(\rho_t(\cdot)\int K(\cdot,\boldsymbol{y})\rho_t(\boldsymbol{y})\nabla\Phi(\boldsymbol{y})\,\mathrm{d}\boldsymbol{y}\right)\\
&= \nabla\Phi_t\cdot\int K(\cdot,\boldsymbol{y})\rho_t(\boldsymbol{y})\nabla\Phi(\boldsymbol{y})\,\mathrm{d}\boldsymbol{y}\\
&= (S_t\boldsymbol{x})^\top(S_t\Sigma_t\boldsymbol{x})\\
&= \frac{1}{2}\boldsymbol{x}^\top\left(S_t^2\Sigma_t + \Sigma_t S_t^2\right)\boldsymbol{x} \in \mathrm{Im}\,\psi_\Sigma.
\end{aligned}$$

Thus, the proof is complete. $\qquad\square$

We remark that the fact the Stein AIG flow remains Gaussian is highly non-trivial. In fact it requires $\frac{\delta \mathcal{H}}{\delta \rho_t}$ to lie in the cotangent space of the Gaussian submanifold. One sufficient condition is that the variational derivatives of both the kinetic and potential energies lie in this same space, and the former could be interpreted as: The Gaussian submanifold is totally geodesic under the given metric, meaning that any geodesic flow with an initial position and velocity chosen from the Gaussian submanifold remains Gaussian. Fortunately both the Wasserstein metric and the Stein metric satisfy this property.

## J  Proofs for Section 3.2

*Proof of Theorem 3.6.* From the proof of Theorem 3.1 we know that (16) has a unique solution that is continuous in $t$ and bounded for any $t \in [0, T]$ with $T < \infty$. Thus, the linear system (15) also has a unique solution.

Now we check that the sample mean $\boldsymbol{\mu}_t$ and covariance matrix $C_t$ satisfy (16). We simplify the right-han-side of (13) using $\boldsymbol{\mu}_t$ and $C_t$.

$$RHS = \boldsymbol{x}_i^{(t)} - \frac{1}{N} \sum_{j=1}^{N} \left( (\boldsymbol{x}_i^{(t)})^\top \boldsymbol{x}_j^{(t)} + 1 \right) Q^{-1} (\boldsymbol{x}_j^{(t)} - \boldsymbol{b})$$

$$= \boldsymbol{x}_i^{(t)} - \frac{1}{N} \sum_{j=1}^{N} Q^{-1} (\boldsymbol{x}_j^{(t)} - \boldsymbol{b}) \left( (\boldsymbol{x}_i^{(t)})^\top \boldsymbol{x}_j^{(t)} + 1 \right)$$

$$= \boldsymbol{x}_i^{(t)} - Q^{-1} \frac{1}{N} \sum_{j=1}^{N} \boldsymbol{x}_j^{(t)} \left( (\boldsymbol{x}_j^{(t)})^\top \boldsymbol{x}_i^{(t)} + 1 \right) + Q^{-1} \boldsymbol{b} \frac{1}{N} \sum_{j=1}^{N} \left( (\boldsymbol{x}_j^{(t)})^\top \boldsymbol{x}_i^{(t)} + 1 \right)$$

$$= (I - Q^{-1}(C_t + \boldsymbol{\mu}_t \boldsymbol{\mu}_t^\top) + Q^{-1} \boldsymbol{b} \boldsymbol{\mu}_t^\top) \boldsymbol{x}_i^{(t)} - Q^{-1} \boldsymbol{\mu}_t + Q^{-1} \boldsymbol{b}. \qquad (79)$$

Let $X_t = (\boldsymbol{x}_1^{(t)}, \cdots, \boldsymbol{x}_N^{(t)})^\top$. Then we have that

$$\boldsymbol{\mu}_t = \frac{X_t^\top \mathbf{1}}{N}, \quad C_t = \frac{X_t^\top X_t}{N} - \boldsymbol{\mu}_t \boldsymbol{\mu}_t^\top.$$

Then (79) can be written in the matrix form as

$$\dot{X}_t = X_t (I - (C_t + \boldsymbol{\mu}_t \boldsymbol{\mu}_t^\top) Q^{-1} + \boldsymbol{\mu}_t \boldsymbol{b}^\top Q^{-1}) - \mathbf{1} \boldsymbol{\mu}_t^\top Q^{-1} + \mathbf{1} \boldsymbol{b}^\top Q^{-1}. \qquad (80)$$

Multiplying by $\mathbf{1}^\top / N$ on the left, we get

$$\dot{\boldsymbol{\mu}}_t^\top = \boldsymbol{\mu}_t^\top - \boldsymbol{\mu}_t^\top (C_t + \boldsymbol{\mu}_t \boldsymbol{\mu}_t^\top) Q^{-1} + \boldsymbol{\mu}_t^\top \boldsymbol{\mu}_t \boldsymbol{b}^\top Q^{-1} - (\boldsymbol{\mu}_t - \boldsymbol{b})^\top Q^{-1}.$$

Thus, we have

$$\dot{\boldsymbol{\mu}}_t = (I - Q^{-1} C_t) \boldsymbol{\mu}_t - (1 + \boldsymbol{\mu}_t^\top \boldsymbol{\mu}_t) Q^{-1} (\boldsymbol{\mu}_t - \boldsymbol{b}).$$

Note that $\dot{C}_t = (\dot{X}_t^\top X_t + X_t^\top \dot{X}_t)/N - \boldsymbol{\mu}_t \dot{\boldsymbol{\mu}}_t - \dot{\boldsymbol{\mu}}_t \boldsymbol{\mu}_t^\top$. Substituting (80) we obtain

$$\dot{C}_t = 2C_t - C_t \left( C_t + \boldsymbol{\mu}_t (\boldsymbol{\mu}_t - \boldsymbol{b})^\top \right) Q^{-1} - Q^{-1} \left( C_t + (\boldsymbol{\mu}_t - \boldsymbol{b}) \boldsymbol{\mu}_t^\top \right) C_t.$$

Next we show that (14) and (15) satisfies (13).

$$RHS = (I - Q^{-1}(C_t + \boldsymbol{\mu}_t \boldsymbol{\mu}_t^\top) + Q^{-1} \boldsymbol{b} \boldsymbol{\mu}_t^\top) \boldsymbol{x}_i^{(t)} - Q^{-1} \boldsymbol{\mu}_t + Q^{-1} \boldsymbol{b}$$

$$= (I - Q^{-1}(C_t + \boldsymbol{\mu}_t \boldsymbol{\mu}_t^\top) + Q^{-1} \boldsymbol{b} \boldsymbol{\mu}_t^\top) \left( \boldsymbol{x}_i^{(t)} - \boldsymbol{\mu}_t \right) +$$

$$\quad (I - Q^{-1} C_t) \boldsymbol{\mu}_t - (1 + \boldsymbol{\mu}_t^\top \boldsymbol{\mu}_t) Q^{-1} (\boldsymbol{\mu}_t - \boldsymbol{b})$$

$$= (I - Q^{-1}(C_t + \boldsymbol{\mu}_t \boldsymbol{\mu}_t^\top) + Q^{-1} \boldsymbol{b} \boldsymbol{\mu}_t^\top) A_t (\boldsymbol{x}_i^{(0)} - \boldsymbol{\mu}_0) + \dot{\boldsymbol{\mu}}_t$$

$$= \dot{A}_t (\boldsymbol{x}_i^{(0)} - \boldsymbol{\mu}_0) + \dot{\boldsymbol{\mu}}_t = \dot{\boldsymbol{x}}_i^{(t)}.$$

By Theorem C.3 (13) has a unique solution. Thus, the solution of (13) is given by (14)–(16). $\qquad \square$

The proof of Theorem 3.7 is quite long and tedious so we defer it to Appendix L.

*Proof of Theorem 3.8.* If $C_0$ is non-singular, this is a direct corollary of Theorem 3.6 and Theorem 3.2. To also accommodate for the singular case, we provide a direct proof by calculation. Since $C_0 Q = Q C_0$, we know that $C_0$ and $Q$ are simultaneously diagonalizable. There exists some orthogonal matrix $P_0$ such that we have the spectral decompositions

$$C_0 = P_0^\top D_0 P_0, \quad Q = P_0^\top Q_0 P_0,$$

where $D_0 = \mathrm{diag}(\lambda_1^{(0)}, \cdots, \lambda_d^{(0)})$ and $Q_0 = \mathrm{diag}(q_1, \cdots, q_d)$ are diagonal matrices. Let $D_t := P_0 C_t P_0^\top$. (18) can be rewritten as

$$\boldsymbol{x}_i^{(t)} = P_0^\top \mathrm{diag}\big((e^{-2t} + (1 - e^{-2t})\lambda_1^{(0)}/q_1)^{-1/2}, \cdots, (e^{-2t} + (1 - e^{-2t})\lambda_d^{(0)}/q_d)^{-1/2}\big) P_0 \boldsymbol{x}_i^{(0)}.$$

Thus, by taking the derivative with respect to $t$, we obtain

$$\dot{\boldsymbol{x}}_i^{(t)} = e^{-2t} P_0^\top \mathrm{diag}\left(\frac{1 - \lambda_1^{(0)}/q_1}{(e^{-2t} + (1 - e^{-2t})\lambda_1^{(0)}/q_1)^{3/2}}, \cdots, \frac{1 - \lambda_d^{(0)}/q_d}{(e^{-2t} + (1 - e^{-2t})\lambda_d^{(0)}/q_d)^{3/2}}\right) P_0 \boldsymbol{x}_i^{(0)}$$

$$= U\big(e^{-2t} I + (1 - e^{-2t})Q^{-1} C_0\big)^{-3/2} \boldsymbol{x}_i^{(0)},$$

where

$$U = e^{-2t} P_0^\top \mathrm{diag}\big(1 - \lambda_1^{(0)}/q_1, \cdots, 1 - \lambda_d^{(0)}/q_d\big) P_0 = e^{-2t}(I - Q^{-1} C_0).$$

On the other hand, we check that

$$\boldsymbol{x}_i^{(t)} - \frac{1}{N} \sum_{j=1}^N \big((\boldsymbol{x}_i^{(t)})^\top \boldsymbol{x}_j^{(t)} + 1\big) Q^{-1} \boldsymbol{x}_j^{(t)}$$

$$= \big(e^{-2t} I + (1 - e^{-2t})Q^{-1} C_0\big)^{-1/2} \boldsymbol{x}_i^{(0)} - \frac{1}{N} \sum_{j=1}^N \big((\boldsymbol{x}_i^{(0)})^\top \big(e^{-2t} I + (1 - e^{-2t})Q^{-1} C_0\big)^{-1} \boldsymbol{x}_j^{(0)} + 1\big) \cdot$$

$$Q^{-1} \big(e^{-2t} I + (1 - e^{-2t})Q^{-1} C_0\big)^{-1/2} \boldsymbol{x}_j^{(0)}$$

$$= \big(e^{-2t} I + (1 - e^{-2t})Q^{-1} C_0\big)^{-1/2} \boldsymbol{x}_i^{(0)} - \frac{1}{N} \sum_{j=1}^N Q^{-1} \big(e^{-2t} I + (1 - e^{-2t})Q^{-1} C_0\big)^{-1/2} \boldsymbol{x}_j^{(0)}$$

$$- \frac{1}{N} \sum_{j=1}^N (\boldsymbol{x}_j^{(0)})^\top \big(e^{-2t} I + (1 - e^{-2t})Q^{-1} C_0\big)^{-1} \boldsymbol{x}_i^{(0)} \cdot$$

$$Q^{-1} \big(e^{-2t} I + (1 - e^{-2t})Q^{-1} C_0\big)^{-1/2} \boldsymbol{x}_j^{(0)}$$

$$= \big(e^{-2t} I + (1 - e^{-2t})Q^{-1} C_0\big)^{-1/2} \boldsymbol{x}_i^{(0)}$$

$$- \frac{1}{N} \sum_{j=1}^N Q^{-1} \big(e^{-2t} I + (1 - e^{-2t})Q^{-1} C_0\big)^{-1/2} \boldsymbol{x}_j^{(0)} (\boldsymbol{x}_j^{(0)})^\top \big(e^{-2t} I + (1 - e^{-2t})Q^{-1} C_0\big)^{-1} \boldsymbol{x}_i^{(0)}$$

$$= \big(e^{-2t} I + (1 - e^{-2t})Q^{-1} C_0\big)^{-1/2} \boldsymbol{x}_i^{(0)}$$

$$- Q^{-1} \big(e^{-2t} I + (1 - e^{-2t})Q^{-1} C_0\big)^{-1/2} C_0 \big(e^{-2t} I + (1 - e^{-2t})Q^{-1} C_0\big)^{-1} \boldsymbol{x}_i^{(0)}$$

$$= e^{-2t}(I - Q^{-1} C_0)\big(e^{-2t} I + (1 - e^{-2t})Q^{-1} C_0\big)^{-3/2} \boldsymbol{x}_i^{(0)}$$

$$= U\big(e^{-2t} I + (1 - e^{-2t})Q^{-1} C_0\big)^{-3/2} \boldsymbol{x}_i^{(0)}.$$

Thus, we conclude that (18) is a solution of (13). By Theorem C.3, the solution for (13) is unique and hence the theorem follows. □

*Proof of Theorem 3.9.* For any particle system of the R-SVGF, we can derive that

$$\dot{X}_t = \left((1 - \nu)\Big(\frac{X_t X_t^\top}{N} + \frac{\mathbf{1}\mathbf{1}^\top}{N}\Big) + \nu I_d\right)^{-1} \left(X_t - \frac{1}{N}(X_t X_t^\top + \mathbf{1}\mathbf{1}^\top)X_t Q^{-1}\right)$$

$$= \left((1 - \nu)\Big(\frac{X_t X_t^\top}{N} + \frac{\mathbf{1}\mathbf{1}^\top}{N}\Big) + \nu I_d\right)^{-1} X_t (I - C_t Q^{-1}).$$

By Sherman–Morrison formula we have

$$\left(\nu I_d + (1-\nu)\frac{\mathbf{11}^\top}{N}\right)^{-1} = \frac{1}{\nu}I_d - \frac{1-\nu}{\nu}\frac{\mathbf{11}^\top}{N}.$$

By Woodbury matrix identity we derive

$$\left((1-\nu)\left(\frac{X_t X_t^\top}{N} + \frac{\mathbf{11}^\top}{N}\right) + \nu I_d\right)^{-1}$$

$$=\left(\nu I_d + (1-\nu)\frac{\mathbf{11}^\top}{N}\right)^{-1} - \left(\nu I_d + (1-\nu)\frac{\mathbf{11}^\top}{N}\right)^{-1} X_t \cdot$$

$$\frac{1}{N}\left(\frac{1}{1-\nu}I_d + X_t^\top\left(\nu I_d + (1-\nu)\frac{\mathbf{11}^\top}{N}\right)^{-1}X_t\right)^{-1} X_t^\top\left(\nu I_d + (1-\nu)\frac{\mathbf{11}^\top}{N}\right)^{-1}$$

$$=\left(\frac{1}{\nu}I_d - \frac{1-\nu}{\nu}\frac{\mathbf{11}^\top}{N}\right) - \frac{1}{N\nu^2}X_t\left(\frac{1}{1-\nu}I_d + \frac{1}{\nu}C_t\right)^{-1}X_t^\top.$$

Substituting this into (81), we have

$$\dot{X}_t = \left(\frac{1}{\nu}I_d - \frac{1-\nu}{\nu}\frac{\mathbf{11}^\top}{N}\right)X_t(I - C_t Q^{-1}) - \frac{1}{N\nu^2}X_t\left(\frac{1}{1-\nu}I_d + \frac{1}{\nu}C_t\right)^{-1}X_t^\top X_t(I - C_t Q^{-1})$$

$$=\frac{1}{\nu}X_t - \frac{1}{\nu}X_t C_t Q^{-1} - \frac{1}{\nu}X_t\left(\frac{1}{1-\nu}I_d + \frac{1}{\nu}C_t\right)^{-1}\left(\frac{1}{\nu}C_t - \frac{1}{\nu}C_t^2 Q^{-1}\right)$$

$$=\frac{1}{\nu}X_t\left(I_d - \left(\frac{1}{1-\nu}I_d + \frac{1}{\nu}C_t\right)^{-1}\frac{1}{\nu}C_t\right)(I - C_t Q^{-1})$$

$$=X_t(\nu I_d + (1-\nu)C_t)^{-1}(I - C_t Q^{-1}) \tag{81}$$

Multiplying by $X_t^\top$ on the left we get

$$\frac{X_t^\top \dot{X}_t}{N} = (\nu I_d + (1-\nu)C_t)^{-1}C_t(I - C_t Q^{-1}).$$

Thus, the derivative of covariance matrix $C_t$ is given by

$$\dot{C}_t = \frac{X_t^\top \dot{X}_t}{N} + \frac{\dot{X}_t^\top X_t}{N}$$

$$=(\nu I_d + (1-\nu)C_t)^{-1}C_t(I - C_t Q^{-1}) + (I - Q^{-1}C_t)(\nu I_d + (1-\nu)C_t)^{-1}C_t$$

$$=2(\nu I_d + (1-\nu)C_t)^{-1}C_t - (\nu I_d + (1-\nu)C_t)^{-1}C_t^2 Q^{-1} - Q^{-1}(\nu I_d + (1-\nu)C_t)^{-1}C_t^2.$$

Next we show that (19) and (20) satisfies (81).

$$RHS = X_0 A_t^\top (\nu I_d + (1-\nu)C_t)^{-1}(I - C_t Q^{-1}) = X_0 \dot{A}_t^\top = \dot{X}_t.$$

Similar to the proof of Theorem 3.4, it could be shown that the R-SVGF also has a unique solution and the proof is complete. $\square$

Next we show Theorem 3.10.

**Lemma J.1.** *The covariance matrix $C_t$ ($t = 1, 2, \cdots$) of the discrete-time finite particle system satisfies the following equation*

$$C_{t+1} = \left(I + \epsilon(I - Q^{-1}C_t)\right)C_t\left(I + \epsilon(I - Q^{-1}C_t)\right)^\top.$$

*Proof.* Let $X = (\boldsymbol{x}_1, \cdots, \boldsymbol{x}_N)^\top$. Then (21) can be written as

$$X_{t+1} = X_t + \epsilon X_t(I - C_t Q^{-1}).$$

Thus, we have

$$C_{t+1} = \frac{X_t^\top X_t}{N} = \left(I + \epsilon(I - Q^{-1}C_t)\right)C_t\left(I + \epsilon(I - Q^{-1}C_t)\right)^\top.$$

$\square$

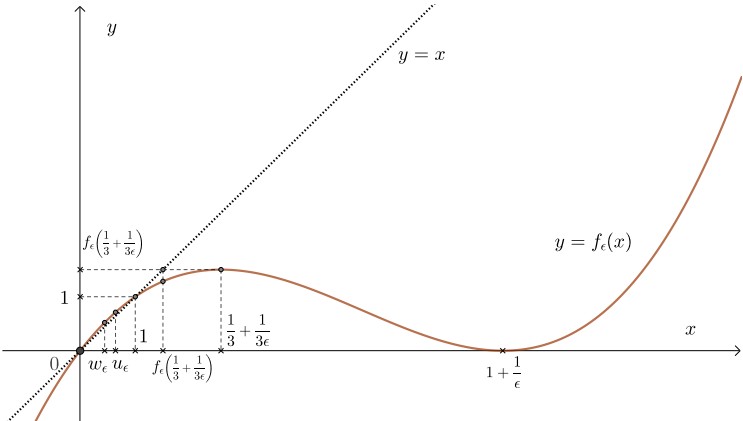

Figure 6: Plot of the function $f_\epsilon(x) = (1 + \epsilon(1 - x))^2 x$.

*Proof of Theorem 3.10.* First since $C_0 Q = Q C_0$ they are simultaneously diagonalizable. By Theorem 3.8 any $C_t$ and $Q$ are simultaneously diagonalizable. Thus, without loss of generality we assume all $C_t$ and $Q$ are diagonal matrices. Then by Lemma J.1 we know

$$Q^{-1} C_{t+1} = (I + \epsilon(I - Q^{-1} C_t))^2 Q^{-1} C_t.$$

If we define $f_\epsilon(x) = (1 + \epsilon(1 - x))^2 x$ then every entry (i.e., eigenvalue) of $Q^{-1} C_t$ follows the $f_\epsilon$-iteration trajectory of the corresponding entry of $Q^{-1} C_0$.

The fixed points of $f_\epsilon(x) = (1 + \epsilon(1 - x))^2 x$ are $\{0, 1, 2/\epsilon + 1\}$. If $0 < \epsilon < 1$ then $|f'(0)| > 1$, $|f'(2/\epsilon + 1)| > 1$, and $|f'(1)| < 1$. By Proposition 1.9 of [26], 1 is an attracting fixed point while 0 and $2/\epsilon + 1$ are repelling fixed points. By definition there exists an interval around 1 such that for all initial points in that interval, the trajectory of any eigenvalue of $Q^{-1} C_t$ converges to 1. We now quantify that result further.

Note that $f_\epsilon'(x) = (1 + \epsilon - \epsilon x)(1 + \epsilon - 3\epsilon x)$ is an upward-opening parabola whose zeros are $1/3 + 1/(3\epsilon)$ and $1 + \epsilon$. Thus, $f_\epsilon'$ is monotone decreasing on $[0, 1/3 + 1/(3\epsilon)]$. In particular, both equation $f_\epsilon'(x) = 1$ and $f_\epsilon'(x) = 1 - \epsilon$ have two distinct roots, the smaller ones lying on $(0, 1/3 + 1/(3\epsilon))$. Define $w_\epsilon$ to be the smaller root of $f_\epsilon'(x) = 1$ and $u_\epsilon$, the smaller root of $f_\epsilon'(x) = 1 - \epsilon$. Since $f_\epsilon'$ is monotone descreasing and $f_\epsilon'(1) = 1 - 2\epsilon$, we have $0 < w_\epsilon < u_\epsilon < 1$.

Thus, for any $x \in [w_\epsilon, 1/3 + 1/(3\epsilon)]$ we have $0 \leq f_\epsilon'(x) \leq 1$ and $0 \leq f_\epsilon(x) \leq f_\epsilon(1/3 + 1/(3\epsilon)) < 1/3 + 1/(3\epsilon)$ (here we have used the condition $\epsilon < 0.5$). On the other hand, since $f_\epsilon'(0) \geq 1$ and 0 and 1 are two fixed points, it holds that $f_\epsilon(x) \geq x$ for any $x \in [0, 1]$, which implies that for any $x \in [w_\epsilon, 1/3 + 1/(3\epsilon)]$ we have $f_\epsilon(x) \geq f_\epsilon(w_\epsilon) \geq w_\epsilon$. Hence we know that $f_\epsilon'$ is a contraction map and $f_\epsilon([w_\epsilon, 1/3 + 1/(3\epsilon)]) \subset [w_\epsilon, 1/3 + 1/(3\epsilon)]$, which implies that there is a unique fixed point (i.e., 1) such that the $f_\epsilon$-iteration trajectory converges to it.

Next we prove that for any $x \in (0, 1 + 1/\epsilon)$ the trajectory falls into the interval $[w_\epsilon, 1/3 + 1/(3\epsilon)]$ after finite iterations. If $x \in (1/3 + 1/(3\epsilon), 1 + 1/\epsilon)$ then after one iteration we get $f_\epsilon(x) \in (0, f_\epsilon(1/3 + 1/(3\epsilon))) \subset (0, 1/3 + 1/(3\epsilon))$. We claim that if $x \in (0, w_\epsilon)$ then after $t_0 := \left\lceil \dfrac{\log w_\epsilon / x}{2 \log(1 + \epsilon(1 - w_\epsilon))} \right\rceil$ we have $f_\epsilon^{(t_0)}(x) \in [w_\epsilon, 1]$. Firstly $f_\epsilon^{(t)}(x) \leq 1$ for any $t$. It suffices to prove $f_\epsilon^{(t_0)}(x) \geq w_\epsilon$. Suppose $f_\epsilon^{(t_0)}(x) < w_\epsilon$. Then for any $0 \leq t \leq t_0$ we have $f_\epsilon^{(t)}(x) < w_\epsilon$. By definition of $t_0$ we know

$$f_\epsilon^{(t_0)}(x) > (1 + \epsilon(1 - w_\epsilon))^{2t_0} x \geq w_\epsilon.$$

This is a contradiction. Thus, the claim holds and in conclusion for any $x \in (0, 1 + 1/\epsilon)$ the $f_\epsilon$-iteration trajectory converges to the fixed point 1.

Finally we consider the case when $x \in [u_\epsilon, 1/3 + 1/(3\epsilon)]$. Since $f_\epsilon'$ is monotone descreasing here we have $f_\epsilon'(x) \in [0, 1 - \epsilon]$. And by similar argument we know $f_\epsilon([u_\epsilon, 1/3 + 1/(3\epsilon)]) \subset [u_\epsilon, 1/3 + 1/(3\epsilon)]$. Thus, we conclude

$$|f_\epsilon^{(t)}(x) - 1| \leq (1 - \epsilon)|f_\epsilon^{(t-1)}(x) - 1| \leq \cdots \leq (1 - \epsilon)^t |x - 1| \leq e^{-\epsilon t}|x - 1|.$$

$\square$

## K  Proofs for Section 4

The following result was derived in [43]. We provide the proof below for completeness.

**Lemma K.1.** *Let $\rho^*$ be a probability measure and $\rho_\theta$ be a Gaussian measure with parameters $\theta = (\boldsymbol{\mu}, \Sigma)$. Then we have the following expressions:*

$$\nabla_{\boldsymbol{\mu}} \mathrm{KL}(\rho_\theta \parallel \rho^*) = \mathbb{E}_{\boldsymbol{x} \sim \rho_\theta}[\nabla V(\boldsymbol{x})], \quad \nabla_\Sigma \mathrm{KL}(\rho_\theta \parallel \rho^*) = \frac{1}{2}\left(\mathbb{E}_{\boldsymbol{x} \sim \rho_\theta}[\nabla^2 V(\boldsymbol{x})] - \Sigma^{-1}\right). \quad (82)$$

*Proof.* We compute that

$$\begin{aligned}
\nabla_{\boldsymbol{\mu}} \mathrm{KL}(\rho_\theta \parallel \rho^*) &= \nabla_{\boldsymbol{\mu}} \int \log \frac{\rho_\theta(\boldsymbol{x})}{\rho^*(\boldsymbol{x})} \rho_\theta(\boldsymbol{x}) \, \mathrm{d}\boldsymbol{x} \\
&= \int \frac{\nabla_{\boldsymbol{\mu}}\rho_\theta(\boldsymbol{x})}{\rho_\theta(\boldsymbol{x})} \rho_\theta(\boldsymbol{x}) \, \mathrm{d}\boldsymbol{x} + \int \log \frac{\rho_\theta(\boldsymbol{x})}{\rho^*(\boldsymbol{x})} \nabla_{\boldsymbol{\mu}}\rho_\theta(\boldsymbol{x}) \, \mathrm{d}\boldsymbol{x} \\
&\overset{*}{=} \nabla_{\boldsymbol{\mu}} \int \rho_\theta(\boldsymbol{x}) \, \mathrm{d}\boldsymbol{x} - \int \log \frac{\rho_\theta(\boldsymbol{x})}{\rho^*(\boldsymbol{x})} \nabla_{\boldsymbol{x}}\rho_\theta(\boldsymbol{x}) \, \mathrm{d}\boldsymbol{x} \\
&= \int \nabla_{\boldsymbol{x}} \log \frac{\rho_\theta(\boldsymbol{x})}{\rho^*(\boldsymbol{x})} \rho_\theta(\boldsymbol{x}) \, \mathrm{d}\boldsymbol{x} \\
&= \int \nabla_{\boldsymbol{x}}\rho_\theta(\boldsymbol{x}) \, \mathrm{d}\boldsymbol{x} - \int \nabla_{\boldsymbol{x}} \log \rho^*(\boldsymbol{x}) \cdot \rho_\theta(\boldsymbol{x}) \, \mathrm{d}\boldsymbol{x} \\
&= \mathbb{E}_{\boldsymbol{x} \sim \rho_\theta}[\nabla V(\boldsymbol{x})],
\end{aligned}$$

where we have used the fact that $\rho_\theta$ is a Gaussian density in $*$. Similarly, we have

$$\begin{aligned}
\nabla_\Sigma \mathrm{KL}(\rho_\theta \parallel \rho^*) &= \nabla_\Sigma \int \log \frac{\rho_\theta(\boldsymbol{x})}{\rho^*(\boldsymbol{x})} \rho_\theta(\boldsymbol{x}) \, \mathrm{d}\boldsymbol{x} \\
&= \int \frac{\nabla_\Sigma \rho_\theta(\boldsymbol{x})}{\rho_\theta(\boldsymbol{x})} \rho_\theta(\boldsymbol{x}) \, \mathrm{d}\boldsymbol{x} + \int \log \frac{\rho_\theta(\boldsymbol{x})}{\rho^*(\boldsymbol{x})} \nabla_\Sigma \rho_\theta(\boldsymbol{x}) \, \mathrm{d}\boldsymbol{x} \\
&\overset{(a)}{=} \int \log \frac{\rho_\theta(\boldsymbol{x})}{\rho^*(\boldsymbol{x})} \left(-\frac{1}{2}\Sigma^{-1} + \frac{1}{2}\Sigma^{-1}(\boldsymbol{x} - \boldsymbol{\mu})(\boldsymbol{x} - \boldsymbol{\mu})^\top \Sigma^{-1}\right) \rho_\theta(\boldsymbol{x}) \, \mathrm{d}\boldsymbol{x} \\
&\overset{(b)}{=} \frac{1}{2} \int \log \frac{\rho_\theta(\boldsymbol{x})}{\rho^*(\boldsymbol{x})} \nabla^2_{\boldsymbol{x}} \rho_\theta(\boldsymbol{x}) \, \mathrm{d}\boldsymbol{x} \\
&= \frac{1}{2} \int \nabla^2_{\boldsymbol{x}} \log \frac{\rho_\theta(\boldsymbol{x})}{\rho^*(\boldsymbol{x})} \cdot \rho_\theta(\boldsymbol{x}) \, \mathrm{d}\boldsymbol{x} \\
&= \frac{1}{2}\left(\mathbb{E}_{\boldsymbol{x} \sim \rho_\theta}[\nabla^2 V(\boldsymbol{x})] - \Sigma^{-1}\right).
\end{aligned}$$

Here again in $(a)$ and $(b)$ we have used the closed-form expression of Gaussian densities. Thus,

$$\nabla_{\boldsymbol{\mu}} E(\boldsymbol{\mu}, \Sigma) = \mathbb{E}_{\boldsymbol{x} \sim \rho_\theta}[\nabla V(\boldsymbol{x})], \quad \nabla_\Sigma E(\boldsymbol{\mu}, \Sigma) = \frac{1}{2}\left(\mathbb{E}_{\boldsymbol{x} \sim \rho_\theta}[\nabla^2 V(\boldsymbol{x})] - \Sigma^{-1}\right).$$

$\square$

*Proof of Theorem 4.1.* By definition of Gaussian approximate gradient descent, we have that

$$\dot{\theta}_t = -G_{\theta_t}^{-1}(\nabla_{\theta_t} E(\theta_t)), \text{ where } E(\theta_t) = \mathrm{KL}(\rho_{\theta_t} \parallel \rho^*) \text{ and } \theta = (\boldsymbol{\mu}, \Sigma).$$

Applying Theorem B.3, we obtain that $\dot{\theta}_t = -G_{\theta_t}^{-1}(\nabla_{\theta_t} E(\theta_t))$ is given by

$$\begin{cases}
\dot{\boldsymbol{\mu}}_t = -\left(2\nabla_{\Sigma_t} E(\boldsymbol{\mu}_t, \Sigma_t)\Sigma_t \boldsymbol{\mu}_t + (1 + \boldsymbol{\mu}_t^\top \boldsymbol{\mu}_t)\nabla_{\boldsymbol{\mu}_t} E(\boldsymbol{\mu}_t, \Sigma_t)\right) \\
\dot{\Sigma}_t = -\Sigma_t\left(2\Sigma_t \nabla_{\Sigma_t} E(\boldsymbol{\mu}_t, \Sigma_t) + \boldsymbol{\mu}_t \nabla_{\boldsymbol{\mu}_t}^\top E(\boldsymbol{\mu}_t, \Sigma_t)\right) - \left(2\nabla_{\Sigma_t} E(\boldsymbol{\mu}_t, \Sigma_t)\Sigma_t + \nabla_{\boldsymbol{\mu}_t} E(\boldsymbol{\mu}_t, \Sigma_t)\boldsymbol{\mu}_t^\top\right)\Sigma_t
\end{cases}.$$

$$(83)$$

Substituting (82) into (83), we get

$$\begin{cases} \dot{\boldsymbol{\mu}}_t = \left(I - \mathbb{E}_{\boldsymbol{x}\sim\rho_t}[\nabla^2 V(\boldsymbol{x})]\Sigma_t\right)\boldsymbol{\mu}_t - (1 + \boldsymbol{\mu}_t^\top\boldsymbol{\mu}_t)\mathbb{E}_{\boldsymbol{x}\sim\rho_t}[\nabla V(\boldsymbol{x})] \\ \dot{\Sigma}_t = 2\Sigma_t - \Sigma_t\left(\Sigma_t\mathbb{E}_{\boldsymbol{x}\sim\rho_t}[\nabla^2 V(\boldsymbol{x})] + \boldsymbol{\mu}_t\mathbb{E}_{\boldsymbol{x}\sim\rho_t}[\nabla^\top V(\boldsymbol{x})]\right) \\ \qquad - \left(\mathbb{E}_{\boldsymbol{x}\sim\rho_t}[\nabla^2 V(\boldsymbol{x})]\Sigma_t + \mathbb{E}_{\boldsymbol{x}\sim\rho_t}[\nabla^\top V(\boldsymbol{x})]\boldsymbol{\mu}_t\right)\Sigma_t \end{cases} .$$

Next we prove the convergence.

$$\begin{aligned} \dot{E}(\boldsymbol{\mu}_t, \Sigma_t) &= \mathrm{tr}(\nabla_{\Sigma_t} E(\boldsymbol{\mu}_t, \Sigma_t)^\top \dot{\Sigma}_t) + \nabla_{\boldsymbol{\mu}_t} E(\boldsymbol{\mu}_t, \Sigma_t)^\top \dot{\boldsymbol{\mu}}_t \\ &= -\mathrm{tr}\left((2\nabla_\Sigma E(\boldsymbol{\mu}, \Sigma)\Sigma + \nabla_{\boldsymbol{\mu}} E(\boldsymbol{\mu}, \Sigma)\boldsymbol{\mu}^\top)^\top (2\nabla_\Sigma E(\boldsymbol{\mu}, \Sigma)\Sigma + \nabla_{\boldsymbol{\mu}} E(\boldsymbol{\mu}, \Sigma)\boldsymbol{\mu}^\top)\right) \\ &\quad - \nabla_{\boldsymbol{\mu}}^\top E(\boldsymbol{\mu}, \Sigma)\nabla_{\boldsymbol{\mu}} E(\boldsymbol{\mu}, \Sigma) \le 0. \end{aligned}$$

Noticing that

$$0 \le -\int_0^t \dot{E}(\boldsymbol{\mu}_s, \Sigma_s)\,\mathrm{d}s = E(\boldsymbol{\mu}_0, \Sigma_0) - E(\boldsymbol{\mu}_t, \Sigma_t) < \infty,$$

we obtain that $\dot{E}(\boldsymbol{\mu}_t, \Sigma_t) \to 0$ as $t \to \infty$, which means that there exists $\boldsymbol{\mu}_\infty, \Sigma_\infty$ such that $\rho_t$ converges to $\rho_\infty$, $\rho_\infty$ is the density of $\mathcal{N}(\boldsymbol{\mu}_\infty, \Sigma_\infty)$. (Since $E(\boldsymbol{\mu}_t, \Sigma_t)$ is given by the KL divergence between $\rho_t$ and $\rho^*$, by Lemma K.1 it will diverge if $\boldsymbol{\mu}_t$ or $\Sigma_t$ diverges.) In particular, it satisfies that

$$\begin{cases} 2\nabla_\Sigma E(\boldsymbol{\mu}_\infty, \Sigma_\infty) + \nabla_{\boldsymbol{\mu}} E(\boldsymbol{\mu}_\infty, \Sigma_\infty)\boldsymbol{\mu}_\infty^\top = 0 \\ \nabla_{\boldsymbol{\mu}} E(\boldsymbol{\mu}_\infty, \Sigma_\infty) = \mathbf{0} \end{cases},$$

which is equivalent to

$$\begin{cases} \nabla_\Sigma E(\boldsymbol{\mu}_\infty, \Sigma_\infty) = 0 \\ \nabla_{\boldsymbol{\mu}} E(\boldsymbol{\mu}_\infty, \Sigma_\infty) = \mathbf{0} \end{cases},$$

and implies that $\rho_\infty = \rho_{\theta^*}$. $\qquad\square$

**Lemma K.2.** *Given $\alpha$-strongly convex measure $\rho^*$, we define $\theta^*$ as the unique minimizer of $\mathrm{KL}(\rho_\theta \parallel \rho^*)$ where $\rho_\theta$ denotes a Gaussian measure with parameters $\theta$. Then it holds that*

$$\|\mathbb{E}_{\rho_\theta}[\nabla V(\boldsymbol{x})]\|_2^2 + \mathrm{tr}\left((\mathbb{E}_{\rho_\theta}[\nabla^2 V(\boldsymbol{x})] - \Sigma^{-1})\Sigma(\mathbb{E}_{\rho_\theta}[\nabla^2 V(\boldsymbol{x})] - \Sigma^{-1})\right)$$
$$\ge 2\alpha \left(\mathrm{KL}(\rho_\theta \parallel \rho^*) - \mathrm{KL}(\rho_{\theta^*} \parallel \rho^*)\right).$$

This is proven in the Appendix D of [43]. The proof idea is to consider the Gaussian approximate Wasserstein gradient flow from $\rho_\theta$ with the target $\rho^*$.

**Lemma K.3.** *Given $\alpha$-strongly convex measure $\rho^*$, we define $\theta^*$ as the unique minimizer of $\mathrm{KL}(\rho_\theta \parallel \rho^*)$ where $\rho_\theta$ denotes a Gaussian measure with parameters $\theta$. Then the Wasserstein-2 distance between $\rho_\theta$ and $\rho_{\theta^*}$ satisfies that*

$$\alpha \mathcal{W}_2^2(\rho_\theta, \rho_{\theta^*}) \le \mathrm{KL}(\rho_\theta \parallel \rho^*) - \mathrm{KL}(\rho_{\theta^*} \parallel \rho^*).$$

This is Lemma E.2 of [13].

*Proof of Theorem 4.2.* From Lemma K.1 and the proof of Theorem 4.1 we know that

$$\begin{aligned} \dot{E}(\boldsymbol{\mu}_t, \Sigma_t) &= -\mathrm{tr}\left((2\nabla_\Sigma E(\boldsymbol{\mu}, \Sigma)\Sigma + \nabla_{\boldsymbol{\mu}} E(\boldsymbol{\mu}, \Sigma)\boldsymbol{\mu}^\top)^\top (2\nabla_\Sigma E(\boldsymbol{\mu}, \Sigma)\Sigma + \nabla_{\boldsymbol{\mu}} E(\boldsymbol{\mu}, \Sigma)\boldsymbol{\mu}^\top)\right) \\ &\quad - \nabla_{\boldsymbol{\mu}}^\top E(\boldsymbol{\mu}, \Sigma)\nabla_{\boldsymbol{\mu}} E(\boldsymbol{\mu}, \Sigma) \\ &= -\mathrm{tr}\left((\mathbb{E}_{\rho_\theta}[\nabla^2 V(\boldsymbol{x})]\Sigma - I + \mathbb{E}_{\rho_\theta}[\nabla V(\boldsymbol{x})]\boldsymbol{\mu}^\top)^\top (\mathbb{E}_{\rho_\theta}[\nabla^2 V(\boldsymbol{x})]\Sigma - I + \mathbb{E}_{\rho_\theta}[\nabla V(\boldsymbol{x})]\boldsymbol{\mu}^\top)\right) \\ &\quad - \|\mathbb{E}_{\rho_\theta}[\nabla V(\boldsymbol{x})]\|_2^2. \end{aligned}$$

For convenience we let $\boldsymbol{\eta}_t = \mathbb{E}_{\rho_\theta}[\nabla V(\boldsymbol{x})]$ and $S_t = (\mathbb{E}_{\rho_\theta}[\nabla^2 V(\boldsymbol{x})] - \Sigma^{-1})\Sigma^{1/2}$. Then we get

$$\begin{aligned} -\dot{E}(\boldsymbol{\mu}_t, \Sigma_t) &= \begin{bmatrix} \mathrm{vec}^\top(S_t) & \boldsymbol{\eta}_t \end{bmatrix} \begin{bmatrix} I_d \otimes \Sigma_t & \boldsymbol{\mu}_t \otimes \Sigma_t^{1/2} \\ \boldsymbol{\mu}_t^\top \otimes \Sigma_t^{1/2} & (1 + \boldsymbol{\mu}_t^\top\boldsymbol{\mu}_t)I_d \end{bmatrix} \begin{bmatrix} \mathrm{vec}(S_t) \\ \boldsymbol{\eta}_t \end{bmatrix} \\ &=: \begin{bmatrix} \mathrm{vec}^\top(S_t) & \boldsymbol{\eta}_t \end{bmatrix} M_t \begin{bmatrix} \mathrm{vec}(S_t) \\ \boldsymbol{\eta}_t \end{bmatrix}. \end{aligned}$$

Noting that $M_t \to M_\infty$, for any $\epsilon > 0$ there exists $T > 0$ such that the smallest eigenvalue $\gamma_t/\alpha$ of $M_t$ satisfies $\gamma_t \geq \gamma - \epsilon$ for any $t > T$, where $\gamma/\alpha$ is the smallest eigenvalue of $M_\infty$.

Moreover, by Lemma K.2 we have

$$\begin{bmatrix} \text{vec}^\top(S_t) & \boldsymbol{\eta}_t \end{bmatrix} \begin{bmatrix} \text{vec}(S_t) \\ \boldsymbol{\eta}_t \end{bmatrix} \geq 2\alpha(\text{KL}(\rho_\theta \parallel \rho^*) - \text{KL}(\rho_\infty \parallel \rho^*)).$$

Thus, for $t > T$ the derivative of KL divergence is controlled, i.e.,

$$-\partial_t \text{KL}(\rho_t \parallel \rho^*) \geq 2(\gamma - \epsilon)(\text{KL}(\rho_\theta \parallel \rho^*) - \text{KL}(\rho_\infty \parallel \rho^*)).$$

By Grönwall's inequality we know

$$\text{KL}(\rho_\theta \parallel \rho^*) - \text{KL}(\rho_\infty \parallel \rho^*) = \mathcal{O}(e^{-2(\gamma - \epsilon)t}).$$

By Lemma K.3 this implies

$$\mathcal{W}_2^2(\rho_\theta, \rho_{\theta^*}) = \mathcal{O}(e^{-2(\gamma - \epsilon)t}).$$

Noting that

$$\mathcal{W}_2^2(\rho_\theta, \rho_{\theta^*}) = \|\boldsymbol{\mu} - \boldsymbol{\mu}^*\|_2^2 + \text{tr}\left((\Sigma^{1/2} - (\Sigma^*)^{1/2})^2\right),$$

we conclude

$$\|\boldsymbol{\mu}_t - \boldsymbol{\mu}^*\| = \mathcal{O}(e^{-(\gamma - \epsilon)t}), \quad \|\Sigma_t - \Sigma^*\| = \mathcal{O}(e^{-(\gamma - \epsilon)t}), \quad \forall \epsilon > 0.$$

Finally, we provide a lower bound on $\gamma$. Note that for any $u > 0$ if $\boldsymbol{\mu}^* \neq \mathbf{0}$ we have

$$\begin{bmatrix} \dfrac{1}{1+u} I_d \otimes \Sigma^* & \boldsymbol{\mu}^* \otimes (\Sigma^*)^{1/2} \\ \boldsymbol{\mu}^{*\top} \otimes (\Sigma^*)^{1/2} & (1+u)\boldsymbol{\mu}^{*\top}\boldsymbol{\mu}^* I_d \end{bmatrix} \succeq 0.$$

Thus,

$$\begin{bmatrix} I_d \otimes \Sigma^* & \boldsymbol{\mu}^* \otimes (\Sigma^*)^{1/2} \\ \boldsymbol{\mu}^{*\top} \otimes (\Sigma^*)^{1/2} & (1 + \boldsymbol{\mu}^{*\top}\boldsymbol{\mu}^*)I_d \end{bmatrix} \succeq \begin{bmatrix} \frac{u}{1+u} I_d \otimes \Sigma^* & \\ & (1 - u\boldsymbol{\mu}^{*\top}\boldsymbol{\mu}^*)I_d \end{bmatrix} =: \Omega_u.$$

Since $\Sigma^*$ satisfies that

$$\mathbb{E}_{\rho_{\theta^*}}[\nabla^2 V(\boldsymbol{x})] - (\Sigma^*)^{-1} = 0,$$

and $\nabla^2 V(\boldsymbol{x}) \preceq \beta I_d$, we know the smallest eigenvalue of $\Sigma^*$ is at least $1/\beta$. Thus, the smallest eigenvalue of $\Omega_u$ is

$$\min\left\{\frac{u}{\beta(1+u)}, 1 - ur\right\}, \quad \text{where } u > 0, r = \boldsymbol{\mu}^{*\top}\boldsymbol{\mu}^*.$$

We find $u$ such that this quantity is maximized and get

$$\frac{\gamma}{\alpha} \geq \max_{u>0} \min\left\{\frac{u}{\beta(1+u)}, 1 - ur\right\} = \frac{2}{1 + \beta(1+r) + \sqrt{(1 + \beta(1+r))^2 - 4\beta}} > \frac{1}{\beta(1+r) + 1}.$$

If $\boldsymbol{\mu}^* = 0$, we still have

$$\frac{\gamma}{\alpha} = \min\left\{\frac{1}{\beta}, 1\right\} > \frac{1}{\beta + 1}.$$

$\square$

## L   Proofs of Uniform in Time Convergence

To show Theorem 3.7 we need a lemma on the convergence of empirical measures in the *i.i.d.* setting. There are results for general measures on this given by [25, 47, 24] but for our purpose we always have a Gaussian distributions as the limit and there are tight results with faster convergence rates as shown in [9, 45, 46]:

**Lemma L.1** (Convergence of empirical measures for Gaussian distributions). *Fix the dimension $d \geq 1$. There exists a constant $C_d$ such that for all $N \geq 1$, with $\mu_N = \frac{1}{N} \sum_{k=1}^{N} \delta_{X_k}$ where $\{X_k\}$ is i.i.d. sequence drawn from $\mu \sim \mathcal{N}(\mathbf{0}, I_d)$, we have*

$$\mathbb{E}\left[\mathcal{W}_2^2\left(\mu_N, \mu\right)\right] \leq C_d \times \begin{cases} N^{-1} \log \log N & \text{if } d = 1 \\ N^{-1} (\log N)^2 & \text{if } d = 2 \\ N^{-2/d} & \text{if } d \geq 3 \end{cases}.$$

*Proof of Theorem 3.7.* Suppose the sample mean and covariance at time $t$ is $\boldsymbol{m}_t$ and $C_t$, and that the mean and covariance of the mean-field limit is $\boldsymbol{\mu}_t$ and $\Sigma_t$. We bound $\mathbb{E}[\mathcal{W}_2^2(\zeta_N^{(t)}, \rho_t)]$ using Theorems 3.1 and 3.6 and Lemma L.1 in six steps.

**Step I.** We prove that $(\boldsymbol{x}_i^{(t)})$ has the same distribution as $(\widetilde{\boldsymbol{x}}_i^{(t)})$ where $\widetilde{\boldsymbol{x}}_i^{(t)} = C_t^{1/2} C_0^{-1/2} (\boldsymbol{x}_i^{(0)} - \boldsymbol{m}_0) + \boldsymbol{m}_t$. Note that according to Theorem 3.6, where we have $\boldsymbol{x}_i^{(t)} = A_t(\boldsymbol{x}_i^{(0)} - \boldsymbol{m}_0) + \boldsymbol{m}_t$. Here $A_t$ is the unique (matrix) solution of the linear system

$$\dot{A}_t = \left(I - Q^{-1}(C_t + \boldsymbol{\mu}_t \boldsymbol{\mu}_t^\top) + Q^{-1} \boldsymbol{b} \boldsymbol{\mu}_t^\top\right) A_t, \quad A_0 = I, \tag{84}$$

and $\boldsymbol{m}_t$ and $C_t$ are the unique solution of the ODE system

$$\begin{cases} \dot{\boldsymbol{m}}_t = (I - Q^{-1} C_t) \boldsymbol{m}_t - (1 + \boldsymbol{m}_t^\top \boldsymbol{m}_t) Q^{-1} (\boldsymbol{m}_t - \boldsymbol{b}) \\ \dot{C}_t = 2C_t - C_t \left(C_t + \boldsymbol{m}_t (\boldsymbol{m}_t - \boldsymbol{b})^\top\right) Q^{-1} - Q^{-1} \left(C_t + (\boldsymbol{m}_t - \boldsymbol{b}) \boldsymbol{m}_t^\top\right) C_t \end{cases}. \tag{85}$$

Since $C_t$ is the sample covariance, we have

$$A_t C_0 A_t^\top = C_t,$$

which simplies that

$$(C_t^{-1/2} A_t C_0^{1/2})(C_t^{-1/2} A_t C_0^{1/2})^\top = I.$$

Thus, $P_t = C_t^{-1/2} A_t C_0^{1/2}$ is an orthogonal matrix, and we have

$$A_t = C_t^{1/2} P_t C_0^{-1/2}.$$

Since the multivariate Gaussian distribution $\mathcal{N}(\mathbf{0}, I_d)$ is invariant under orthogonal transformation, the joint distribution of $\left(C_0^{-1/2}(\boldsymbol{x}_i^{(0)} - \boldsymbol{m}_0)\right)_{i=1}^{N}$ is also invariant under $P_t$. Thus, we have $(\boldsymbol{x}_i^{(t)} - \boldsymbol{m}_t)$ has the same distribution as $(\widetilde{\boldsymbol{x}}_i^{(t)} - \boldsymbol{m}_t)$ and Step I is proven.

**Step II.** Establish uniform decay rates for $\|C_t - Q\|_F$ and $\|\boldsymbol{m}_t - \boldsymbol{b}\|$. We begin by checking the energy function

$$0 \leq E(\boldsymbol{m}_t, C_t) = \frac{1}{2} \left(\operatorname{tr}(Q^{-1} C_t) - \log \det(Q^{-1} C_t) - d + (\boldsymbol{m}_t - \boldsymbol{b})^\top Q^{-1} (\boldsymbol{m}_t - \boldsymbol{b})\right).$$

As shown in the proof of Theorem 3.1, we have

$$\dot{E}(\boldsymbol{m}_t, C_t) = -\left\|C_t Q^{-1} - I + \boldsymbol{m}_t (\boldsymbol{m}_t - \boldsymbol{b})^\top Q^{-1}\right\|_F^2 - \|Q^{-1}(\boldsymbol{m}_t - \boldsymbol{b})\|^2 \leq 0.$$

Thus, $E(\boldsymbol{m}_t, C_t) \leq E(\boldsymbol{m}_0, C_0)$ for any $t \geq 0$. Furthermore, similar to the proof of Theorem 3.1 we check that

$$\begin{aligned}
&- \dot{E}(\boldsymbol{m}_t, C_t) \\
&= \begin{bmatrix} \operatorname{vec}^\top \left(Q^{-1} C_t - I\right) & (\boldsymbol{m}_t - \boldsymbol{b})^\top Q^{-1} \end{bmatrix} \begin{bmatrix} I_{d^2} & \boldsymbol{m}_t \otimes I_d \\ \boldsymbol{m}_t^\top \otimes I_d & (1 + \boldsymbol{m}_t^\top \boldsymbol{m}_t) I_d \end{bmatrix} \begin{bmatrix} \operatorname{vec} \left(Q^{-1} C_t^{-1} - I\right) \\ Q^{-1}(\boldsymbol{m}_t - \boldsymbol{b}) \end{bmatrix} \\
&\geq \gamma_t \begin{bmatrix} \operatorname{vec}^\top \left(Q^{-1} C_t - I\right) & (\boldsymbol{m}_t - \boldsymbol{b})^\top Q^{-1} \end{bmatrix} \begin{bmatrix} I_{d^2} & \\ & 2Q \end{bmatrix} \begin{bmatrix} \operatorname{vec} \left(Q^{-1} C_t - I\right) \\ Q^{-1}(\boldsymbol{m}_t - \boldsymbol{b}) \end{bmatrix} \\
&= \gamma_t \operatorname{tr}\left((Q^{-1} C_t - I)^\top (Q^{-1} C_t - I)\right) + 2\gamma_t (\boldsymbol{m}_t - \boldsymbol{b})^\top Q (\boldsymbol{m}_t - \boldsymbol{b}) \\
&\geq 2\gamma_t \cdot \left(\operatorname{tr}(Q^{-1} C_t - I) - \log \det(Q^{-1} C_t) + (\boldsymbol{m}_t - \boldsymbol{b})^\top Q (\boldsymbol{m}_t - \boldsymbol{b})\right) \\
&= 4\gamma_t E(\boldsymbol{m}_t, C_t).
\end{aligned}$$

where $\gamma_t$ is the smallest eigenvalue of

$$
\begin{bmatrix}
I_{d^2} & \frac{1}{\sqrt{2}}\boldsymbol{m}_t \otimes Q^{-1/2} \\
\frac{1}{\sqrt{2}}\boldsymbol{m}_t^\top \otimes Q^{-1/2} & \frac{1}{2}(1 + \boldsymbol{m}_t^\top \boldsymbol{m}_t)Q^{-1}
\end{bmatrix},
$$

and as shown in the proof of Theorem 3.1 it has a lower bound

$$
\gamma_t > \frac{1}{1 + \boldsymbol{m}_t^\top \boldsymbol{m}_t + q_{\max}},
$$

where $q_{\max}$ is the largest eigenvalue of $Q$. Now since

$$
\frac{1}{2}(\boldsymbol{m}_t - \boldsymbol{b})^\top Q^{-1}(\boldsymbol{m} - \boldsymbol{b}) \leq E(\boldsymbol{m}_t, C_t) \leq E(\boldsymbol{m}_0, C_0),
$$

we know $(\boldsymbol{m}_t - \boldsymbol{b})^\top (\boldsymbol{m}_t - \boldsymbol{b}) \leq 2q_{\max}E(\boldsymbol{m}_0, C_0)$. Thus, $\|\boldsymbol{m}_t\|$ is upper bounded by some quantity $F_1 = F_{1;Q,\boldsymbol{b},C_0,\boldsymbol{m}_0}$. Hence $\gamma_t$ is uniformly lower bounded by

$$
\gamma^* := \inf_{t \geq 0} \gamma_t \geq \frac{1}{1 + F_1^2 + q_{\max}}.
$$

Thus, by Grönwall's inequality we have $E(\boldsymbol{m}_t, C_t) \leq e^{-4\gamma^* t}E(\boldsymbol{m}_0, C_0)$. There exists $F_2 = F_{2;Q,\boldsymbol{b},C_0,\boldsymbol{m}_0}$ such that $\|\boldsymbol{m}_t - \boldsymbol{b}\| \leq e^{-2\gamma^* t}F_2$.

Now similarly

$$
0 \leq \frac{1}{2}\left(\operatorname{tr}(Q^{-1}C_t - 1) - \log\det(Q^{-1}C_t)\right) \leq E(\boldsymbol{m}_t, C_t) \leq e^{-4\gamma^* t}E(\boldsymbol{m}_0, C_0)
$$

also renders an upper bound for $\|C_t - Q\|_F$ with exponential decay noting that $\operatorname{tr}(A) - \log\det(I + A)$ is quadratic in $\|A\|_F$ when $\|A\|_F$ is small, i.e., there exists $F_3 = F_{3;Q,\boldsymbol{b},C_0,\boldsymbol{m}_0}$ such that

$$
\|C_t - Q\|_F \leq e^{-2\gamma^* t}F_3.
$$

By Theorem 3.1 we know that $\boldsymbol{\mu}_t, \Sigma_t$ satisfy the same ODEs as $\boldsymbol{m}_t, C_t$:

$$
\begin{cases}
\dot{\boldsymbol{\mu}}_t = (I - Q^{-1}\Sigma_t)\boldsymbol{\mu}_t - (1 + \boldsymbol{\mu}_t^\top \boldsymbol{\mu}_t)Q^{-1}(\boldsymbol{\mu}_t - \boldsymbol{b}) \\
\dot{\Sigma}_t = 2\Sigma_t - \Sigma_t\left(\Sigma_t + \boldsymbol{\mu}_t(\boldsymbol{\mu}_t - \boldsymbol{b})^\top\right)Q^{-1} - Q^{-1}\left(\Sigma_t + (\boldsymbol{\mu}_t - \boldsymbol{b})\boldsymbol{\mu}_t^\top\right)\Sigma_t
\end{cases}. \tag{86}
$$

Thus, similarly we have

$$
\|\boldsymbol{\mu}_t - \boldsymbol{b}\| \leq e^{-2\gamma^* t}F_2', \quad \|\Sigma_t - Q\|_F \leq e^{-2\gamma^* t}F_3'.
$$

**Step III.** Show that $\|C_t - \Sigma_t\|_F$ and $\|\boldsymbol{m}_t - \boldsymbol{\mu}_t\|$ can be controlled after sufficient time.

For any $\epsilon > 0$ we define

$$
\Theta_\epsilon := \left\{(\boldsymbol{v}, S) \in \mathbb{R}^d \times \operatorname{Sym}^+(d, \mathbb{R}) : \|\boldsymbol{v} - \boldsymbol{b}\| \leq \epsilon, \text{ and } (1 - \epsilon)Q \preceq S \preceq (1 + \epsilon)Q\right\},
$$

and consider the relative energy function

$$
E(\boldsymbol{m}_t, C_t, \boldsymbol{\mu}_t, \Sigma_t) = \frac{1}{2}\left(\operatorname{tr}\left(C_t^{-1}\Sigma_t - I\right) - \log\det(C_t^{-1}\Sigma_t) + (\boldsymbol{m}_t - \boldsymbol{\mu}_t)^\top Q^{-1}(\boldsymbol{m}_t - \boldsymbol{\mu}_t)\right).
$$

We show that for $\epsilon$ small enough if $(\boldsymbol{m}_t, C_t), (\boldsymbol{\mu}_t, \Sigma_t) \in \Theta_\epsilon$, then

$$
\dot{E}(\boldsymbol{m}_t, C_t, \boldsymbol{\mu}_t, \Sigma_t) \leq 0.
$$

We derive that

$$
\begin{aligned}
&\dot{E}(\boldsymbol{m}_t, C_t, \boldsymbol{\mu}_t, \Sigma_t) \\
&= \frac{1}{2}\operatorname{tr}\left(C_t^{-1}(\Sigma_t - C_t)\left(\Sigma_t^{-1}\dot{\Sigma}_t - C_t^{-1}\dot{C}_t\right)\right) + (\boldsymbol{m}_t - \boldsymbol{\mu}_t)^\top Q^{-1}(\dot{\boldsymbol{m}}_t - \dot{\boldsymbol{\mu}}_t).
\end{aligned}
$$

Here note that

$$
- C_t^{-1}(\Sigma_t - C_t)\left(\Sigma_t^{-1}\dot{\Sigma}_t - C_t^{-1}\dot{C}_t\right)
$$

$$
\overset{\mathrm{tr}}{=}C_t^{-1}(\Sigma_t - C_t)\Big((\Sigma_t + \boldsymbol{\mu}_t(\boldsymbol{\mu}_t - \boldsymbol{b})^\top - C_t - \boldsymbol{m}_t(\boldsymbol{m}_t - \boldsymbol{b})^\top)Q^{-1}
$$

$$
+ \Sigma_t^{-1}Q^{-1}(\Sigma_t + (\boldsymbol{\mu}_t - \boldsymbol{b})\boldsymbol{\mu}_t^\top)\Sigma_t - C_t^{-1}Q^{-1}(C_t + (\boldsymbol{m}_t - \boldsymbol{b})\boldsymbol{m}_t^\top)C_t\Big)
$$

$$
\overset{\mathrm{tr}}{=}2(C_t - \Sigma_t)^2 Q^{-1}C_t^{-1}
$$

$$
+ 2(C_t - \Sigma_t)\left(\boldsymbol{m}_t(\boldsymbol{m}_t - \boldsymbol{b})^\top - \boldsymbol{\mu}_t(\boldsymbol{\mu}_t - \boldsymbol{b})^\top\right)Q^{-1}C_t^{-1}
$$

$$
\overset{\mathrm{tr}}{\geq}(C_t - \Sigma_t)^2 Q^{-1}C_t^{-1}
$$

$$
- \left(\boldsymbol{m}_t(\boldsymbol{m}_t - \boldsymbol{b})^\top - \boldsymbol{\mu}_t(\boldsymbol{\mu}_t - \boldsymbol{b})^\top\right)^2 Q^{-1}C_t^{-1}
$$

$$
\overset{\mathrm{tr}}{\geq}(C_t - \Sigma_t)^2 Q^{-1}C_t^{-1}
$$

$$
- \frac{1}{1-\epsilon}\left(\boldsymbol{\mu}_t(\boldsymbol{m}_t - \boldsymbol{\mu}_t)^\top + (\boldsymbol{m}_t - \boldsymbol{\mu}_t)(\boldsymbol{m}_t - \boldsymbol{b})^\top\right)^2 Q^{-2}.
$$

On the other hand, we have

$$
- (\boldsymbol{m}_t - \boldsymbol{\mu}_t)^\top Q^{-1}(\dot{\boldsymbol{m}}_t - \dot{\boldsymbol{\mu}}_t)
$$

$$
=(\boldsymbol{m}_t - \boldsymbol{\mu}_t)^\top(Q^{-2} - Q^{-1})(\boldsymbol{m}_t - \boldsymbol{\mu}_t) + (\boldsymbol{m}_t - \boldsymbol{\mu}_t)^\top Q^{-2}(C_t\boldsymbol{m}_t - \Sigma_t\boldsymbol{\mu}_t)
$$

$$
- (\boldsymbol{m}_t^\top\boldsymbol{m}_t - \boldsymbol{\mu}_t^\top\boldsymbol{\mu}_t)(\boldsymbol{m}_t - \boldsymbol{\mu}_t)^\top Q^{-2}\boldsymbol{b} + (\boldsymbol{m}_t - \boldsymbol{\mu}_t)^\top Q^{-2}(\boldsymbol{m}_t\boldsymbol{m}_t^\top\boldsymbol{m}_t - \boldsymbol{\mu}_t\boldsymbol{\mu}_t^\top\boldsymbol{\mu}_t)
$$

$$
=:I_1 + I_2 + I_3 + I_4.
$$

Here we have

$$
I_2 = (\boldsymbol{m}_t - \boldsymbol{\mu}_t)^\top Q^{-2}C_t(\boldsymbol{m}_t - \boldsymbol{\mu}_t) + (\boldsymbol{m}_t - \boldsymbol{\mu}_t)^\top Q^{-2}(C_t - \Sigma_t)\boldsymbol{\mu}_t
$$

$$
\geq (1-\epsilon)(\boldsymbol{m}_t - \boldsymbol{\mu}_t)^\top Q^{-1}(\boldsymbol{m}_t - \boldsymbol{\mu}_t) + (\boldsymbol{m}_t - \boldsymbol{\mu}_t)^\top Q^{-2}(C_t - \Sigma_t)\boldsymbol{\mu}_t
$$

$$
\geq (1-\epsilon)(\boldsymbol{m}_t - \boldsymbol{\mu}_t)^\top Q^{-1}(\boldsymbol{m}_t - \boldsymbol{\mu}_t) - \frac{1}{2}\operatorname{tr}\left((C_t - \Sigma_t)^2 Q^{-1}C_t^{-1}\right)
$$

$$
- \frac{1}{2}\operatorname{tr}\left((\boldsymbol{\mu}_t(\boldsymbol{m}_t - \boldsymbol{\mu}_t)^\top)^2 Q^{-3}C_t\right)
$$

$$
\geq (1-\epsilon)(\boldsymbol{m}_t - \boldsymbol{\mu}_t)^\top Q^{-1}(\boldsymbol{m}_t - \boldsymbol{\mu}_t) - \frac{1}{2}\operatorname{tr}\left((C_t - \Sigma_t)^2 Q^{-1}C_t^{-1}\right)
$$

$$
- \frac{1}{2}(1+\epsilon)\operatorname{tr}\left((\boldsymbol{\mu}_t(\boldsymbol{m}_t - \boldsymbol{\mu}_t)^\top)^2 Q^{-2}\right),
$$

and

$$
I_3 + I_4 =(\boldsymbol{m}_t - \boldsymbol{\mu}_t)^\top Q^{-2}\Big(\boldsymbol{\mu}_t\boldsymbol{\mu}_t^\top(\boldsymbol{m}_t - \boldsymbol{\mu}_t) + (\boldsymbol{m}_t\boldsymbol{m}^\top - \boldsymbol{\mu}_t\boldsymbol{\mu}_t^\top)(\boldsymbol{m}_t - \boldsymbol{b})\Big).
$$

Combining all these together we have

$$
- \dot{E}(\boldsymbol{m}_t, C_t, \boldsymbol{\mu}_t, \Sigma_t)
$$

$$
\geq(\boldsymbol{m}_t - \boldsymbol{\mu}_t)^\top(Q^{-2} - \epsilon Q^{-1})(\boldsymbol{m}_t - \boldsymbol{\mu}_t) - \frac{1}{2}(1+\epsilon)\operatorname{tr}\left((\boldsymbol{\mu}_t(\boldsymbol{m}_t - \boldsymbol{\mu}_t)^\top)^2 Q^{-2}\right)
$$

$$
- \frac{1}{2(1-\epsilon)}\operatorname{tr}\left(\left(\boldsymbol{\mu}_t(\boldsymbol{m}_t - \boldsymbol{\mu}_t)^\top + (\boldsymbol{m}_t - \boldsymbol{\mu}_t)(\boldsymbol{m}_t - \boldsymbol{b})^\top\right)^2 Q^{-2}\right)
$$

$$
+ (\boldsymbol{m}_t - \boldsymbol{\mu}_t)^\top Q^{-2}\Big(\boldsymbol{\mu}_t\boldsymbol{\mu}_t^\top(\boldsymbol{m}_t - \boldsymbol{\mu}_t) + (\boldsymbol{m}_t\boldsymbol{m}^\top - \boldsymbol{\mu}_t\boldsymbol{\mu}_t^\top)(\boldsymbol{m}_t - \boldsymbol{b})\Big)
$$

$$
=(\boldsymbol{m}_t - \boldsymbol{\mu}_t)^\top\left(Q^{-2} - \epsilon Q^{-1} - \frac{\epsilon(2-\epsilon)}{2(1-\epsilon)}Q^{-2}\boldsymbol{\mu}_t\boldsymbol{\mu}_t^\top\right)(\boldsymbol{m}_t - \boldsymbol{\mu}_t) + \mathcal{O}(\epsilon^2).
$$

Since $0 < q_{\min}I \preceq Q \preceq q_{\max}I$, and $\|\boldsymbol{m}_t\|$ and $\|\boldsymbol{\mu}_t\|$ are bounded by $F_1$ as shown in Step II, there exists $\epsilon_0 = \epsilon_{Q,\boldsymbol{b},C_0,\boldsymbol{m}_0,\Sigma_0,\boldsymbol{\mu}_0}$ ($\epsilon_0$ can be seen as a continuous function) such that for any $\epsilon \leq \epsilon_0$ we have $\dot{E}(\boldsymbol{m}_t, C_t, \boldsymbol{\mu}_t, \Sigma_t) \leq 0$ as long as we have $(\boldsymbol{m}_t, C_t) \in \Theta_\epsilon$ and $(\boldsymbol{\mu}_t, \Sigma_t) \in \Theta_\epsilon$.

Given $Q, \boldsymbol{b}, C_0, \boldsymbol{m}_t, \Sigma_0, \boldsymbol{\mu}_0$ suppose at time $t_0$ we have $(\boldsymbol{m}_{t_0}, C_{t_0}) \in \Theta_{\epsilon_0}$ and $(\boldsymbol{\mu}_{t_0}, \Sigma_{t_0}) \in \Theta_{\epsilon_0}$. Then for any $t > t_0$ we know

$$E(\boldsymbol{m}_t, C_t, \boldsymbol{\mu}_t, \Sigma_t) \leq E(\boldsymbol{m}_{t_0}, C_{t_0}, \boldsymbol{\mu}_{t_0}, \Sigma_{t_0})$$

$$\leq \frac{1}{4} \operatorname{tr}((C_{t_0} - \Sigma_{t_0})^2 C_{t_0}^{-2}) + (\boldsymbol{m}_{t_0} - \boldsymbol{\mu}_{t_0})^\top Q^{-1} (\boldsymbol{m}_{t_0} - \boldsymbol{\mu}_{t_0})$$

$$\leq \frac{1}{4(1 - \epsilon_0)^2 q_{\min}^2} \|C_{t_0} - \Sigma_{t_0}\|_F^2 + \frac{1}{q_{\min}} \|\boldsymbol{m}_{t_0} - \boldsymbol{\mu}_{t_0}\|^2.$$

Note that

$$\lim_{\|A\|_F \to 0} \frac{\operatorname{tr}(A) - \log \det(I + A)}{\operatorname{tr}(A^\top A)} = \frac{1}{2}.$$

Fixing any $\delta > 0$ as long as $\epsilon_0$ is small enough we have

$$E(\boldsymbol{m}_t, C_t, \boldsymbol{\mu}_t, \Sigma_t) \geq \frac{1}{4 + \delta} \operatorname{tr}((C_t - \Sigma_t)^2 C_t^{-2}) + (\boldsymbol{m}_t - \boldsymbol{\mu}_t)^\top Q^{-1} (\boldsymbol{m}_t - \boldsymbol{\mu}_t)$$

$$\geq \frac{1}{(4 + \delta)(1 + \epsilon_0)^2 q_{\max}^2} \|C_t - \Sigma_t\|_F^2 + \frac{1}{q_{\max}} \|\boldsymbol{m}_t - \boldsymbol{\mu}_t\|^2.$$

Therefore, we conclude that for any given $Q, \boldsymbol{b}, C_0, \boldsymbol{m}_0, \Sigma_0$ and $\boldsymbol{\mu}_0$ there exists $\epsilon_0$ as stated above and $F_4 = F_{4;Q,\boldsymbol{b},C_0,\boldsymbol{m}_0,\Sigma_0,\boldsymbol{\mu}_0} > 0$ such that as long as $(\boldsymbol{m}_{t_0}, C_{t_0}) \in \Theta_{\epsilon_0}$ and $(\boldsymbol{\mu}_{t_0}, \Sigma_{t_0}) \in \Theta_{\epsilon_0}$ then for any $t > t_0$

$$\|C_t - \Sigma_t\|_F^2 \leq F_4 \left( \|C_{t_0} - \Sigma_{t_0}\|_F^2 + \|\boldsymbol{m}_{t_0} - \boldsymbol{\mu}_{t_0}\|^2 \right),$$

$$\|\boldsymbol{m}_t - \boldsymbol{\mu}_t\|^2 \leq F_4 \left( \|C_{t_0} - \Sigma_{t_0}\|_F^2 + \|\boldsymbol{m}_{t_0} - \boldsymbol{\mu}_{t_0}\|^2 \right).$$

**Step IV.** Uniformly bound $\|C_t - \Sigma_t\|_F^2$ and $\|\boldsymbol{m}_t - \boldsymbol{\mu}_t\|^2$ using $\|C_0 - \Sigma_0\|_F^2 + \|\boldsymbol{m}_0 - \boldsymbol{\mu}_0\|^2$.

Note that by definition of the Frobenius norm (or any matrix norm), given $\epsilon_0 > 0$ there exists $\epsilon_1 \in (0, \epsilon_0)$ such that for any $S \in \operatorname{Sym}(d, \mathbb{R})$ as long as the norm is small enough, i.e., $\|S - Q\|_F \leq \epsilon_1$, then we have $(1 - \epsilon_0)Q \preceq S \preceq (1 + \epsilon_0)Q$. Then by Step II we know that if we set $F_5 = \max\{F_2, F_3, F_2', F_3'\}$ and $t_0 > -\frac{1}{2\gamma^*} \log \frac{\epsilon_1}{F_5}$ then the following bounds hold:

$$\|\boldsymbol{m}_t - \boldsymbol{b}\| < \epsilon_1, \quad \|C_t - Q\|_F < \epsilon_1, \quad \|\boldsymbol{\mu}_t - \boldsymbol{b}\| < \epsilon_1, \quad \|\Sigma_t - Q\|_F < \epsilon_1.$$

Now it is straight forward to check from (85) and (86) and the results in Step II that there exists $F_6 = F_{6;Q,\boldsymbol{b},C_0,\boldsymbol{m}_0,\Sigma_0,\boldsymbol{\mu}_t}$ such that for any $t \geq 0$

$$\frac{\mathrm{d}}{\mathrm{d}t} \|\boldsymbol{m}_t - \boldsymbol{\mu}_t\|^2 \leq F_6 \|\boldsymbol{m}_t - \boldsymbol{\mu}_t\|, \quad \frac{\mathrm{d}}{\mathrm{d}t} \|C_t - \Sigma_t\|_F^2 \leq F_6 \|C_t - \Sigma_t\|_F^2.$$

Thus, by Grönwall's inequality we have

$$\|\boldsymbol{m}_t - \boldsymbol{\mu}_t\|^2 \leq e^{F_6 t} \|\boldsymbol{m}_0 - \boldsymbol{\mu}_0\|^2, \quad \|C_t - \Sigma_t\|_F^2 \leq e^{F_6 t} \|C_0 - \Sigma_0\|_F^2.$$

Combining this with Step III, we know for any $t \geq 0$, there exists $F_7 = e^{F_6 t_0} F_4$ (only depending on $Q, \boldsymbol{b}, C_0, \boldsymbol{m}_0, \Sigma_0, \boldsymbol{\mu}_0$) such that

$$\|C_t - \Sigma_t\|_F^2 \leq F_7 \left( \|C_0 - \Sigma_0\|_F^2 + \|\boldsymbol{m}_0 - \boldsymbol{\mu}_0\|^2 \right),$$

$$\|\boldsymbol{m}_t - \boldsymbol{\mu}_t\|^2 \leq F_7 \left( \|C_0 - \Sigma_0\|_F^2 + \|\boldsymbol{m}_0 - \boldsymbol{\mu}_0\|^2 \right).$$

**Step V.** Let $\boldsymbol{y}_i^{(t)} := \Sigma_t^{1/2} \Sigma_0^{-1/2} (\boldsymbol{x}_i^{(0)} - \boldsymbol{\mu}_0) + \boldsymbol{\mu}_t$. Define $\xi_N^{(t)} = \frac{1}{N} \sum_{i=1}^N \delta_{\boldsymbol{y}_i^{(t)}}$ and $\widetilde{\zeta}_N^{(t)} = \frac{1}{N} \sum_{i=1}^N \delta_{\widetilde{\boldsymbol{x}}_i^{(t)}}$. We show that

$$\mathbb{E}\left[ \mathcal{W}_2^2 \left( \xi_N^{(t)}, \widetilde{\zeta}_N^{(t)} \right) \right] \overset{\circ}{\leq} \mathbb{E}\left[ \frac{1}{N} \sum_{i=1}^N \left\| \widetilde{\boldsymbol{x}}_N^{(t)} - \boldsymbol{y}_N^{(t)} \right\|^2 \right] \overset{\diamond}{=} o\left( \frac{\log \log N}{N} \right).$$

Since $\circ$ is trivial by the definition of the Wasserstein metric, we only need to check $\diamond$. In fact,

$$\frac{1}{N}\sum_{i=1}^{N}\left\|\widetilde{\boldsymbol{x}}_N^{(t)}-\boldsymbol{y}_N^{(t)}\right\|^2$$

$$\leq\frac{3}{N}\sum_{i=1}^{N}\left\|\left(\Sigma_t^{1/2}\Sigma_0^{-1/2}-C_t^{1/2}C_0^{-1/2}\right)\left(\boldsymbol{x}_i^{(0)}-\boldsymbol{\mu}_0\right)\right\|^2$$

$$+3\left\|C_t^{1/2}C_0^{-1/2}(\boldsymbol{m}_0-\boldsymbol{\mu}_0)\right\|^2+3\|\boldsymbol{\mu}_t-\boldsymbol{m}_t\|^2$$

$$=3\left\|\Sigma_t^{1/2}\Sigma_0^{-1/2}C_0^{1/2}-C_t^{1/2}\right\|_F^2+3\left\|C_t^{1/2}C_0^{-1/2}(\boldsymbol{m}_0-\boldsymbol{\mu}_0)\right\|^2+3\|\boldsymbol{\mu}_t-\boldsymbol{m}_t\|^2$$

$$=:I_5+I_6+I_7.$$

Note that

$$I_6\leq 3\|C_t\|_F\,\|C_0^{-1}\|_F\,\|\boldsymbol{m}_0-\boldsymbol{\mu}_0\|^2,$$
$$I_7\leq 3F_7\left(\|C_0-\Sigma_0\|_F^2+\|\boldsymbol{m}_0-\boldsymbol{\mu}_0\|^2\right).$$

By the central limit theorem $\sqrt{N}(\boldsymbol{m}_0-\boldsymbol{\mu}_0)$ converges to $\mathcal{N}(\mathbf{0},\Sigma_0)$ in distribution. Thus, $\sqrt{\frac{N}{\log\log N}}(\boldsymbol{m}_0-\boldsymbol{\mu}_0)$ converges to $\mathbf{0}$ in distribution and hence also in probability. (There could even be almost sure results using the law of iterated logarithm but converge in probability is good enough.)

Similarly by CLT every entries of $\sqrt{N}(C_0-\Sigma_0)$ converges to a Gaussian distribution. Thus, $\sqrt{\frac{N}{\log\log N}}(C_0-\Sigma_0)$ converges to zero matrix in probability. Therefore, we have $\frac{N}{\log\log N}\|C_t-\Sigma_t\|_F^2\to 0$ in probability.

By Step II, we have $\|C_t\|_F\leq\|Q\|_F+F_3$. All these constants here ($F_3$, $F_7$, $\|C_0^{-1}\|_F$) can be seen or chosen as a continuous function of $Q,\boldsymbol{b},C_0,\boldsymbol{m}_0,\Sigma_0,\boldsymbol{\mu}_0$ and by continuous mapping theorem they converge to the values of the same function with $C_0=\Sigma_0$ and $\boldsymbol{m}_0=\boldsymbol{\mu}_0$. Thus, we conclude that $\frac{N}{\log\log N}(I_6+I_7)\to 0$ in probability.

Now we derive that

$$I_5\leq\left\|(\Sigma_t^{1/2}-C_t^{1/2})\Sigma_0^{-1/2}C_0^{1/2}+C_t^{1/2}\Sigma_0^{-1/2}(C_0^{1/2}-\Sigma_0^{1/2})\right\|_F^2$$

$$\leq 2\|\Sigma_t^{1/2}-C_t^{1/2}\|_F^2\|\Sigma_0^{-1}\|_F\|C_0\|_F+\|C_t\|_F\|\Sigma_0^{-1}\|_F\|C_0^{1/2}-\Sigma_0^{1/2}\|_F^2.$$

Now we show a lemma: Suppose $A,B\in\mathrm{Sym}^+(d,\mathbb{R})$ are two positive definite matrices. Then we have $\|A^{1/2}-B^{1/2}\|\leq\frac{1}{2\sqrt{\lambda}}\|A-B\|$ where $\lambda$ is the smallest eigenvalue of $A$ and $B$. Note that we are using the spectral norm here.

In fact, denote the largest eigenvector of $A^{1/2}-B^{1/2}$ by $\eta$, and let $\boldsymbol{x}\in\mathbb{R}^d$ be the corresponding eigenvector such that $\boldsymbol{x}^\top\boldsymbol{x}=1$. We have

$$\|A-B\|\geq\boldsymbol{x}^\top(A-B)\boldsymbol{x}$$
$$=\boldsymbol{x}^\top A^{1/2}(A^{1/2}-B^{1/2})\boldsymbol{x}+\boldsymbol{x}^\top(A^{1/2}-B^{1/2})B^{1/2}\boldsymbol{x}$$
$$=\eta\boldsymbol{x}^\top(A^{1/2}+B^{1/2})\boldsymbol{x}\geq 2\eta\sqrt{\lambda}.$$

Thus, we have that $\|A^{1/2}-B^{1/2}\|=\eta\leq\frac{1}{2\sqrt{\lambda}}\|A-B\|$. Moreover, the Frobenius norm is bounded by

$$\|A^{1/2}-B^{1/2}\|_F\leq\sqrt{d}\|A^{1/2}-B^{1/2}\|\leq\frac{\sqrt{d}}{2\sqrt{\lambda}}\|A-B\|\leq\frac{\sqrt{d}}{2\sqrt{\lambda}}\|A-B\|_F.$$

Applying this lemma, we know

$$\left\|C_0^{1/2}-\Sigma_0^{1/2}\right\|_F\leq\frac{\sqrt{d}}{2\sqrt{\lambda}}\|\Sigma_0-C_0\|_F,$$

where $\lambda$ is the smallest eigenvalue of $C_0$ and $\Sigma_0$, which converges to the smallest eigenvalue of $\Sigma_0$ as $N$ goes to infinity.

Next we need to show that the smallest eigenvalue of $C_t$ and $\Sigma_t$ are uniformly (in time) lower bounded by some $\lambda' > 0$ (depending on $Q, \boldsymbol{b}, C_0, \boldsymbol{m}_0, \Sigma_0, \boldsymbol{\mu}_0$). We revisit Step II, where we show that $E(\boldsymbol{m}_t, C_t) \leq E(\boldsymbol{m}_0, C_t)$. Then

$$\mathrm{tr}(Q^{-1}C_t - I) - \log\det(Q^{-1}C_t) \leq 2E(\boldsymbol{m}_0, C_0)$$

leads to a uniform lower bound of the smallest eigenvalue of $C_t$ since $\mathrm{tr}(Q^{-1}C_t - I) - \log\det(Q^{-1}C_t) \to \infty$ as the smallest eigenvalue of $C_t$ goes down to zero. Thus, we have

$$\left\| C_t^{1/2} - \Sigma_t^{1/2} \right\|_F^2 \leq \frac{d}{4\lambda'}\|\Sigma_t - C_t\|_F^2 \leq \frac{dF_7}{4\lambda'}(\|C_0 - \Sigma_0\|_F^2 + \|\boldsymbol{m}_0 - \boldsymbol{\mu}_0\|^2).$$

Following similar arguments we conclude that $\frac{N}{\log\log N}I_5 \to 0$ in probability as $N \to \infty$. Therefore, in probability

$$\frac{N}{\log\log N} \frac{1}{N} \sum_{i=1}^{N} \left\| \widetilde{\boldsymbol{x}}_i^{(t)} - \boldsymbol{y}_i^{(t)} \right\| \leq \frac{N}{\log\log N}(I_5 + I_6 + I_7) \to 0,$$

which implies that

$$\mathbb{E}\left[ \mathcal{W}_2^2\left( \xi_N^{(t)}, \widetilde{\zeta}_N^{(t)} \right) \right] = o\left( \frac{\log\log N}{N} \right).$$

**Step VI.** Apply Lemma L.1 to get the final result. Note that $\boldsymbol{x}_i^{(t)}$ are *i.i.d.* from $\mathcal{N}(\boldsymbol{0}, I)$ and $\boldsymbol{y}_i^{(t)}$ is a linear function of $\boldsymbol{x}_t^{(t)}$ (unlike $\boldsymbol{m}_t$ and $C_t$ which are random, $\boldsymbol{\mu}_t$ and $\Sigma_t$ are deterministic). By Lemma L.1 and Step II ($\boldsymbol{\mu}_t$ and $\Sigma_t$ are uniformly bounded) we have

$$\mathbb{E}\left[ \mathcal{W}_2^2\left( \xi_N^{(t)}, \rho_t \right) \right] \leq C_{Q, \boldsymbol{b}, \Sigma_0, \boldsymbol{\mu}_0} \times \begin{cases} N^{-1}\log\log N & \text{if } d = 1 \\ N^{-1}(\log N)^2 & \text{if } d = 2 \\ N^{-2/d} & \text{if } d \geq 3 \end{cases}. \tag{87}$$

Thus, we derive that

$$\mathbb{E}\left[ \mathcal{W}_2^2(\zeta_N^{(t)}, \rho_t) \right] \overset{(*)}{=} \mathbb{E}\left[ \mathcal{W}_2^2(\widetilde{\zeta}_N^{(t)}, \rho_t) \right]$$

$$\overset{(**)}{\leq} \mathbb{E}\left[ \left( \mathcal{W}_2(\widetilde{\zeta}_N^{(t)}, \xi_N^{(t)}) + \mathcal{W}_2(\xi_N^{(t)}, \rho_t) \right)^2 \right]$$

$$\leq 2\mathbb{E}\left[ \mathcal{W}_2^2\left( \xi_N^{(t)}, \widetilde{\zeta}_N^{(t)} \right) \right] + 2\mathbb{E}\left[ \mathcal{W}_2^2\left( \xi_N^{(t)}, \rho_t \right) \right]$$

$$\overset{(***)}{\leq} C_{Q, \boldsymbol{b}, \Sigma_0, \boldsymbol{\mu}_0} \times \begin{cases} N^{-1}\log\log N & \text{if } d = 1 \\ N^{-1}(\log N)^2 & \text{if } d = 2 \\ N^{-2/d} & \text{if } d \geq 3 \end{cases}.$$

Note that we have used Step I in $(*)$, the triangle inequality in $(**)$, and Step V along with (87) in $(***)$.

$\square$

*Proof of Theorem F.4.* The proof is roughly similar to that of Theorem 3.7. But Step III is a little different (can be strengthened and simplified at the same time).

We show the global contraction of $\|\boldsymbol{m}_t - \boldsymbol{\mu}_t\|$ and also the contraction of $\|C_t - \Sigma_t\|_F$ after sufficient time. Note here $\|\cdot\|_F$ is the Frobenius norm.

Frist we check the derivative of squared Euclidean norm of $\boldsymbol{m}_t - \boldsymbol{\mu}_t$.

$$\frac{1}{2}\frac{\mathrm{d}}{\mathrm{d}t}\|\boldsymbol{m}_t - \boldsymbol{\mu}_t\|^2 = (\boldsymbol{m}_t - \boldsymbol{\mu}_t)^\top(\dot{\boldsymbol{m}}_t - \dot{\boldsymbol{\mu}}_t)$$

$$= -(\boldsymbol{m}_t - \boldsymbol{\mu}_t)^\top Q^{-1}(\boldsymbol{m}_t - \boldsymbol{\mu}_t) \leq -\frac{1}{q_{\max}}\|\boldsymbol{m}_t - \boldsymbol{\mu}_t\|^2.$$

Thus, $\|\boldsymbol{m}_t - \boldsymbol{\mu}_t\| \leq e^{-\frac{t}{q_{\max}}} \|\boldsymbol{m}_0 - \boldsymbol{\mu}_0\|$.

To bound $\|C_t - \Sigma_t\|_F$ we define

$$\mathcal{S}_\epsilon := \left\{ S \in \mathrm{Sym}^+(d, \mathbb{R}) : (1 - \epsilon)Q \preceq S \preceq (1 + \epsilon)Q \right\}.$$

This time we do not need the relative energy but can directly check the derivative of the squared Frobenius norm:

$$\frac{1}{2}\frac{\mathrm{d}}{\mathrm{d}t}\|C_t - \Sigma_t\|_F^2 = \frac{1}{2}\frac{\mathrm{d}}{\mathrm{d}t}\,\mathrm{tr}((C_t - \Sigma_t)^2) \overset{\mathrm{tr}}{=} (C_t - \Sigma_t)(\dot{C}_t - \dot{\Sigma}_t)$$

$$\overset{\mathrm{tr}}{=} 2(C_t - \Sigma_t)^2 - 2(C_t - \Sigma_t)(C_t^2 - \Sigma_t^2)Q^{-1}$$

$$\overset{\mathrm{tr}}{=} 2(C_t - \Sigma_t)^2 - (C_t - \Sigma_t)(C_t + \Sigma_t)(C_t - \Sigma_t)Q^{-1} - (C_t - \Sigma_t)^2(C_t + \Sigma_t)Q^{-1}$$

$$\overset{\mathrm{tr}}{=} 2(C_t - \Sigma_t)^2 - (C_t - \Sigma_t)^2\,(C_t + \Sigma_t)\,Q^{-1} - (C_t - \Sigma_t)(C_t + \Sigma_t)(C_t - \Sigma_t)Q^{-1}. \tag{88}$$

where $\overset{\mathrm{tr}}{=}$ denotes equal in trace. Now if $C_t, \Sigma_t \in \mathcal{S}_\epsilon$ then

$$\mathrm{tr}\left((C_t - \Sigma_t)(C_t + \Sigma_t)(C_t - \Sigma_t)Q^{-1}\right) = \mathrm{tr}\left(Q^{-1/2}(C_t - \Sigma_t)(C_t + \Sigma_t)(C_t - \Sigma_t)Q^{-1/2}\right)$$

$$\geq \mathrm{tr}\left(2(1 - \epsilon)Q^{-1/2}(C_t - \Sigma_t)Q(C_t - \Sigma_t)Q^{-1/2}\right) \overset{\diamond}{\geq} 2(1 - \epsilon)\,\mathrm{tr}\left((C_t - \Sigma_t)^2\right), \tag{89}$$

and

$$\mathrm{tr}\left((C_t - \Sigma_t)^2(C_t + \Sigma_t)Q^{-1}\right) - 2(1 - \epsilon)\,\mathrm{tr}\left((C_t - \Sigma_t)^2\right)$$

$$= \mathrm{tr}\left((C_t - \Sigma_t)^2(C_t + \Sigma_t - 2(1 - \epsilon)Q)Q^{-1}\right)$$

$$= \frac{1}{2}\,\mathrm{tr}\left((C_t - \Sigma_t)\left((C_t + \Sigma_t - 2(1 - \epsilon)Q)Q^{-1} + Q^{-1}(C_t + \Sigma_t - 2(1 - \epsilon)Q)\right)(C_t - \Sigma_t)\right) \geq 0. \tag{90}$$

Note that $\diamond$ is not trivially true. We show it as a lemma: Suppose $A$ is a symmetric matrix and $B$ is a positive definite symmetric matrix. Then $\mathrm{tr}(ABAB^{-1}) \geq \mathrm{tr}(A^2)$.

In fact, we write $B = P\Lambda P^\top$ where $P$ is an orthogonal matrix and $\Lambda = \mathrm{diag}\{\lambda_2, \cdots, \lambda_d\}$ is a diagonal matrix. Then

$$\mathrm{tr}(ABAB^{-1}) = \mathrm{tr}(AP\Lambda P^\top AP\Lambda^{-1}P^\top) = \left\|\Lambda^{1/2}P^\top AP\Lambda^{-1/2}\right\|_F^2.$$

Denoting $P^\top AP = (a_{ij})$ we get

$$\left\|\Lambda^{1/2}P^\top AP\Lambda^{-1/2}\right\|_F^2 = \sum_{i,j=1}^d \left(\frac{\sqrt{\lambda_i}}{\sqrt{\lambda_j}}a_{ij}\right)^2$$

$$= \frac{1}{2}\sum_{i,j=1}^d \left(\frac{\lambda_i}{\lambda_j}a_{ij}^2 + \frac{\lambda_j}{\lambda_i}a_{ji}^2\right) \geq \sum_{i,j=1}^d a_{ij}^2 = \mathrm{tr}(A^2).$$

Thus, substituting (90) and (89) into (88) we get

$$\frac{\mathrm{d}}{\mathrm{d}t}\|C_t - \Sigma_t\|_F^2 = -2\,\mathrm{tr}\left((C_t - \Sigma_t)(\dot{C}_t - \dot{\Sigma}_t)\right) \leq -4(1 - 2\epsilon)\|C_t - \Sigma_t\|_F^2 \leq 0.$$

Suppose at time $t_0$ both $C_{t_0}$ and $\Sigma_{t_0}$ lie in $\mathcal{S}_\epsilon$. Then for any $t \geq t_0$ we have

$$\|C_t - \Sigma_t\|_F \leq e^{-2(1 - 2\epsilon)(t - t_0)}\|C_{t_0} - \Sigma_{t_0}\|_F.$$

$\square$

