# OpenReview forum: "Towards Understanding the Dynamics of Gaussian-Stein Variational Gradient Descent"
_NeurIPS.cc/2023/Conference — NeurIPS 2023 poster_

### Official Review · Reviewer_BMUk · 2023-06-20

**Soundness:** 4 excellent
**Presentation:** 4 excellent
**Contribution:** 4 excellent
**Rating:** 7
**Confidence:** 4

**Summary:**

This paper focuses on the theoretical understanding and algorithmic contributions of Gaussian-Stein Variational Gradient Descent (Gaussian-SVGD) for Gaussian Variational Inference (GVI). The paper discusses the dynamics of Gaussian-SVGD and provides convergence rates in both mean-field and finite-particle settings when the target is Gaussian. It also shows that Gaussian-SVGD converges to the best Gaussian approximation to the target in KL-divergence for non-Gaussian targets. The authors propose two algorithmic frameworks for density-based and particle-based implementations of Gaussian-SVGD, which encompass various GVI algorithms proposed in the literature as special cases. The Gaussian-SVGD, and other related algorithms, are compared against one another on three examples: Gaussian target, Bayesian logistic regression, and a Gaussian mixture model. The authors highlight future directions for research, including establishing convergence results for particle-based Gaussian-SVGD algorithms under log-concave and more general targets, and exploring acceleration techniques.

**Strengths:**

The paper is very well written and the mathematical content is clearly articulated. The authors have done a good job of relating this work to other similar works from the literature. The paper is well-structured and the authors' particular contributions to this line of research are clearly outlined.

The technical detail, as far as I can see, is correct and the contribution of this work to the ML community, in particular the new convergence rates, are of significant interest. The paper is closely linked to other related works but is still highly original.

The paper provides a nice balance between establishing important theoretical results and providing a usable algorithm for users to implement in practice.

**Weaknesses:**

This is a nice piece of work and there are no areas where I can see significant weaknesses. However there are a few things that the authors may wish to consider to improve their paper:

* Starting from a very simple question, "Why should I use Gaussian-SVGD?" I think the authors have skipped some of the motivation for this work. If you know that the target is Gaussian, then of course you would just fit a Gaussian to it and would not require a particle-based variational inference approach. If the target is non-Gaussian, then why would you use Gaussian-SVGD over standard SVGD?
* It is not clear why regularized SVGD is introduced. What is the motivation here? Is it simply to show that you can derive a Gaussian-SVGD algorithm with the regularized Stein metric?
* The kernel $K(\cdot,y)$ is introduced in Def 2.2 without being defined. Additionally, I think in eq. (1) you need to replace $\rho$ with $\rho_t$.
* The simulation section is unfortunately a bit weak compared to other similar papers. The authors only consider three simple models and the details are largely glossed over, (e.g. what is $d$? How many particles? etc.). The authors seem to have run out of space and only provide a short simulation study for the logistic regression example. It would be better to include a more detailed simulation study in the main paper and to consider more challenging models.
* Additionally, on the point of experiments, it is a bit surprising that SVGD is not also included in this list.

**Questions:**

* This sort of relates to my previous question approach motivation. It is well-known that SVGD does not scale well to high-dimensional target distributions. Is it possible to show, either theoretically (i.e. results like Theorem 3.7) or empirically, that Gaussian-SVGD is better than SVGD when $d$ is large?
* Similarly, what is known about how many particles are needed compared to SVGD?
* Particle-based methods are preferred over standard variational methods which fit a Gaussian distribution to the target because of their nonparametric nature. In the experiments section, it seems like the Gaussian mixture model would have been the more interesting example to include in the paper, particularly in the context of how good the approximation to the target is compared against SVGD. Could this be included?

**Limitations:**

The paper could discuss more the theoretical limitation of this work. However, from an ethical perspective, I do not believe that this paper touches on issues that would have a negative societal impact.

---

> ### Author Rebuttal · Authors · 2023-08-08
>
> Thanks for the detailed review and incisive questions.
>
> 1.**About why we should use Gaussian-SVGD; "If the target is non-Gaussian, why would you use Gaussian-SVGD over standard SVGD?"**:
>
> We interpret the question more as **when** we should use Gaussian-SVGD rather than **why**. It is important to note that **standard SVGD (e.g. using a bilinear or RBF kernel) does not solve GVI**. On one hand, if our goal is to find the best Gaussian approximation, we should no doubt use Gaussian-SVGD instead of SVGD. Furthermore, if the goal is to estimate the mean and variance from a target distribution, Gaussian-SVGD is still preferred because standard SVGD can subject to significant error in estimating the covariance especially when the dimension is high [B].
>
> However, as shown in [A] for a class of log-concave targets, samples generated via GVI (i.e., doing a Gaussian targets) would provide accurate estimators of the mean vector and the covariance matrix, and Gaussian-SVGD is essentially a deterministic algorithm to efficiently and accurately perform GVI.
>
> For a large class of Bayesian inference problems, thanks to the Bernstein von Mises theorem, the posterior distribution is approximately Gaussian in the limit of large samples under appropriate regularity conditions. In fact, there is an array of works providing **stochastic** algorithms (which is based on randomized discretizations of the BW Gradient flow) for solving GVI (see, e.g., [17, 36]), and [55, 68, 70, 57]  show the improved practical performance of GVI on various practical problems. We show in this work that there is a surprising **deterministic** discretization (BWPF) of the BW Gradient flow if we aim to solve GVI.
>
> Furthermore, when one refers to standard SVGD, one also needs to specify the choice of kernels. In a way, our work suggests a principled way of selecting the kernel (which could be one of the bi-linear kernels) in standard SVGD for provably estimating the posterior mean and covariance for log-concave targets.
>
> [A] A Katsevich, P Rigollet. "On the approximation accuracy of gaussian variational inference." arXiv preprint arXiv:2301.02168, 2023.
>
> [B] Ba, Jimmy, et al. "Understanding the variance collapse of SVGD in high dimensions." ICLR. 2021.
>
>
> 2.**About why regularized SVGD is introduced**:  What we want to show is that the regularized Stein metric with kernel $K_2$ coincides with the Stein metric with a different bilinear kernel, which we can call as $K_4$. This phenomenon is interesting and also makes the algorithm fall under our general framework of Gaussian-SVGD. We will modify the paragraph on "Three Bilinear Kernels" on Page 3 to include $K_4$. Moreover, $K_4$ is important because it interpolates between $K_2$ and $K_3$. Note that $K_2$ and $K_3$ both have advantages and disadvantages, and $K_4$ can strike a balance between them by choosing $\lambda$ (See Table 1 and Figure 1).
>
> 3.**Is Gaussian-SVGD is better than SVGD when d is large? and What is known about how many particles are needed compared to SVGD?**
>
> These two are very deep question, actually (from a theoretical perspective). For the first question, we do observe empirically that Gaussian-SVGD is better than SVGD (say with Gaussian or Matern kernels) when d is large, from our preliminary experiments in terms of estimating posterior covariances of log-concave targets. The intuitive reasoning is the parametric nature of SVGD incurs significantly less variance compared to nonparametric SVGD.
>
> In an ongoing work, we are attempting to have a high-dimensional analysis (i.e., double asymptotic analysis) of Gaussian SVGD. Comparing to general SVGD theoretically in this regime, and providing a theoretical answer to the second question, are harder because we currently don't have tools to analyze SVGD well, even in fixed or low-dimensional setup. Nevertheless, a positive answer to these question would be great, and we hope to get there someday!
>
> 4.**About Gaussian mixture model, simulation section and SVGD**:
>
> First, we would like to point out that we have a Gaussian mixture model experiment in Section D of the appendix.  Taking your suggestion, we plan to move all experiments to the main paper if we are allowed an additional page. If not, we will have the Gaussian mixture model in the main draft and other experiments in the appendix.
>
> Our primary goal is to make progress towards understanding SVGD theoretically, given its widespread usage in practice. Nevertheless, we will be happy to add more numerical experiments, with different dimensions for the current models and also some new models.
>
> For the experiments, when one refers to SVGD it is important to specify which kernel to use. Given that you simply to refer to SVGD, we take it that you refer to some non-bi-linear kernels? That is a great suggestion, and we will be happy to add a comparison to SVGD with a Gaussian or Matern kernel in our experiments.

---

> > ### Comment · Reviewer_BMUk · 2023-08-20
> >
> > Dear Authors,
> >
> > Thank you for your response to my questions. I will maintain my current score.

---

### Official Review · Reviewer_EWuB · 2023-07-04

**Soundness:** 4 excellent
**Presentation:** 3 good
**Contribution:** 2 fair
**Rating:** 5
**Confidence:** 4

**Summary:**

This paper delves into an examination of Gaussian-SVGD through the analysis of mean-field PDE and discrete particle systems. It offers evidence for finite particle convergence and supports these findings with empirical validations.


**Strengths:**

Through clear modeling, the article provides a comprehensive analysis for Gaussian-SVGD.

The paper gives new interpretations for Wasserstein Gradient Flow and SVGD under linear systems, which are very interesting.


**Weaknesses:**

The setup of the paper is relatively artificial, because in actual scenarios, linear kernels are rarely used to implement SVGD.

The paper does not put much effort into justifying the significance of Gaussian variational inference, such as making comparisons with other distribution classes. If there were clear conclusions, it would have a significant impact on the significance of this paper.

**Questions:**

Can you elaborate some advantages of Gaussian-SVGD compared with RBF-SVGD?


**Limitations:**

The setup of the article is somewhat artificial. It would be more meaningful if the significance of this linear system could be demonstrated.

---

> ### Author Rebuttal · Authors · 2023-08-08
>
> 1.**Linear kernels are rarely used to implement SVGD. It would be more meaningful if the significance of this linear system could be demonstrated. Signficance of GVI**
>
> First, we should emphasize again that Gaussian-SVGD is **NOT** SVGD with a linear kernel when the target is not Gaussian. We should use Gaussian-SVGD in order to perform GVI and cannot directly use RBF-SVGD for the same purpose. Moreover, there is no simple modification to make RBF-SVGD work for the GVI task. (The linear approximation in Equation (26) does not guarantee Gaussian dynamics for RBF kernel.) The linear kernel is used in our paper due to the fact that it leads to a **deterministic** algorithm that can be used to solve GVI.
>
>
> Second, we elaborate on why GVI (or more generally variational inference) is preferred to exact sampling in many situations. The key argument is that **if one is only interested in summary statistics of the target such as the mean and covariance (as is often the case in Bayesian inference), generating samples may not be the most suitable way to achieve this goal**. In contrast to exact sampling, variational inference (VI), aims to find, among all measures in a certain parameterized family P, the closest measure to the target distribution. In general, VI algorithms usually show better performance both computationally and statistically than sampling algorithms. In specific, classical MCMC methods like MALA can be very expensive in terms of time cost, and it is notoriously difficult to identify clear-cut stopping criteria for the algorithm; The particle-based RBF-SVGD suffers from severe particle degeneracy especially in high dimensions [C] and cannot guarantee good covariance estimation.
>
> For a large class of Bayesian inference problems, thanks to the Bernstein von Mises theorem, the posterior distribution is approximately Gaussian in the limit of large samples under appropriate regularity conditions. Thus, Gaussian VI is considered an important and basic task in this field. Notably a recent work [A] shows that in terms of mean and covariance estimation (of the posterior) using GVI has improved rates over other methods like Laplace approximation. In fact, there is an array of works providing **stochastic** algorithms (which is based on randomized discretizations of the BW Gradient flow) for solving GVI (see, e.g., [17, 36]), and [55, 68, 70, 57]  show the improved practical performance of GVI on various practical problems. We, however, present **deterministic** algorithms to do the same task, which are even more practically useful.
>
> Finally, the flexibliy offered by non-parametric nature has hindered researchers from obtaining theoretical results that align with practice. Given this, our contributions in this work as Reviewer BMUk pointed out provides "a nice balance between establishing important theoretical results and providing a usable algorithm for users to implement in practice". **Understanding SVGD on Gaussian families is an important step towards understanding the general situation. Surprisingly this special case has not been well studied in the literature and our work provides nearly-complete solution to this important special case.**
>
>
> [A] A Katsevich, P Rigollet. "On the approximation accuracy of gaussian variational inference." arXiv preprint arXiv:2301.02168, 2023.
>
> [C] Zhuo, Jingwei, et al. "Message passing Stein variational gradient descent." ICML, 2018.
>
> To conclude, the **advantages of Gaussian-SVGD compared with RBF-SVGD** come in the following two perspective:
>
> - Gaussian-SVGD performs GVI, which is a more preferred approach for Bayesian inference.
>
> - Gaussian-SVGD provides a theoretically principled framework for understanding SVGD that **aligns with practice**. Gaussian-SVGD provides a **deterministic algorithm** to implement GVI. Uniform-in-time propogation type result (i.e., Theorem 3.7) are available for Gaussian-SVGD. Such results are not available currently for RBF-SVGD.
>
> If a practitioner asks "When should I use Gaussian-SVGD", our paper provides a nearly-complete answer to this question. However, if a practitioner asks the question, "When should I use RBF-SVGD", currently no convincing answer could be given. The best we can think of is something along the lines of try it out and see if it works well, which is extremely ad-hoc.

---

> > ### Author Response · Authors · 2023-08-15
> > **update/feedback**
> >
> > Dear Reviewer EWuB,
> >
> > As the deadline for the discussion phase is fast approaching, we were curious if you had any feedback for our response. Please also let us know if you have any further questions. Thank you and looking forward to hearing from you.
> >
> > Sincerely,
> >
> > Authors

---

> > > ### Comment · Reviewer_EWuB · 2023-08-16
> > >
> > > Thank you for your clarification. I have reevaluate the rating, but I still suggest the authors to add more real-world justifications to support this algorithm.

---

> > > > ### Author Response · Authors · 2023-08-16
> > > > **Official Comment by Authors**
> > > >
> > > > Thank you for reevaluating your rating. Taking your suggestion, we will add further clarifications/discussions (along with the points in our previous reply) on use of GVI in practice.

---

### Official Review · Reviewer_Bt7r · 2023-07-06

**Soundness:** 3 good
**Presentation:** 3 good
**Contribution:** 3 good
**Rating:** 6
**Confidence:** 3

**Summary:**

This paper studies the Stein variational gradient descent and its variants with a bilinear kernel on the space of Gaussian measures. The authors prove the rate of convergence of the dynamics, proposed finite particle algorithms and proved a uniform-in-time propagation of chaos,  and finally prove the convergence rate of the finite particle algorithms.

**Strengths:**

1. The study of the dynamics properties of  SVGD and its variants on Gaussian space seems to be interesting, which is a natural analog to the Bures-Wasserstein gradient flow. This paper gives a comprehensive analysis of such algorithms, including the well-posedness of the dynamics and the rate of convergence.

2. The proof and the results seem to be correct. I didn't check all the details in the proof, but both the proof and the results make sense to me.


3. The paper is in general well-written, the theorems are stated properly, and I think it is not hard for readers to understand the main idea.

**Weaknesses:**

I do not observe an obvious weakness of this paper, but I do have a few questions, please refer to the "Questions" part.

**Questions:**

1. Intuitive explanation of the convergence rate: as proved in the paper, the rate of convergence of SVGD for centered Gaussian is $O(e^{- t})$, which do not depend on $\lambda$. This seems to be interesting, but I'm wondering if there's an intuitive explanation for this result? The dependence of $O(e^{- t/\lambda})$ in the Wasserstein gradient flow case makes more sense to me intuitively, since larger $\lambda$ will give a more "flat" potential.

2. Regarding the non-Gaussian target:
    - It is shown in Theorem 4.1 that when $\rho_{\theta^*}$ be the unique Gaussian measure that minimize $D_{KL}(\rho_{\theta^*}||\rho^*)$, then the Gaussian-SVGD will converge to $\rho_{\theta^*}$. But when there are multiple $\rho_{\theta^*}$ that achieve the minimization, will the dynamic converge to one of them?

    - Does the particle dynamics Algorithm 2 yield a similar uniform-in-time propagation of chaos results as in Theorem 3.7 with the same rate for Gaussian-SVGD?



Minor points and typos:

1. Some ambiguity in definitions：
    - The definition of the kernels on page 3 (lines 111-118) is not entirely clear to me, in particular, what is $\mu$ and $\Sigma$ here? If I understand the latter part correctly, I think $\mu$ and $\Sigma$ will be the mean and covariance of $\rho_t$, which is actually time variant. I suggest the authors clarify this point in the definition.
    - The same problem appears in equation (26) when defining $\widehat{\nabla V}$, since $\widehat{\nabla V}$ depends on $t$, which is a time-dependent vector field.


2. Definition 2, line 101: $G_{\rho}^{Wass}$ should be $G_{\rho}^{Stein}$

**Limitations:**

No limitations and potential negative societal impact.

---

> ### Author Rebuttal · Authors · 2023-08-08
>
> Thanks for the detailed review and incisive questions.
>
> 1.**Intuitive explanation of the convergence rate:**
>
> The intuition could be obtained from the looking at corresponding ODE arising in the analysis. The mean-field dynamics of SVGD with bilinear kernel (for $K_1, K_2$ or $K_3$) takes form of linear ODEs $\dot{\Sigma}_t \approx - \Sigma_t\Sigma^{-1} $ where $\Sigma$ is the scaling matrix of the kernel and $\Sigma_t$ is the covaraince matrix of the solution of the mean-field dynamics of SVGD started from Gaussian initial data. The above linear ODEs warrant the rate of convergence of $\Sigma_t$ to be $O(e^{-t})$ for $K_1,K_2$ and $O(e^{-\frac{t}{\lambda}})$ for $K_3$ since the scaling matrix $\Sigma$ is equal to the identity matrix in the first two cases and equal to the covaraince matrix of the target distribution for the case of $K_3$.
>
> Furthermore, this also explains the fact that by choosing different regimes of $\lambda$, regularized SVGD can interpolate between the two extremes, WGF and SVGD (See Table 1).
>
> 2.**Regarding the non-Gaussian target:**
>
> In the context of Theorem 4.1, when there are multiple $\rho_{\theta^*}$ that achieve the minimization, the dynamic will converge to one of them, but we can't distinguish where it converges as it depends on the initialization. We will clarify this further in the revision.
>
> 3.**Uniform-in-time propagation of chaos for particle dynamics**
>
> Currently, we don't have results of uniform-in-time propagation of chaos for Algorithm 2 with a general target, but we believe this should hold for the class of log-concave targets. Rigorously proving this is an interesting and challenging direction for future work.
>
> 4.**Typos:**
>
> Thanks very much for pointing out the typos. We will fix them, along with the ambiguities mentioned, in the revision.

---

### Official Review · Reviewer_1EPQ · 2023-07-09

**Soundness:** 3 good
**Presentation:** 1 poor
**Contribution:** 2 fair
**Rating:** 3
**Confidence:** 4

**Summary:**

This work performs a theoretical investigation of Gaussian SVGD, a special case of SVGD restricted to the submanifold of Gaussian densities by means of a bilinear kernel. The authors characterize the mean-field dynamics of Gaussian SVGD, with a particular focus on Gaussian targets and obtain finite-particle guarantees in certain settings. Moving beyond Gaussian targets, the authors show that the dynamics of mean-field Gaussian SVGD converges to the best Gaussian approximation (In KL divergence) to the target density, and obtain convergence rates for the mean-field dynamics with the target density is log-smooth and log-concave

**Strengths:**

The work provides a thorough treatment of the mean-field dynamics of Gaussian SVGD. The experimental evaluation is also satisfactory.

**Weaknesses:**

My primary concerns are as follows :

1. Most of the results presented in this work consider continuous-time mean-field dynamics under the assumption of a Bilinear kernel and Gaussian target density. In my opinion, these assumptions adversely impact the significance of these results, as they fail to explain one of the major appeals of SVGD, i.e., the fact that it provides a non-parametric approximation to a large class of target densities. (e.g. [1] shows discrete-time mean-field guarantees for subgaussian densities and [2] extends it to the class of densities satisfying a generalized Talagrand’s inequality). While the exponentially fast convergence guarantees for the mean-field dynamics are seemingly appealing, they fall short of providing a satisfactory explanation of the behavior of SVGD in practically relevant settings.


2. Despite the simplifying assumption of Gaussian targets and bilinear kernels, the work does not provide any quantitative discrete-time finite-particle convergence rates. To the best of my understanding, Theorem 3.9 is the only result that considers finite-particle discrete-time dynamics, but it does not provide any quantitative rates.


3. It is not clear why one should prefer Gaussian SVGD over the Bures-Wasserstein Gradient Flow [3] since the only setting where it seems to outperform Bures-Wasserstein Gradient Flow is that of centered Gaussians, which is a highly restrictive condition.


4. The results in Section 4 leave much to be desired. Considering the fact that the best known results for Bures-Wasserstein gradient flow [3] cover both log-concave and strongly log-concave densities (see also [4] for a computable JKO discretization of the Bures-Wasserstein flow with convergence guarantees for log-concave and log-strongly concave densities), it is not at all clear what the benefits of Gaussian SVGD are when the target is not Gaussian.


5. The overall presentation of the results requires a lot of work. For instance, it is not clearly stated where exactly in the Appendix each Theorem is proved.


6. In addition to the limited applicability of these results in explaining the behavior of SVGD (as the most practically relevant case is that of non-logconcave targets) and the absence of satisfactory finite-particle discrete-time guarantees, I am somewhat concerned about the technical novelty of these results. To the best of my understanding, It seems like the mean-field continuous-time guarantees of this work can be easily obtained from the well-established results on SVGD [5,6] by restricting to the (finite-dimensional) Gaussian submanifold. While limited theoretical contribution in itself is not a major weakness, I find it difficult to recommend acceptance given the unsatisfactory practical usefulness of these results.


[1] Salim et. al., “A Convergence Theory for SVGD in the Population Limit under Talagrand's Inequality T1”

[2] Sun et. al., “Convergence of Stein Variational Gradient Descent under a Weaker Smoothness Condition”

[3] Lambert et. al., “Variational inference via Wasserstein gradient flows”

[4] Diao et. al., “Forward-backward Gaussian variational inference via JKO in the Bures-Wasserstein Space”

[5] Duncan et. al., “On the geometry of Stein variational gradient descent”

[6] Liu, “Stein Variational Gradient Descent as Gradient Flow”




**Questions:**

1. Could you obtain discrete-time finite-particle rates for Gaussian SVGD under the settings considered in your results. If not, could you highlight what would be the technical challenges involved in obtaining such a result?


2. Could you clarify why the mean-field continuous-time Gaussian SVGD dynamics is of interest, when (to the best of my knowledge) : 1) It does not satisfactorily explain the behavior of SVGD in practical scenarios, 2) It is outperformed by Bures-Wasserstein Gradient Flows (as well as the practically implementable Bures-JKO scheme) in most settings


3. Could you comment on the technical novelty of these results? To the best of my understanding, the mean-field guarantees can be obtained by adapting the existing analysis of the Gradient Flow induced by SVGD [5,6] to the submanifold of Gaussian densities (a technique which is well-established for Wasserstein gradient flows in [3]).


4. I would recommend reworking the overall presentation. Explicitly stating where each theorem is proven in the Appendix would be a good starting point.


**Limitations:**

I found the discussion on the limitations of this work to be somewhat unsatisfactory. The authors state that limitations are discussed on Section 6 (Other Related Works) and Section 7 (Conclusion). Section 6 discusses some of the prior literature and Section 7 highlights some future directions. Neither of these sections adequately discuss the limitations of this work (e.g. applicability is primarily restricted to centered Gaussian targets for most of the results, finite-particle discrete-time rates are absent in most settings)

---

> ### Author Rebuttal · Authors · 2023-08-08
>
> 1.***Could you obtain discrete-time finite-particle rates***
>
> Yes. We have the following **quantitative** result:
>
> **Theorem:** For a centered Gaussian target, suppose the SVGD particle system with $K_{1}$ or $K_{2}$ is initialized by $\bigl(\boldsymbol{x}\_{i}^{(0)}\bigr)\_{i=1}^{N}$ such that $\boldsymbol{\mu}\_{0}=\boldsymbol{0}$ and $C_{0}Q=QC\_{0}$. For $0<\epsilon< 0.5$, we have $\boldsymbol{\mu}\_{t}=\boldsymbol{0}$ and $\lVert C_{t}- Q \rVert\to 0$ as long as all the eigenvalues of $Q^{-1}C\_{0}$ lie in the interval $(0, 1+1/\epsilon)$. Furthermore, if we set $u_{\epsilon}$ to be the smaller root of the equation $f\_{\epsilon}'(u)=1-\epsilon$ (it has $2$ distinct roots) where $f\_{\epsilon}(x) := (1 + \epsilon(1-x))^2x$, then we have linear convergence, i.e.,
> $$
> \lVert C\_{t}- Q \rVert\leq (1-\epsilon)^{t}\lVert C_{0}-Q \rVert\leq e^{-\epsilon t}\lVert C\_{0}-Q \rVert
> $$
>
> as long as all the eigenvalues of $Q^{-1}C\_{0}$ lie in the interval $[u\_{\epsilon}, 1/3+1/(3\epsilon)]$.
>
> This is a direct refinement of Theorem 3.9 and is the 1st such result for finite-particle, discrete-time setting matching practice. The only other rates (for deterministic SVGD) by [62] do not align with practice. Obtaining similar results for general targets is extremely challenging and is one of the central open problems in SVGD.
>
> 2.**Bilinear kernel and Gaussian family assumptions**.
>
> None of the existing results in the literature cover this simple case. [49] requires that radial kernels, and bilinear kernels are not. [27]  relaxed this. But they need boundedness, which is also not satisfied by bilinear kernels.
>
> For Gaussian targets, we have shown uniform in time propagation of chaos in Theorem 3.7. Prior works only derived bounds for the empirical measure that **grow with time $t$**. This is unrealistic because SVGD only exactly recovers the target distribution or solves the GVI problem when it converges at $t=\infty$, resulting in $\infty$ in their bounds.
>
>  **Understanding SVGD on Gaussian families is an important step towards understanding the general situation. This special case has not been well studied in the literature and our work provides nearly-complete solution to this important special case.**
>
>
> 3.**Relation with BWGF**. There is a little bit misunderstanding here we guess and we apologize for lack of clarity previously.
>
> - Gaussian-SVGD is not 1 algorithm or 1 flow. **It is a family of flows with different bilinear kernels.** BWF is one of them (the one with kernel $K_3$, See Page 3). WGF and $K_3$-SVGD agree but only on the Gaussian submanifold and they induce BW flow.
>
> - We show each flow in this family induces two algorithms, one updates the density parameters ( so **stochastic**), the other updates the particles (so **deterministic**). The particle-based one is more stable in practice. **BWGD is the stochastic algorithm from BW flow appeared in [3] whereas BWPF is the particle-based one proposed by us.** It is hard to derive the deterministic particle-based counterpart of BWGD from the perspective of WGF. But from the perspective of $K_3$-SVGD, it is natural.
>
> 4. **With all due respects, we simply disagree with the claim on lack of theory novelty.**
>
> - We prove the **first** result on **uniform in time** propagation of chaos in SVGD. Previously such results were not available even for the setting we consider. Our proof ideas are generalizable to other setting although admittedly more challenging.
>
> - We explicitly solved the finite-particle dynamic system for Gaussian targets (See Theorem 3.6 and Theorem 3.8). Although there is no novelty in checking a given solution, **finding the solution is not trivial** in the first place.
>
> - We point out that our results with bilinear kernels **cannot be obtained as special cases of previous works on non-parametric SVGD**. [1] and [2] did not cover bilinear kernels. They require their kernels to be bounded and have bounded derivatives, which **is essentially required by their proof** and not satisfied by bilinear kernels. **Interestingly this is not an issue of technique but rather an issue of bilinear kernels.** The fact that Gaussian-SVGD using bilinear kernels is capable of solving GVI for general targets is **non-trivial** and that it provides deterministic algorithms for GVI is even more striking. No such **deterministic** particle method was available for GVI prior to our work.
>
> - For the mean field guarantee, [5] (Thm 22) or [6] is not applicable to our setting. Their condition cannot be easily checked in general (in fact not hold) for the Gaussian-SVGD setting we consider. Deriving tight results for special settings **rather than imposing unrealistic conditions for general settings** is a small but more solid step towards solving the general problem.
>
> - Directly applying the technique of [3] to SVGD would make the calculation too complicated to proceed. Therefore, we choose a different approach by calculating the explicit form of the restricted Riemannian metric tensor (e.g. See Theorem A.3). This approach is standard in Riemannian geometry but has added some novelty to our context.
>
> 5. The results in **[3] and [4]** provide rates for discrete-time finite-particle algorithms in the log-concave and non-log-concave settings. However, their algorithm is stochastic and their results only hold in expectation. The fluctuations of their **stochastic algorithms** are not quantified.
>
> Our finite-time discrete-time result is an exact guarantee for the **deterministic** algorithm! Compared to [3] and [4], it holds for Gaussian targets. So our work complements the results in the works of [3] and [4]. Extending the results in [3] and [4] to quantifying the fluctuations and extending our results for non-Gaussian targets are both interesting future directions to make a fair comparison between these different algorithms.
>
> 6.**Thanks for your suggestion. We will explicitly state where each theorem is proven in the main section of the paper.**

---

> > ### Author Response · Authors · 2023-08-15
> > **update/feedback**
> >
> > Dear Reviewer 1EPQ,
> >
> > As the deadline for the discussion phase is fast approaching, we were curious if you had any feedback for our response. Please also let us know if you have any further questions. Thank you and looking forward to hearing from you.
> >
> > Sincerely,
> >
> > Authors

---

> > ### Comment · Reviewer_1EPQ · 2023-08-19
> >
> > $\newcommand{\cN}{\mathcal{N}}$ I thank the authors for their response. Upon re-examining the proofs once again (which, admittedly, has been somewhat time-consuming given the issues with the structure and presentation) and reading the authors’ rebuttals, I am sorry to say that the rebuttal fails to adequately address my concerns. I discuss the major shortcomings below
> >
> > **Section 3:** The results of Section 3 considers the problem of sampling from $\cN(b, Q)$. For this target, the potential is given by $V(x) = \tfrac{1}{2}(x-b)^{T} Q^{-1} (x-b)$ and $\nabla V(x) = Q^{-1}(x-b)$. It is clear that evaluating $V$ and $\nabla V$ require knowledge of $b,Q$. To this end:
> >
> > ***Each of the dynamics under consideration in Section 3 requires knowledge of $b$ and $Q$ in order to sample from $\cN(b,Q)$. This is most apparent in the ODE system of Equation (3) and also observed in the finite particle systems of Equation (13) and Equations (19)***
> >
> > This renders the practical utility of these results useless as one can directly sample from $\cN(b,Q)$ if $b$ and $Q$ is known. The authors make the statement that *“Deriving tight results for special settings rather than imposing unrealistic conditions for general settings is a small but more solid step towards solving the general problem”*. Personally, I cannot think of a setting more unrealistic than sampling from $\cN(b,Q)$ with known $b, Q$
> >
> > While one could argue that despite the absence of practical utility, these results shed light on the behavior of SVGD. I would find such an argument to be quite weak since the setup considered in Section 3 is very far removed from practice. In particular, **I do not believe any applications where one would sample from $\cN(b, Q)$ with known $b, Q$  by using SVGD. Secondly, practical applications of SVGD typically do not use bilinear kernels**. On this note, I do not find the boundedness assumption of prior works to be very restrictive as this is satisfied by a large class of commonly used kernels (e.g. RBF, Laplace) whereas bilinear kernels are not commonly used.
> >
> > **Theorem 3.7:** The uniform-in-time propagation of chaos bound applies only to the continuous-time particle system of Equation (13), i.e. Gaussian SVGD with Gaussian Target and Bilinear Kernel. An examination of the proof in Appendix K clearly shows that the key steps in the proof crucially depends on the precise form of the dynamics (which is specific to $\cN(b, Q)$ and requires knowledge of $b, Q$ in advance to implement). **Given the hyper-specific nature of this result, I fail to see how this is an important contribution of its own. I find the statement that *Our proof ideas are generalizable to other setting* to be an overclaim.**
> >
> > **Section 4:** While the results on Section 4 admit some utility, I find the scope to be quite limited. **In particular, both Theorems 4.1 and 4.2 consider a continuous-time system and not a practical algorithm**. Convergence rates are only obtained under log-smoothness and log-concavity. On the contrary, **the work of Diao et. al. [4] actually gives an implementable discrete time algorithm with quantitative convergence rates for both log-concave and strongly log-concave targets**. Admittedly, the rates are in expectation but considering the fact that the stochasticity in their algorithm arises from Monte-Carlo estimation of Gaussian expectations, I believe high-probability guarantees should follow easily (as $ \nabla V$ is Lipschitz and $\nabla^2 V$ is bounded above and below in the PSD sense).
> >
> >  Overall, the absence of discrete-time finite-particle rates in Section 4 for the actual algorithm under consideration (i.e. Algorithm 2) significantly impacts the utility of these results. I find it concerning that the authors present the finite-particle system in Equation 26 to be an important contribution but do not prove any concrete nonasymptotic convergence guarantees.
> >
> > In its current state, I find the theoretical contributions of this work to be quite incomplete and the presentation to be immensely sloppy. To this end, I choose to keep my current score.

---

> > > ### Author Response · Authors · 2023-08-20
> > >
> > > We thank the reviewer 1EPQ for their feedback. Although we have provided detailed arguments in the reply to AC, we still want to emphasize one issue here. Throughout the page-long response of the reviewer, we believe **the only spot-on argument** for rejecting is that we have everything else but a discrete-time finite-particle bound for general targets. We admit that we are currently not able to achieve this desired result and would like to comment more regarding their concern.
> > >
> > > **Firstly getting a discrete-time finite-particle bound for general targets for Gaussian-SVGD is really a difficult problem.** Indeed, in the rebuttal we have provided the reason why the methods in previous works fail for our case. One key point is the troublesome unboundedness of bilinear kernels. (Why consider bilinear kernels? Note that bilinear kernels are used in Gaussian-SVGD for the purpose of performing GVI. Even for original SVGD we see no reason why such special and elegant case should be ignored in the studies.) Moreover, as the reviewer carefully figured out from examining our proof, our new approach does not directly yield discrete-time results for general targets even though the idea is general in principle.
> > >
> > > **Secondly this single weakness should by no means overweigh all the merits of our algorithmic and theoretical contributions.** As the reviewer had pointed out in their first feedback, "our experimental evaluation is satisfactory". While a complete theory is not avaliable, there is empirical evidence that the algorithm we propose has superior performance compared to previous methods. And such deterministic particle-based algorithm has its importance from algorithmic perspectives. Moreover, we provide a general framework that unifies various previous algorithms via different bilinear kernels along with a systematic way of comparing them. These algorithmic contributions are further strengthened by novel theoretical results with new insights, which we believe that we have already emphasized enough in the rebuttal. For example, if the reviewer is familiar with the literature of propagation of chaos or has once implemented SVGD, they should understand that the uniform vs non-uniform in time is a huge issue. Also, we believe anyone with proper math training could appreciate the beauty of our explicit solution for the finite particle system with Gaussian targets and bilinear kernels. We feel that it is a pity that all these important and interesting contributions were ignored by the reviewer in their response.
> > >
> > > Therefore, we would request the reviewer to re-evaluate our contribution in a more comprehensive manner. For certain algorithms, it might take years to develop its full theory. If a paper should be rejected only because it hasn't achieved a complete theory for the proposed algorithm, then the SVGD by Liu \& Wang and many other famous algorithms could not have been published at all, which would have been a great loss for the whole community.

---

### Official Review · Reviewer_ZXYv · 2023-07-26

**Soundness:** 3 good
**Presentation:** 3 good
**Contribution:** 3 good
**Rating:** 6
**Confidence:** 2

**Summary:**

This work studies the behavior of Gaussian-SVGD (SVGD projected to the space of Gaussian distributions, using bilinear kernels) and its variants.  The authors studied the behavior of the mean-field PDEs and established for Gaussian targets finite-particle convergence results which are significantly better than previous results for nonparametric kernels and targets. They have also derived density- and particle-based implementations of Gaussian-SVGD for general targets, which are shown to generalize recent works on Gaussian variational inference (GVI) with algorithmic guarantees.  One of the proposed algorithms outperform recent works on a Bayesian logistic regression experiment.

**Strengths:**

- The theoretical results appear interesting (to me as a non-expert, and note that I didn't check the proofs): the mean-field results sometimes improve over recent works on Gaussian variational inference with algorithmic guarantees, and the finite-particle results apply to standard SVGD, albeit with the other restrictions.

- The discussions on connections between Gaussian SVGD and existing GVI approaches are also interesting, and one of the proposed methods appear promising empirically.

**Weaknesses:**

- While the restrictions to Gaussian families and affine kernels are similar to some of the recent works, it inevitably limits the scope of this work as much of the interest around SVGD arises from its flexibility.

- While Gaussian variational inference in general has demonstrated competitive empirical performance, it is unclear if the proposed approach maintains this property, as it is not clear if the more complex algorithms interact nicely with common modifications such as minibatching.  This is also relevant because while some of the recent works (e.g., [17, 36]) also studied alternative GVI approaches, they come with full algorithmic guarantees, whereas for the present method there is no qualitative convergence guarantees for the discrete-time, finite-particle case.

**Questions:**

For the experiments it would be helpful to compare with ordinary gradient descent on the variational parameters, to provide some ideas on the practical utility.

**Limitations:**

Yes.

---

> ### Author Rebuttal · Authors · 2023-08-08
>
> Thanks for the detailed review and incisive questions.
>
> 1. **Restrictions to Gaussian families and affine kernels:**
>    - For a large class of Bayesian inference problems, thanks to the Bernstein von Mises theorem, the posterior (target) distribution is approximately Gaussian in the limit of large samples under appropriate regularity conditions. Notably a recent work [A] shows that in terms of mean and covariance estimation (of the posterior) using GVI has improved rates over other methods like Laplace approximation. There is an array of works providing **stochastic** algorithms (which is based on randomized discretizations of the BW Gradient flow) for solving GVI (see, e.g., [17, 36]), and [55, 68, 70, 57]  show the improved practical performance of GVI on various practical problems. We show in this work that there is a surprising **deterministic** discretization (BWPF) of the BW Gradient flow if we aim to solve GVI. Furthermore, we also have **quantitative rates** for mean and covariance estimation for **deterministic** SVGD algorithm in the finite-particle discrete-time setting (which we describe next), similar to the results for the **stochastic** algorithms mentioned above.
>
> [A] A Katsevich, P Rigollet. "On the approximation accuracy of gaussian variational inference."
> arXiv preprint arXiv:2301.02168, 2023.
>
>   - In terms of assumptions on kernel in prior works on SVGD, we also highlight that prior work by [49] requires that the kernel be radial which rules out the important class of bilinear kernels that we consider. The work of [27]  relaxed the radial kernel assumption. However, they required boundedness assumptions which we avoid in this work for the case of bilinear kernels.
>
>   **To sum up, understanding Gaussian-SVGD is an important step towards understanding the general situations with other kernels (note here that flexibilty offered by fully-nonparametric SVGD suffers from the issue of kernel choice). Surprisingly this special case has not been well studied in the literature and our work provides nearly-complete solution to this important special case.**
>
> 2. **About discrete-time, finite-particle guarantees:**
> - We have an updated version of Theorem 3.9 (discrete-time, finite-particle) with the following **quantitative** convergence rates:
>
>      **Theorem:** For a centered Gaussian target, suppose the SVGD particle system with $K_{1}$ or $K_{2}$ is initialized by $\bigl(\boldsymbol{x}\_{i}^{(0)}\bigr)\_{i=1}^{N}$ such that $\boldsymbol{\mu}\_{0}=\boldsymbol{0}$ and $C_{0}Q=QC\_{0}$. For $0<\epsilon< 0.5$, we have $\boldsymbol{\mu}\_{t}=\boldsymbol{0}$ and $\lVert C_{t}- Q \rVert\to 0$ as long as all the eigenvalues of $Q^{-1}C\_{0}$ lie in the interval $(0, 1+1/\epsilon)$. Furthermore, if we set $u_{\epsilon}$ to be the smaller root of the equation $f\_{\epsilon}'(u)=1-\epsilon$ (it has $2$ distinct roots) where $f\_{\epsilon}(x) := (1 + \epsilon(1-x))^2x$, then we have linear convergence, i.e.,
> $$
> \lVert C\_{t}- Q \rVert\leq (1-\epsilon)^{t}\lVert C_{0}-Q \rVert\leq e^{-\epsilon t}\lVert C\_{0}-Q \rVert
> $$
>
> as long as all the eigenvalues of $Q^{-1}C\_{0}$ lie in the interval $[u\_{\epsilon}, 1/3+1/(3\epsilon)]$.
>
>
>   The above result is essentially direct refinement of the current result in Theorem 3.9. and is the first such result for finite-particle, discrete-time setting which matches the observed empirical performance. The only other comparable result for **deterministic** SVGD is by [62], where the results are not in alignment with practice. Obtaining similar results for general targets is extremely challenging and is one of the central open problems regarding SVGD. In view of this, our result provides a first concrete step towards developing a theory of SVGD that aligns with practice.
>
> We also remark here that the fully algorithmic results for GVI in e.g., [17, 36] are for **stochastic** algorithms and are only proved in expectation.
>
>
> 3. **About ordinary gradient descent:** We thank the reviewer for this great suggestion. We will be happy to add this result in our revision.

---

> > ### Comment · Reviewer_ZXYv · 2023-08-20
> >
> > Thank you for your response, and I especially appreciate the added result on discrete-time finite-particle convergence.  I am keeping my score unchanged to reflect the fact that I cannot vouch on the technical contributions of the results.  I believe the work would be greatly strengthened if the authors could either
> > - add comparisons to OGD-implemented GVI in more practical scenarios (with minibatching, or on larger-scale datasets with computational constraints), or
> > - discuss in more detail how Gaussian-SVGD may help with the analysis of SVGD with general kernels,
> >
> > in which case the contribution towards a broader audience would be more clear.  On the latter point, I would note that I'm more positive with the use (and analysis) of affine kernels due to the connection to [46] and GVI, but radial kernels still appear to be the more viable choice if a non-Gaussian variational family is needed, which constitute an important scenario.

---

> > > ### Author Response · Authors · 2023-08-20
> > > **Official reply by Authors**
> > >
> > > Thanks for your acknowledgement.
> > >
> > > 1) Indeed, thanks to your earlier suggestion, we are currently running the asked comparison experiments to add in our revision. We will be happy to add this in the revision.
> > >
> > > 2) It is currently unclear to us what exact family of distributions are characterized (either in a parametric or non-parametric way) by using the RBF kernel. This is generally related to lack of deep understanding of SVGD itself. However, we are examining Elliptical-SVGD, i.e., with the metric being the Bures-Wasserstein metric corresponding to the family of elliptical distribution (instead of Gaussian as in our current work); see [A,B] for details. The kernel in this case could be obtained here by a delicate and careful reverse-computation. The set of tools and insights we developed in this work are extremely useful to establish similar results (both the discrete and continuous settings) for the problem of Elliptical Variational Inference, which has applications for performing variational inference under heavy-tails [C]. While the technical details are ​in the initial phases of an ongoing work (and are definitely beyond the scope of th​is submission,​ and require a separate paper​ to be completely presented), taking your suggestion, we will be happy to discuss this briefly in this draft​, as a potential direction for generalization of our work to non-Gaussian variational inference.
> > >
> > > A]-Muzellec, Boris, and Marco Cuturi. "Generalizing point embeddings using the Wasserstein space of elliptical distributions." Advances in Neural Information Processing Systems 31 (2018).
> > >
> > > [B]-Matthias Gelbrich. On a formula for the l2 Wasserstein metric between measures on Euclidean and Hilbert spaces. Mathematische Nachrichten,147(1):185–203, 1990.
> > >
> > > [C]-Domke, Justin, and Daniel R. Sheldon. "Importance weighting and variational inference." Advances in neural information processing systems 31 (2018).

---

### Author Rebuttal · Authors · 2023-08-08

General Clarifications
=================

1.**Quantitative results for finite particle, discrete-time setting**

We have an updated version of Theorem 3.9 (discrete-time, finite-particle) with the following **quantitative** convergence rates:

 **Theorem:** For a centered Gaussian target, suppose the SVGD particle system with $K_{1}$ or $K_{2}$ is initialized by $\bigl(\boldsymbol{x}\_{i}^{(0)}\bigr)\_{i=1}^{N}$ such that $\boldsymbol{\mu}\_{0}=\boldsymbol{0}$ and $C_{0}Q=QC\_{0}$. For $0<\epsilon< 0.5$, we have $\boldsymbol{\mu}\_{t}=\boldsymbol{0}$ and $\lVert C_{t}- Q \rVert\to 0$ as long as all the eigenvalues of $Q^{-1}C\_{0}$ lie in the interval $(0, 1+1/\epsilon)$. Furthermore, if we set $u_{\epsilon}$ to be the smaller root of the equation $f\_{\epsilon}'(u)=1-\epsilon$ (it has $2$ distinct roots) where $f\_{\epsilon}(x) := (1 + \epsilon(1-x))^2x$, then we have linear convergence, i.e.,
$$
\lVert C\_{t}- Q \rVert\leq (1-\epsilon)^{t}\lVert C_{0}-Q \rVert\leq e^{-\epsilon t}\lVert C\_{0}-Q \rVert
$$

as long as all the eigenvalues of $Q^{-1}C\_{0}$ lie in the interval $[u\_{\epsilon}, 1/3+1/(3\epsilon)]$.


This is a direct refinement of Theorem 3.9 and is the 1st such result for finite-particle, discrete-time setting matching practice. The only other rates (for deterministic SVGD) by [62] do not align with practice. Obtaining similar results for general targets is extremely challenging and is one of the central open problems in SVGD.


2.**Gaussian-SVGD versus bilinear SVGD**

We like to clarify that our algorithm (i.e, Gaussian-SVGD) is **different** from SVGD with a bilinear kernel, when the target is not Gaussian.  Standard (bilinear or RBF) SVGD can sample from the target but does not solve Gaussian variational inference (GVI) whereas Gaussian-SVGD can solve GVI. To achieve this, in Equation (26) we use a linear approximation $\widehat{\nabla V}$ instead of the actual $\nabla V$. (Note that this linear approximation does not guarantee Gaussian dynamics for RBF kernel.)

- If one wants to perform GVI, one essentially need to use Gaussian-SVGD instead of bilinear SVGD or RBF-SVGD.

- To estimate the posterior mean or covariance of a **not necessarily Gaussian target**, one can use either Gaussian-SVGD or standard (bilinear or RBF) SVGD. The particle-implementation of Gaussian-SVGD provides a **deterministic** algorithm for GVI, and is shown to perform well in our experiments. Furthermore, [A] shows the GVI provably does better than many other existing methods like Laplacian approximation. In contrast, there are no such theoretical guarantees for bilinear SVGD, RBF-SVGD or general SVGD in this settings.


[A] A Katsevich, P Rigollet. "On the approximation accuracy of gaussian variational inference." arXiv preprint arXiv:2301.02168, 2023.

3.**Further clarifications from prior works**

We now explain further the advantages of studying the Gaussian-SVGD and the different types of results available in the literature on SVGD. First note that the flexibility offered by the \emph{nonparametric} aspect of SVGD also leads to unintended consequences. Indeed, on one hand, from a practical perspective, the question of how to pick the right kernel for implementing the SVGD algorithm is unclear. Existing approaches are mostly ad-hoc and do not provide clear instructions on the selection of kernels. On the other hand, developing a deeper theoretical understanding of SVGD dynamics is challenging due to its nonparametric formulation. [49] derived the continuous-time PDE for the evolving density that emerges as the mean-field limit of the finite-particle SVGD systems, and shows the well-posedness of the PDE solutions. Furthermore, the following different types of convergences could be examined regarding SVGD, some of which has been analyzed previously in the literature:

- (A) Unified convergence of the empirical measure for $N$ finite particles to the continuous target as time $t$ and $N$ jointly grow to infinity;

- (B) Convergence of mean-field SVGD to the target distribution over time;
- (C) Convergence of the empirical measure for finite particles to the mean-field distribution at any finite given time $t\in [0,\infty)$;
- (D) Convergence of finite-particle SVGD to the equilibrium over time;
- (E) Convergence of the empirical measure for finite particles to the continuous target at time $t=\infty$.

From a practical point of view (A) is the ideal type of result that fully characterizes the algorithmic behavior of SVGD, which could be obtained by combining either (B) and (C) or (D) and (E). Regarding (B), [44] showed the convergence of mean-field SVGD in kernel Stein discrepancy, which is known to imply weak convergence under appropriate assumptions. The works of [19],[34],[60] and [65] sharpened the results with weaker conditions or explicit rates. The work [27]  extended the above result to the stronger Fisher information metric and Kullback-Leibler divergence based on a regularization technique. The works [44] and [34] obtained time-dependent mean-field convergence (C) of $N$ particles under various assumptions using techniques from the literature of propagation of chaos. The work of [62] obtained even stronger results for (C) and combined (B) to get the first unified convergence (A) in terms of KSD. However, they have a rather slow rate $1/\sqrt{\log\log N}$, resulting from the fact that their bounds for (C) still depends on the time $t$ (sum of step sizes) double-exponentially. Moreover, there has not been any work that studies the convergence (D) and (E) for SVGD, which illustrate a new way to characterize the unified convergence (A).

4.**Prior work** by [49] requires that the kernel be radial which rules out the important class of bilinear kernels that we consider. The work of [27]  relaxed the radial kernel assumption. However, they required boundedness assumptions which we avoid in this work for the case of bilinear kernels.

---

### Author Response · Authors · 2023-08-18
**Reviewer 1EPQ**

Dear AC,

Apologies for the second reminder. However, Reviewer 1EPQ hasn't responded yet to our rebuttal, even after sending a reminder. While they provided the lowest score to your submission, we have answered all their questions and clarified several misunderstandings on their part. Furthermore, the discussion period is inching towards the deadline (Aug 21st) with also the weekend in between, which makes us even more anxious about their non-response.

We are sending this email to you and also tagging the reviewers in the hope that we get a response. In the event that Reviewer 1EPQ doesn't respond in a timely manner, we would greatly appreciate it if you could ignore their review when making the final decisions regarding our submission.

Thanks for your time and consideration.

Sincerely,

Authors

---

> ### Comment · Area_Chair_WWWY · 2023-08-19
> **Re: Reviewer 1EPQ**
>
> Dear Authors,
>
> It seems that the reviewer has now responded to the rebuttal. Please take time to read those and comment if needed.
>
> Best,
> AC

---

> > ### Author Response · Authors · 2023-08-19
> >
> > We would like to extend our sincere gratitude for taking your time reading the note and encouraging timely response from reviewers. We thank all reviewers again for their constructive feedback.
> >
> > In particular, we appreciate reviewer 1EPQ's efforts in re-examining our technical details. However, we are writing this letter because we feel compelled to express our concern about their comments as they challenge our work either **without careful scrutiny** (the first time) or **with strong prejudice against our problem setup** (the second time), which could have a negative impact on their evaluations.
> >
> > Moreover, reviewer 1EPQ thinks our presentation were *"sloppy"* (not even fair) while all the other four reviewers rated this particular dimension as good or excellent, which inevitably makes us slightly question their expertise in this field.
> >
> > We provide specific arguments in response to their feedback as follows:
> >
> > Since they mentioned *"practical"* issues multiple times, we would like to point out that **practical algorithms are primarily proposed to better solve a problem**. Our practical contribution in this work is the **first deterministic particle-based algorithm** for Gaussian Variational Inference (GVI). Obtaining a complete theory could take years. In this current work, we have already provided appealing theoretical results for the special case of Gaussian targets (even in the discrete-time and finite-particle setting), along with empirical evidence showing superior performance for general targets. While a complete theory would be ideal, it is unfair to judge our algorithmic contribution as *what they called "useless"* simply for the reason that there is currently no complete general bound. Note that even after more than 7 years since the publication of the SVGD paper by Liu \& Wang, there is no complete theory on understanding it. The work of Shi \& Mackey 2022 might be one of the best results so far but is still far from alignment with practice.
> >
> > Also, reviewer 1EPQ seemed to have strong objections against **bilinear kernels**. As we had made clear in the rebuttal, in Gaussian-SVGD we use bilinear kernels to perform GVI. BWGD (Lambert et al) re-emerged as one of the algorithms under our (bilinear kernel) framework and FB-GVI (Diao et al) improved upon BWGD. If the bilinear kernel were really impractical as the reviewer had claimed, these two previous methods should have been regarded as impractical as well.  And by similar logic, one can even argue that the whole field of GVI were impractical. Furthermore, regarding FB-GVI, the reviewer's claim that **"high-probability guarantees should follow easily"** is not fully correct. It would need strong assumptions on the stochastic gradients to derive a high-probability bound whereas our results are entirely **deterministic**. This further highlights the advantage in having an algorithm with deterministic particle updates for GVI, complementing the stochastic algorithms like BWGD and FB-GVI.
> >
> > **Regarding Theorem 3.7**, as we had pointed out in the rebuttal, even for the continuous time setting, we provide the **first uniform-in-time** propagation of chaos result. It is unfair to say that there is lack of contribution due to the continuity of time. Apparently **uniform vs non-uniform** is more significant than **continuous vs discrete**, and getting a continuous-time result is always a first step towards understanding discrete-time dymanics. **Last but not least**, the idea of "uniform-in-time" comes from the general observation that the difference of some summary statistics between two particle systems (governed by the same ODE) with different initializations reduces after some bounded time rather than keeps growing exponentially in time. Although our current proof relies on the precise dynamics, the idea is generalizable (e.g. from one particular ODE to an ODE class) and **definitely NOT an overclaim**.
> >
> > There are other minor points from their response that do not sound very reasonable to us. For example, they claimed that sampling Gaussian with known $b$ and $Q$ is *"unrealistic"*, and carefully clarified that $b$ and $Q$ are indeed known in our paper. However, the usual setup of sampling or GVI is exactly that **the target density is known up to a normalization constant** (e.g. see the SVGD paper by Liu \& Wang).
> >
> > In summary, we **have serious doubts about the reviewing principles of reviewer 1EPQ**. On one hand, they *sentenced (please forgive us for using this word due to their sarcastic tone)* our algorithm as impractical only based on the theoretical setting rather than experiments, which is **not a justified way to evaluate algorithms**. On the other hand, while our theoretical results aim to provide new insights to SVGD and GVI, they seemed to believe that theoretical insights are *"useless"* unless they immediately buy you results for general settings.
> >
> > We sincerely hope that you could take our concerns into consideration and make a fair judgment!

---

### Decision · Program_Chairs · 2023-09-21

**Decision:**

Accept (poster)

**Comment:**

There has been thorough discussion of this paper in the reviewing process. The main object being studied is a sampling algorithm called Gaussian SVGD, i.e., SVGD with bilinear kernels and Gaussian initialization. There are two main theoretical contributions:

* A **continuous-time** analysis of **mean-field** Gaussian-SVGD for general targets (linear convergence for both gaussian and strongly log-concave targets)
* Both continuous and discrete-time analysis of **finite-particle** Gaussian-SVGD for **Gaussian targets** (quantitative rate provided in rebuttal)

While concerns were raised regarding the finite-particle analysis being specific to Gaussian targets, the solid theoretical contribution of the first part analyzing general targets satisfied the reviewers. Additionally, it was noted that the uniform-in-time propagation of chaos results, though for Gaussian targets, improve upon prior work. Based on these, I recommend acceptance.